# Amortized Bayesian Inference with Hybrid Expert-in-the-Loop and Learnable Summary Statistics

## Abstract

Amortized Bayesian inference (ABI), a subset of simulation-based inference (SBI) fueled by neural networks, has rapidly grown in popularity across diverse scientific fields. Summary statistics are an essential dimensionality reduction component of ABI workflows and most methods to-date rely either on hand-crafted (i.e., based on domain expertise) or end-to-end learned summary statistics. In this work, we explore three hybrid methods to harness the complementary strengths of both sources. The first method directly conditions a neural approximator on both summary types, thereby extending traditional end-to-end approaches in a straightforward way. The second method embeds both expert and learned summaries into a joint representation space which is explicitly optimized to encode decorrelated features. The third method employs an auxiliary generative model to learn a latent summary representation that is statistically independent from the expert summaries. We explore various aspects of our hybrid methodology across different experiments and model instances, including perfect domain expertise and imperfect artificial experts represented by pre-trained neural networks. Our empirical results suggest that hybrid representations can improve parameter estimation and model comparison in settings of scientific interest, warranting the viability of an "expert-in-the-loop" approach. The performance gains are especially promising in scenarios with low to medium simulation budgets.

## 1 Introduction

Computer simulations are becoming indispensable across the sciences (Lavin et al., 2021). Yet, the *inverse problem* of determining simulation parameters that faithfully emulate or forecast real-world phenomena tends to be arduous and analytically intractable. Generative neural networks are a promising new contender for solving inverse problems through simulation-based inference (SBI, Cranmer et al., 2020; Ardizzone et al., 2018). When trained on model simulations, these networks can perform scalable *amortized* Bayesian inference (ABI) on unseen real data (e.g., Avecilla et al., 2022; Radev et al., 2021b; Gonçalves et al., 2020). ABI has been shown to speed up industrial processes (Zhang & Mikelsons, 2023a) and scale up Bayesian estimation to large data sets (von Krause et al., 2022). However, two major challenges for ABI tend to emerge in practice:

1. Accurate approximate inference given limited simulation budgets;
2. Sufficiently informative data compression given complex structure.

The first challenge arises when the *forward problem* of simulating a real-world process is computationally expensive and cannot be run *ad infinitum* in the context of online learning. Many interesting models and recent applications of SBI fall in this category (e.g., Zhang & Mikelsons, 2023b;a; Zeng et al., 2023; Vandegar et al., 2021). The second challenge arises from the very essence of (Bayesian) model-based reasoning, which seeks to recover informative low-dimensional representations of high-dimensional data with rich structure. In statistics and Bayesian inference, in particular, such representations are commonly termed *summary statistics*; good summary statistics respect the probabilistic symmetry of the data and achieve nearly lossless compression for downstream inference targets (Bloem-Reddy & Teh, 2020). However, good summary statistics are notoriously hard to engineer in practice (Bharti et al., 2022; Marin et al., 2018; Blum et al., 2013).

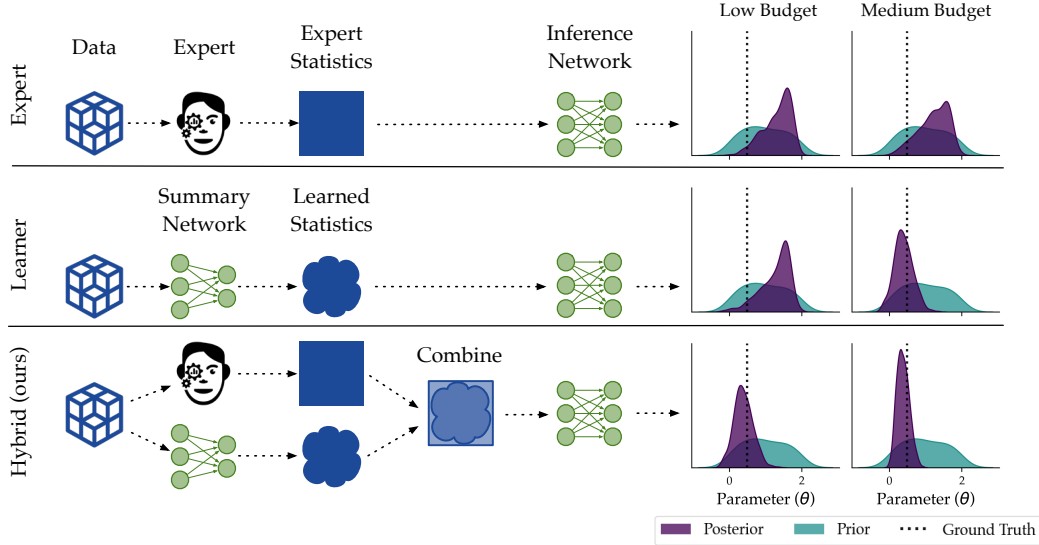

Figure 1: A conceptual illustration of neural simulation-based inference (SBI) with three different methods for obtaining summary statistics. **Top row**. In *expert* approaches, the data are reduced through hand-crafted summary statistics, leveraging domain expertise. **Middle row**. End-to-end *learner* approaches involve passing the data through a neural network (i.e., *summary network*) to acquire learned summaries. **Bottom row**. Hybrid approaches (ours) combine both sources of summary statistics (*expert* and *learned*). Regardless of the approach, summary statistics are then processed by an inference network which solves a Bayesian task (e.g., posterior estimation). The right panel displays posterior distributions for a single parameter of a complex neurocognitive model (**Experiment 2**) obtained using all three approaches with two separate simulation budgets (*low*; *medium*). The hybrid approach leads to more precise estimation in both settings.

Most SBI applications to-date rely on either hand-crafted (i.e., based on domain expertise) or end-to-end learned summaries, with the latter approach being employed even in scenarios with only a few hundred simulations available (e.g., Zhang & Mikelsons, 2023b; Heringhaus et al., 2022; Zeng et al., 2023; Tsilifis et al., 2022). Expert summary statistics may be "leaky" and contain insufficient information for learning to infer from data (cf. Figure 1, top). Differently, end-to-end learning methods typically delegate the task to a neural network which may struggle in settings with low simulation budgets but improve as simulation throughput increases (cf. Figure 1, middle). Crucially, the design space of this summary network becomes a key hyperparameter in any such analysis and an underperforming network may be just as detrimental as poor expert information. Thus, we propose to explore an integrative approach which seamlessly combines both expert and learned statistics into ABI workflows (cf. Figure 1, bottom). In summary, our contributions are:

- an investigation of novel hybrid methods of varying complexity to synthesize expert and learnable summary statistics in ABI tasks;

- a comprehensive evaluation of the relative benefits of expert, learned, and hybrid summary statistics for different models, data types, and simulation budgets;

- an empirical demonstration that the synthesis of expert knowledge and learned summaries can improve ABI, especially in settings with low to medium simulation budgets.

## 2 BACKGROUND

### 2.1 BAYESIAN INFERENCE WITH NEURAL NETWORKS

Given a collection of observations $\boldsymbol{X}$ (i.e., data) and an observation model $p(\boldsymbol{X} \,|\, \boldsymbol{\theta})$ (i.e., a likelihood density), the goal of Bayesian modeling is to approximate the posterior distribution over hidden

(unobserved or unobservable) parameters $\boldsymbol{\theta}$

$$p(\boldsymbol{\theta} \mid \boldsymbol{X}) = \frac{p(\boldsymbol{X} \mid \boldsymbol{\theta})\, p(\boldsymbol{\theta})}{Z} \tag{1}$$

as accurately as possible. Analytic solutions are typically intractable for most non-trivial models, since the normalizing constant, $Z = \mathbb{E}_{p(\boldsymbol{\theta})}\left[p(\boldsymbol{X} \mid \boldsymbol{\theta})\right]$, involves integration over a high-dimensional prior $p(\boldsymbol{\theta})$. Thus, a lot of research in Bayesian modeling has focused on developing methods for efficient *conditional density estimation*, with Markov chain Monte Carlo (MCMC) methods currently standing as the gold standard in terms of theoretical properties and practicability (Plummer, 2023; Carpenter et al., 2017). Nevertheless, a major challenge for MCMC methods remains their subpar efficiency for large data sets, complex models, or real-time inference.

Bayesian modeling becomes even more challenging when the likelihood density $p(\boldsymbol{X} \mid \boldsymbol{\theta})$ is not available in closed-form but only implicitly defined through a randomized simulation program $G$. In that case, most Monte Carlo approaches are not applicable, because even the (log) unnormalized posterior $p(\boldsymbol{X} \mid \boldsymbol{\theta})\, p(\boldsymbol{\theta})$ cannot be evaluated explicitly. Thus, SBI has emerged as a general solution to analyze intractable models (Cranmer et al., 2020). More recently, *neural density estimation* has entered the scene as a flexible framework for modeling complicated distributions (Kobyzev et al., 2020; Papamakarios et al., 2021) and, by extension, conditional distributions arising in Bayesian modeling.

This work focuses on ABI methods which aim to directly approximate $p(\boldsymbol{\theta} \mid \boldsymbol{X})$ by training a generative network $q_\phi(\boldsymbol{\theta} \mid \boldsymbol{X})$ on model simulations $\{\boldsymbol{X}, \boldsymbol{\theta}\}$ (e.g., Ardizzone et al., 2018; Gonçalves et al., 2020). Once trained, such networks can approximate the posteriors for any set of new unlabeled observations $\{\boldsymbol{X}^{\text{new}}\}$ compatible with the model's configuration. Thus, the computational cost of the simulation-based training phase quickly *amortizes* over repeated inference queries, hence the term *amortized inference*. Furthermore, we suspect that our ideas will also apply to neural likelihood estimation (Papamakarios et al., 2019; Kelly et al., 2023), which targets the intractable likelihood density $p(\boldsymbol{X} \mid \boldsymbol{\theta})$ with the goal of applying MCMC with a neural surrogate.

## 2.2 SUMMARY STATISTICS AND SUFFICIENCY

The data $\boldsymbol{X}$ in Bayesian problems usually exhibits some exploitable probabilistic symmetry; common data structures comprise sets of IID vectors, time series, or graphs (Orbanz & Roy, 2014). Thus, even the simplest Bayesian models make use of *summary statistics*. Consider a toy conjugate Gaussian model given by

$$\text{Prior: } \theta \sim \mathcal{N}(0, 1), \quad \text{Likelihood: } x_n \sim \mathcal{N}(\theta, 1) \text{ for } n = 1, \dots, N. \tag{2}$$

The posterior of the Gaussian mean parameter $\theta$ is proportional to

$$p(\theta \mid x_1, \dots, x_N) \propto \mathcal{N}\left(\frac{N}{N+1}\, \bar{x}, \frac{1}{N+1}\right). \tag{3}$$

Thus, to estimate the posterior of $\theta$ for any fixed $N$, we can simply compute the sample mean, $\bar{x} = N^{-1} \sum_{n=1}^{N} x_n$, and throw away the raw data. In this case, the sample mean $\bar{x}$ is a *sufficient summary statistic*, as it allows *lossless compression* of the data for the estimation problem at hand. In other words, sufficient statistics need not reconstruct the data $\boldsymbol{X}$ perfectly; they need to capture the relevant information in $\boldsymbol{X}$ for consistent inference of $\boldsymbol{\theta}$. More generally, a sufficient summary statistic, $S(\boldsymbol{X})$, is one that satisfies the following equality:

$$p(\boldsymbol{\theta} \mid \boldsymbol{X}) = p(\boldsymbol{\theta} \mid S(\boldsymbol{X})) \tag{4}$$

Sufficient summary statistics exist for textbook Bayesian models (e.g., as in our Gaussian example above), but are typically hard to obtain in practice.[1] Indeed, insufficient summary statistics have been long known to plague approximate Bayesian inference (Robert et al., 2011), a predicament which has been dubbed *the curse of insufficiency* (Marin et al., 2018). Thus, the design of good summary statistics in SBI is essential and has attracted much research from various areas (see **Related Work**).

---

[1] *Maximally informative statistics* merely minimize some distance between $p(\boldsymbol{\theta} \mid \boldsymbol{X})$ and $p(\boldsymbol{\theta} \mid S(\boldsymbol{X}))$ and are thus "locally optimal" under the constraints imposed by the nature of the optimization task.

## 3 METHODS

### 3.1 HYBRID SUMMARY STATISTICS

Given raw simulator outputs or real data $\boldsymbol{X} \in \mathcal{X}^N$, with $\mathcal{X}$ not necessarily an Euclidean space, we consider two types of summary functions. The first is parameterized via a *learner* (i.e., a neural network), $L_\vartheta(\boldsymbol{X})$, $L_\vartheta : \mathcal{X}^N \to \mathbb{R}^D$, which appropriately encodes the probabilistic symmetry (e.g., permutation invariance) of the data into a compressed representation. For instance, we can use set transformers (Lee et al., 2019) for learning permutation-invariant representations of set-based data (**Experiments 1** and **2**) or temporal fusion transformers (Lim et al., 2021) for compressing multivariate time series data (**Experiment 3**). The second summary function $H(\boldsymbol{X})$, $H : \mathcal{X}^N \to \mathcal{H}$, is defined by a human or an artificial *expert* tasked to reduce the data based on their domain expertise (e.g., providing quantiles, moments, or predicted labels). The image $\mathcal{H}$ of $H$ may not necessarily conform to an Euclidean space, as the expert may provide a mixture of categorical, ordinal, and continuous summaries. In the following, we discuss how to combine these distinct representations of the same $\boldsymbol{X}$ such that an *inference network* can infer $\boldsymbol{\theta}$ from the collective information contained in $L_\vartheta(\boldsymbol{X})$ and $H(\boldsymbol{X})$.

### 3.2 INFERENCE NETWORKS

To enable amortized ABI with neural networks, we can utilize any conditional neural density estimation family, such as normalizing flows (Papamakarios et al., 2021), score-based models (Geffner et al., 2022), or flow-matching methods (Chen & Lipman, 2023). Here, we use conditional normalizing flows, $F_\phi$, which represent the parameter posterior (Eq. 1) via the change-of-variables formula

$$q_\phi(\boldsymbol{\theta} \mid \boldsymbol{X}) = p(\boldsymbol{z} = F_\phi(\boldsymbol{\theta}; \boldsymbol{X})) \left| \frac{\partial F_\phi}{\partial \boldsymbol{\theta}} \right|, \tag{5}$$

where $p(\boldsymbol{z})$ is a simple base distribution from which we can easily draw samples (e.g., spherical Gaussian). Since the data $\boldsymbol{X}$ can have varying shape, size, or a different structure from $\boldsymbol{\theta}$, we cannot simply concatenate $\boldsymbol{\theta}$ and $\boldsymbol{X}$, and enter them into $F_\phi(\boldsymbol{\theta}; \boldsymbol{X})$. Thus, we need $H(\boldsymbol{X})$ and/or $L_\vartheta(\boldsymbol{X})$ not merely as a form of convenient dimensionality reduction, but as a proxy for making inference networks *flexible and practically applicable* by being agnostic to the peculiarities of $\boldsymbol{X}$'s structure.

### 3.3 DIRECT HYBRID METHOD

Our first hybrid method directly conditions the networks on $\boldsymbol{s}_\vartheta := [L_\vartheta(\boldsymbol{X}), H(\boldsymbol{X})]$. A straightforward optimization criterion for amortized inference is thus

$$\phi^*, \vartheta^* = \arg\min_{\phi, \vartheta} \mathbb{E}_{p(\boldsymbol{X})} \left[ D_{\mathrm{KL}} \left[ p(\boldsymbol{\theta} \mid \boldsymbol{X}) \,\|\, q_\phi(\boldsymbol{\theta} \mid \boldsymbol{s}_\vartheta)) \right] \right], \tag{6}$$

where $D_{\mathrm{KL}}(\cdot \,\|\, \cdot)$ above is the Kullback-Leibler divergence between the analytic (true) and the approximate posterior. This criterion reduces to a tractable, doubly conditional maximum likelihood training objective

$$\phi^*, \vartheta^* = \arg\min_{\phi, \vartheta} \mathbb{E}_{p(\boldsymbol{\theta}, \boldsymbol{X})} \left[ -\log q_\phi(\boldsymbol{\theta} \mid L_\vartheta(\boldsymbol{X}), H(\boldsymbol{X})) \right], \tag{7}$$

which we can approximate through batches of simulations $\{\boldsymbol{X}, \boldsymbol{\theta}\} \sim G$. If we condition only on the learned summaries $L_\vartheta(\boldsymbol{X})$, we recover the criterion from Radev et al. (2020a) which has been shown to automatically approximate maximally informative summary statistics for $\boldsymbol{\theta}$. Intuitively, when we also input the expert summaries, the summary network is implicitly encouraged to learn the *incremental* information about $\boldsymbol{\theta}$ contained in $\boldsymbol{X}$, which is not captured by the expert $H(\boldsymbol{X})$. Note, that the direct hybrid method features no direct structural connection between the expert and learned summaries. Thus, our next two methods focus on prescribing an explicit structure to the implied joint $p(L_\vartheta(\boldsymbol{X}), H(\boldsymbol{X}))$ and conditional $p(L_\vartheta(\boldsymbol{X}) \mid H(\boldsymbol{X}))$ distributions, respectively.

### 3.4 JOINT HYBRID METHOD

What if we wanted to learn a *decorrelated joint representation* of the expert and the learnable summaries, that is, a representation with pairwise independent components? One way to approach

this problem is to attempt to model $p(L_\vartheta(\boldsymbol{X}), H(\boldsymbol{X}))$ directly via a latent generative model with jointly independent latent codes. However, differently from $L_\vartheta(\boldsymbol{X})$, the expert summaries $H(\boldsymbol{X})$ may reside in a very different space (e.g., be discrete or exhibit artificial bounds), resulting in $p(L_\vartheta(\boldsymbol{X}), H(\boldsymbol{X}))$ being a complicated mixture distribution. In addition, the expert may provide a large number of partially redundant summary statistics (Raynal et al., 2019) which themselves may profit from some form of pre-processing or compression.

For the above reasons, we propose to introduce a further learnable transformation, $T_\varphi : \mathcal{E} \to \mathbb{R}^{D'}$, which maps the expert summaries into real vectors $\boldsymbol{s}_\varphi = T_\varphi \circ H(\boldsymbol{X})$ We can interpret these vectors as *expert embeddings*. In practice, we can implement a simple linear transformation or a small neural network realizing a non-linear transformation. Thus, we can obtain joint representations $\boldsymbol{s}_\vartheta^{(\varphi)} := [L_\vartheta(\boldsymbol{X}), T_\varphi(H(\boldsymbol{X}))]$ and enforce a spherical structure on the resulting joint summary space. To achieve the latter, we augment our original criterion (Eq. 7) by the kernel-based Maximum Mean Discrepancy (MMD, Gretton et al., 2012) between the joint summary representation and a spherical Gaussian:

$$\phi^*, \vartheta^*, \varphi^* = \underset{\phi, \vartheta, \varphi}{\arg\min} \, \mathbb{E}_{p(\boldsymbol{\theta}, \boldsymbol{X})} \left[ -\log q_\phi(\boldsymbol{\theta} \mid \boldsymbol{s}_\vartheta^{(\varphi)}) \right] + \lambda \cdot \mathrm{MMD}^2(p(\boldsymbol{s}_\vartheta^{(\varphi)}) \,||\, \mathcal{N}(\boldsymbol{0}, \mathbb{I})) \qquad (8)$$

Here, $\lambda$ is a hyperparameter which controls the importance of having a simple and decorrelated joint summary space. Setting $\lambda = 0$ recovers our direct objective (Eq. 7) with an additional (unconstrained) learnable transformation of the expert summaries. The advantage of using the $\mathrm{MMD}^2$ estimator is that we can easily compute its Monte Carlo estimate based on samples from $\mathcal{N}(\boldsymbol{0}, \mathbb{I})$ and $p(\boldsymbol{s}_\vartheta^{(\varphi)})$. In addition to encouraging a decorrelated representation, the inclusion of a summary regularizer may help to highlight potential simulation gaps via feature-based out-of-distribution (OOD) detection in real-data applications of ABI (Schmitt et al., 2021). A disadvantage of this approach is that having a simple and pairwise decorrelated *joint summary distribution* may be too strict or even an impossible requirement to fulfill in the presence of many redundant dimensions.

### 3.5 CONDITIONAL HYBRID METHOD

Finally, what if we wanted to learn summary representations that are guaranteed to be statistically independent from the expert summaries without transforming the latter in any way? To achieve this requirement, we propose to learn a conditional model over $p(L_\vartheta(\boldsymbol{X}) \mid H(\boldsymbol{X}))$ which we parameterize through an auxiliary generative network $\boldsymbol{z}_\vartheta^{(\psi)} := F_\psi(L_\vartheta(\boldsymbol{X}); H(\boldsymbol{X}))$ with a simple base distribution. We can then train all networks (*inference*, *summary*, *auxiliary*) together by alternating between the following two optimization criteria

$$\phi^*, \vartheta^* = \underset{\phi, \vartheta}{\arg\min} \, \mathbb{E}_{p(\boldsymbol{\theta}, \boldsymbol{X})} \left[ -\log q_\phi(\boldsymbol{\theta} \mid \boldsymbol{z}_\vartheta^{(\psi)}, H(\boldsymbol{X})) \right] \qquad (9)$$

$$\psi^* = \underset{\psi}{\arg\min} \, \mathbb{E}_{p(\boldsymbol{X})} \left[ -\log q_\psi(L_\vartheta(\boldsymbol{X}) \mid H(\boldsymbol{X})) \right]. \qquad (10)$$

The first objective (Eq. 9) is *almost the same* as the one in Eq. 7, with the output of the summary network $L_\vartheta(\boldsymbol{X})$ replaced with its latent representation $\boldsymbol{z}_\vartheta^{(\psi)}$ generated by the auxiliary network $F_\psi$. Importantly, this method does not impose any restrictions on the outputs of $L_\vartheta(\boldsymbol{X})$ themselves, nor does it additionally transform the expert summaries $H(\boldsymbol{X})$. Moreover, it encourages i) maximally informative latent learned summaries $\boldsymbol{z}_\vartheta^{(\psi)}$ for $\boldsymbol{\theta}$ and ii) statistical independence between the learned and the expert summaries $H(\boldsymbol{X})$. Thus, optimal convergence under Eqs. 9 and 10 implies

$$I(\boldsymbol{\theta}, \boldsymbol{z}_{\vartheta^*}^{(\psi^*)} \mid H(\boldsymbol{X})) = \max_\vartheta I(\boldsymbol{\theta}, \boldsymbol{z}_\vartheta^{(\psi^*)} \mid H(\boldsymbol{X})) \qquad (11)$$

$$\boldsymbol{z}_{\vartheta^*}^{(\psi^*)} \perp\!\!\!\perp H(\boldsymbol{X}), \qquad (12)$$

where $I(\cdot, \cdot)$ denotes the mutual information and the parameters of the summary network $\vartheta$ are considered fixed from the lens of $F_\psi$. The derivation can be found in the **Appendix**. Despite its desirable properties, we expect the conditional hybrid method to be the most challenging to optimize, as it necessitates the training of an additional neural density estimator. Moreover, as with the joint hybrid method, practical trade-offs may result from tying to optimize both criteria simultaneously.

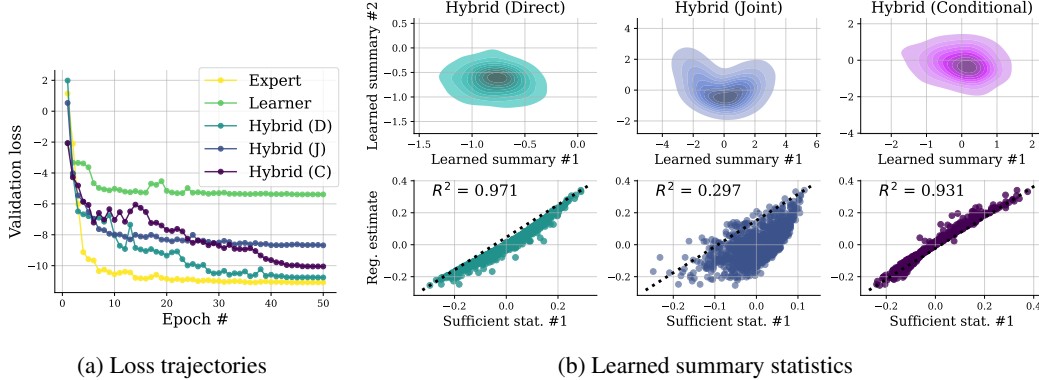

(a) Loss trajectories  (b) Learned summary statistics

Figure 2: **Gaussian Toy example with sufficient statistics**. Results from the low simulation budget setting ($B = 1000$) are shown. Panel a) depicts the trajectory of the average log posterior term (Eq. 7) for all methods. The direct Hybrid (D) converges to the baseline sufficient Expert, while the transformer-based Learner struggles to separate the independent parameter dimensions. The **upper row** of panel b) depicts the marginal distribution of the learned summaries for each *Hybrid* method. The **bottom row** of panel b) plots one of the ground-truth sufficient expert statistics vs. its projection obtained from linear regression on the learned summaries. The joint Hybrid fairs poorly in terms of linear separability of its joint summary space, but is on par with the other Hybrids in terms of estimation, and still considerably outperforms the Learner (detailed results in the **Appendix**).

## 4 RELATED WORK

**Hand-crafted statistics** Rejection-based approximate Bayesian computation (ABC) methods (Sisson et al., 2018; Joyce & Marjoram, 2008), but also more advanced Bayesian optimization approaches (Gutmann & Corander, 2016; Järvenpää et al., 2019), have traditionally relied on low-dimensional hand-crafted summary statistics. Direct regression or classification approaches to ABC allow for entering a much larger number of hand-crafted summary statistics (Raynal et al., 2019; Palestro et al., 2018; Blum et al., 2013; Blum & François, 2010) but still require a preliminary manual selection of the summary vector. Thus, although the latter may be (pre-)processed by a machine learning algorithm, they are not explicitly synthesized with learnable representations of the raw data.

**Learned summary statistics** End-to-end approaches for automatically determining summary statistics have been extensively explored in the context of ABC applications (Fearnhead & Prangle, 2012). Moreover, many neural approaches to SBI have explicitly focused on learning approximately sufficient statistics in tandem with neural approximators (Wrede et al., 2022; Radev et al., 2020a;b; Chen et al., 2021; Wiqvist et al., 2019; Chan et al., 2018), for integration with downstream ABC samplers (Jiang et al., 2017), or in a fully unsupervised manner (Edwards & Storkey, 2016; Albert et al., 2022). None of these methods have explored an expert-in-the-loop approach to augment the learnable summaries with available expert information in parallel to learning the statistics outright.

**Hybrid and expert-in-the-loop approaches** In the context of ABC, Bharti et al. (2022) have employed active learning to improve expert statistics and mitigate the impact of model misspecification. Differently, Tiihonen et al. (2022) have proposed to directly involve human experts for sample quality assurance in an otherwise automated Bayesian optimization of materials. Although these methods have not been employed in amortized inference contexts with summary networks or learnable embeddings, there is certainly a lot of potential for synergy.

## 5 EXPERIMENTS

The following experiments investigate the utility and properties of our hybrid methods across different settings and model configurations. To ensure fair comparisons, we keep the number of summary statistics equal across all experiments. For instance, if a hybrid method uses $D$ expert and $D'$ learned

Table 1: Average, worst, and best posterior contraction and explained variance across $1\,000$ test simulations from each joint model in Ghaderi-Kangavari et al. (2023), each summary type, and two training simulation budgets. Aggregated information from $150\,000$ approximate posteriors.

| | $\mathcal{M}_{1a}$ | $\mathcal{M}_{1b}$ | $\mathcal{M}_{1c}$ | $\mathcal{M}_2$ | $\mathcal{M}_3$ | $\mathcal{M}_{4a}$ | $\mathcal{M}_{4b}$ | $\mathcal{M}_5$ | $\mathcal{M}_6$ | $\mathcal{M}_7$ | $\mathcal{M}_8$ | $\mathcal{M}_9$ | $\mathcal{M}_{10}$ | $\mathcal{M}_{11}$ | $\mathcal{M}_{12}$ |
|---|---|---|---|---|---|---|---|---|---|---|---|---|---|---|---|
| **1 000 Simulations** | | | | | | | | | | | | | | | |
| *Mean posterior contraction (uncertainty reduction, ↑)* | | | | | | | | | | | | | | | |
| Hybrid (D) | **0.82** | **0.84** | 0.75 | **0.78** | 0.62 | **0.62** | **0.70** | **0.75** | **0.77** | **0.95** | **0.80** | **0.85** | 0.75 | **0.98** | **0.87** |
| | [0.47, 0.99] | [0.61, 0.99] | [0.45, 0.99] | [0.45, 0.93] | [0.28, 0.99] | [0.4, 0.93] | [0.34, 0.99] | [0.34, 0.99] | [0.36, 0.99] | [0.88, 0.99] | [0.38, 0.99] | [0.4, 0.99] | [0.36, 0.99] | [0.98, 0.99] | [0.72, 0.99] |
| Hybrid (J) | 0.79 | 0.82 | **0.77** | 0.73 | **0.68** | 0.58 | **0.70** | 0.70 | 0.73 | 0.82 | 0.76 | 0.79 | 0.75 | 0.97 | 0.86 |
| | [0.49, 0.98] | [0.44, 0.98] | [0.5, 0.98] | [0.42, 0.86] | [0.42, 0.98] | [0.44, 0.86] | [0.24, 0.98] | [0.32, 0.98] | [0.31, 0.98] | [0.49, 0.99] | [0.35, 0.98] | [0.49, 0.99] | [0.48, 0.97] | [0.96, 0.99] | [0.72, 0.99] |
| Hybrid (C) | 0.71 | 0.70 | 0.63 | 0.62 | 0.56 | 0.49 | 0.61 | 0.60 | 0.61 | 0.92 | 0.78 | 0.77 | 0.68 | 0.95 | 0.82 |
| | [0.24, 0.98] | [0.27, 0.98] | [0.14, 0.97] | [0.29, 0.76] | [0.17, 0.98] | [0.27, 0.83] | [0.17, 0.98] | [0.22, 0.98] | [0.13, 0.98] | [0.83, 0.98] | [0.4, 0.98] | [0.14, 0.98] | [0.22, 0.98] | [0.9, 0.99] | [0.65, 0.99] |
| Learner | 0.48 | 0.46 | 0.47 | 0.53 | 0.41 | 0.37 | 0.42 | 0.51 | 0.42 | 0.50 | 0.55 | 0.54 | 0.49 | 0.80 | 0.62 |
| | [0.04, 0.99] | [0.03, 0.99] | [0.06, 0.96] | [0.22, 0.78] | [0.05, 0.99] | [0.0, 0.85] | [-0.0, 0.99] | [0.08, 0.99] | [0.03, 0.99] | [0.13, 0.99] | [0.27, 0.93] | [-0.01, 0.98] | [0.06, 0.7] | [0.39, 0.99] | [0.31, 0.99] |
| Expert | 0.80 | 0.79 | 0.73 | 0.69 | 0.60 | 0.54 | 0.69 | 0.67 | 0.66 | 0.78 | 0.74 | 0.84 | 0.72 | 0.96 | 0.80 |
| | [0.39, 0.99] | [0.36, 0.99] | [0.39, 0.99] | [0.36, 0.86] | [0.24, 0.99] | [0.36, 0.8] | [0.33, 0.99] | [0.26, 0.98] | [0.26, 0.98] | [0.4, 0.99] | [0.26, 0.99] | [0.39, 0.99] | [0.34, 0.98] | [0.9, 0.99] | [0.63, 0.99] |
| *Mean explained variance ($R^2$ score, point recovery, ↑)* | | | | | | | | | | | | | | | |
| Hybrid (D) | **0.79** | **0.83** | **0.71** | **0.73** | **0.58** | 0.49 | **0.65** | **0.74** | **0.73** | **0.93** | **0.76** | **0.81** | **0.70** | **0.97** | **0.83** |
| | [0.36, 0.99] | [0.62, 0.99] | [0.45, 0.99] | [0.38, 0.89] | [0.22, 0.98] | [0.21, 0.89] | [0.25, 0.99] | [0.32, 0.99] | [0.3, 0.99] | [0.85, 0.99] | [0.33, 0.99] | [0.39, 0.99] | [0.32, 0.99] | [0.94, 0.99] | [0.61, 0.99] |
| Hybrid (J) | 0.71 | 0.77 | 0.69 | 0.63 | 0.56 | 0.40 | 0.61 | 0.63 | 0.64 | 0.78 | 0.71 | **0.81** | 0.69 | 0.96 | 0.79 |
| | [0.26, 0.98] | [0.29, 0.98] | [0.4, 0.98] | [0.29, 0.78] | [0.14, 0.98] | [0.15, 0.77] | [0.07, 0.98] | [0.22, 0.98] | [0.2, 0.99] | [0.43, 0.99] | [0.28, 0.98] | [0.41, 0.99] | [0.32, 0.98] | [0.91, 0.99] | [0.58, 0.99] |
| Hybrid (C) | 0.71 | 0.70 | 0.66 | 0.59 | 0.55 | 0.40 | 0.60 | 0.62 | 0.61 | 0.92 | **0.76** | 0.80 | **0.70** | 0.96 | 0.82 |
| | [0.32, 0.99] | [0.27, 0.99] | [0.34, 0.98] | [0.29, 0.74] | [0.13, 0.98] | [0.18, 0.82] | [0.16, 0.99] | [0.26, 0.98] | [0.21, 0.99] | [0.84, 0.99] | [0.34, 0.99] | [0.33, 0.99] | [0.36, 0.98] | [0.91, 0.99] | [0.59, 0.99] |
| Learner | 0.43 | 0.42 | 0.40 | 0.47 | 0.34 | 0.29 | 0.39 | 0.48 | 0.38 | 0.48 | 0.49 | 0.54 | 0.44 | 0.79 | 0.61 |
| | [-0.0, 0.98] | [-0.02, 0.98] | [-0.0, 0.98] | [0.06, 0.71] | [-0.02, 0.99] | [-0.02, 0.81] | [-0.01, 0.99] | [0.0, 0.99] | [-0.0, 0.99] | [0.09, 0.98] | [0.26, 0.9] | [-0.02, 0.99] | [-0.0, 0.64] | [0.42, 0.99] | [0.27, 0.99] |
| Expert | 0.76 | 0.76 | 0.69 | 0.63 | 0.57 | 0.42 | 0.64 | 0.66 | 0.62 | 0.78 | 0.73 | 0.80 | 0.68 | 0.95 | 0.77 |
| | [0.25, 0.99] | [0.31, 0.99] | [0.34, 0.99] | [0.26, 0.83] | [0.15, 0.98] | [0.24, 0.74] | [0.25, 0.99] | [0.26, 0.99] | [0.25, 0.99] | [0.42, 0.99] | [0.3, 0.99] | [0.38, 0.99] | [0.29, 0.99] | [0.87, 0.99] | [0.56, 0.99] |
| **10 000 Simulations** | | | | | | | | | | | | | | | |
| *Mean posterior contraction (uncertainty reduction, ↑)* | | | | | | | | | | | | | | | |
| Hybrid (D) | **0.98** | **0.98** | **0.97** | **0.95** | **0.91** | **0.82** | **0.88** | **0.90** | **0.90** | **0.97** | **0.93** | **0.99** | **0.88** | **0.99** | **0.97** |
| | [0.96, 0.99] | [0.96, 0.99] | [0.94, 1.0] | [0.91, 0.97] | [0.81, 0.96] | [0.61, 0.96] | [0.5, 0.99] | [0.52, 0.99] | [0.48, 0.99] | [0.94, 1.0] | [0.82, 1.0] | [0.98, 1.0] | [0.64, 0.99] | [0.98, 1.0] | [0.93, 1.0] |
| Hybrid (J) | **0.98** | **0.98** | **0.97** | 0.93 | 0.89 | 0.80 | 0.85 | 0.89 | 0.90 | **0.97** | 0.92 | **0.99** | 0.88 | **0.99** | **0.97** |
| | [0.96, 0.99] | [0.96, 0.99] | [0.92, 0.99] | [0.86, 0.97] | [0.78, 0.96] | [0.56, 0.97] | [0.41, 0.99] | [0.5, 0.99] | [0.43, 0.99] | [0.95, 1.0] | [0.77, 1.0] | [0.98, 0.99] | [0.66, 0.99] | [0.98, 0.99] | [0.93, 1.0] |
| Hybrid (C) | 0.96 | 0.94 | 0.88 | 0.84 | 0.78 | 0.65 | 0.83 | 0.82 | 0.86 | 0.94 | 0.90 | 0.98 | 0.85 | **0.99** | 0.95 |
| | [0.91, 0.99] | [0.9, 0.99] | [0.65, 0.99] | [0.47, 0.96] | [0.51, 0.99] | [0.33, 0.92] | [0.35, 0.99] | [0.35, 0.99] | [0.39, 0.99] | [0.86, 1.0] | [0.74, 1.0] | [0.96, 0.99] | [0.6, 0.99] | [0.98, 0.99] | [0.89, 0.99] |
| Learner | **0.98** | **0.98** | 0.85 | 0.91 | 0.86 | 0.74 | 0.86 | 0.87 | 0.86 | **0.97** | **0.93** | **0.99** | 0.87 | **0.99** | 0.96 |
| | [0.95, 0.99] | [0.95, 0.99] | [0.17, 0.99] | [0.83, 0.96] | [0.53, 0.99] | [0.48, 0.95] | [0.37, 0.99] | [0.32, 0.99] | [0.25, 0.99] | [0.94, 1.0] | [0.81, 1.0] | [0.98, 1.0] | [0.63, 0.98] | [0.98, 1.0] | [0.89, 0.99] |
| Expert | 0.84 | 0.85 | 0.87 | 0.75 | 0.70 | 0.66 | 0.76 | 0.74 | 0.76 | 0.83 | 0.81 | 0.96 | 0.82 | 0.98 | 0.94 |
| | [0.43, 0.99] | [0.48, 0.99] | [0.69, 0.99] | [0.4, 0.93] | [0.28, 0.99] | [0.4, 0.86] | [0.35, 0.99] | [0.34, 0.99] | [0.32, 0.99] | [0.48, 1.0] | [0.38, 1.0] | [0.88, 1.0] | [0.59, 0.99] | [0.96, 0.99] | [0.85, 0.99] |
| *Mean explained variance ($R^2$ score, point recovery, ↑)* | | | | | | | | | | | | | | | |
| Hybrid (D) | **0.96** | **0.96** | **0.96** | **0.90** | **0.82** | **0.69** | **0.84** | **0.87** | **0.87** | **0.96** | **0.89** | **0.98** | **0.85** | **0.98** | **0.94** |
| | [0.93, 0.99] | [0.94, 0.99] | [0.92, 0.99] | [0.79, 0.96] | [0.7, 0.99] | [0.48, 0.94] | [0.42, 0.99] | [0.5, 0.99] | [0.43, 0.99] | [0.91, 1.0] | [0.64, 1.0] | [0.96, 0.99] | [0.57, 0.99] | [0.96, 0.99] | [0.86, 0.99] |
| Hybrid (J) | **0.96** | **0.96** | 0.95 | 0.87 | 0.80 | 0.67 | 0.80 | 0.86 | 0.86 | **0.96** | 0.88 | **0.98** | **0.85** | **0.98** | **0.94** |
| | [0.92, 0.99] | [0.93, 0.99] | [0.9, 0.99] | [0.75, 0.95] | [0.67, 0.99] | [0.42, 0.94] | [0.32, 0.99] | [0.46, 0.99] | [0.42, 0.99] | [0.9, 1.0] | [0.64, 1.0] | [0.96, 0.99] | [0.57, 0.99] | [0.96, 0.99] | [0.85, 0.99] |
| Hybrid (C) | 0.95 | 0.93 | 0.88 | 0.80 | 0.74 | 0.56 | 0.80 | 0.82 | 0.84 | 0.94 | 0.87 | **0.98** | 0.84 | **0.98** | 0.93 |
| | [0.91, 0.99] | [0.87, 0.99] | [0.66, 0.99] | [0.46, 0.95] | [0.51, 0.99] | [0.29, 0.91] | [0.29, 0.99] | [0.42, 0.99] | [0.38, 0.99] | [0.87, 0.99] | [0.62, 0.99] | [0.95, 0.99] | [0.56, 0.99] | [0.95, 0.99] | [0.82, 0.99] |
| Learner | **0.96** | **0.96** | 0.84 | 0.85 | 0.79 | 0.64 | 0.82 | 0.84 | 0.83 | **0.96** | **0.89** | **0.98** | 0.84 | **0.98** | 0.93 |
| | [0.93, 0.99] | [0.93, 0.99] | [0.09, 0.99] | [0.68, 0.95] | [0.53, 0.99] | [0.31, 0.93] | [0.32, 0.99] | [0.34, 0.99] | [0.29, 0.99] | [0.9, 1.0] | [0.66, 1.0] | [0.95, 0.99] | [0.57, 0.99] | [0.96, 0.99] | [0.81, 0.99] |
| Expert | 0.81 | 0.81 | 0.83 | 0.69 | 0.65 | 0.54 | 0.70 | 0.72 | 0.70 | 0.81 | 0.80 | 0.91 | 0.79 | 0.97 | 0.89 |
| | [0.35, 0.99] | [0.4, 0.99] | [0.6, 0.99] | [0.38, 0.89] | [0.22, 0.99] | [0.32, 0.82] | [0.3, 0.99] | [0.34, 0.99] | [0.3, 0.99] | [0.46, 1.0] | [0.4, 1.0] | [0.76, 0.99] | [0.57, 0.99] | [0.93, 0.99] | [0.74, 0.99] |

summary statistics, the corresponding end-to-end learner will use $D + D'$ summaries. Full training, model, evaluation, and hyperparameter details are available in the **Appendix**.[2]

## 5.1 PROOF OF CONCEPT: GAUSSIAN TOY MODEL

**Setup** We set the stage with a simple 4D version of the conjugate Gaussian model from Eq. 2 with spherical Gaussian prior and likelihood. We simulate 1000 IID data sets of size $N = 50$ and train the following amortized configurations: Expert, Learner, and our three Hybrid methods (abbreviated as D – direct, J – joint, C – conditional). In the Expert setup, the inference network is conditioned on the four sufficient statistics (i.e., empirical means), whereas in the Learner setup, a set transformer is tasked to learn them implicitly from data matrices $\boldsymbol{X} \in \mathbb{R}^{N \times 4}$. In all Hybrid setups, the inference network is fed two out of four empirical means and the dedicated summary networks need to learn the remaining two from the full data. We train all networks for 50 epochs and inspect their convergence, learned summary characteristics, and estimation performance.

**Results** Figure 2 depicts the main results (see **Appendix** for more detailed parameter estimation results and failure mode discussion). Clearly, the Expert setup (using the true sufficient statistics) is the baseline to which the simplest direct Hybrid (D) converges most closely, followed by the conditional and the joint Hybrid (cf. Figure 2a). In contrast, the Learner struggles in the context of the severely limited budget $B$ and small data size $N$, and converges sub-optimally. Furthermore, we

---

[2]Code for all methods and experiments will be released upon acceptance.

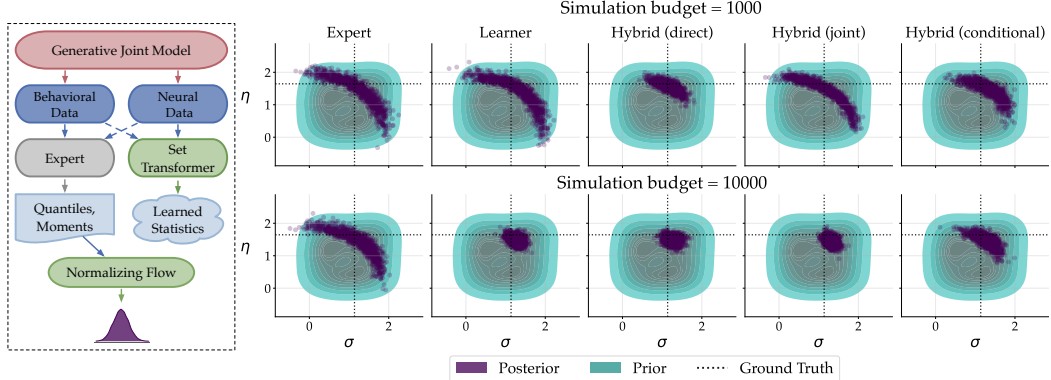

Figure 3: **Left panel**. A graphical illustration of the hybrid setup for estimating neurocognitive joint models. **Right panel**. Example posteriors of two parameters representing a mapping from EEG signals to cognitive parameters, $\eta$ and $\sigma$, obtained from model $\mathcal{M}_7$. See Table 1 and **Appendix** for more detailed results.

obtain 1000 summary representations from all three Hybrid methods and visualize their bivariate distributions in the top panel of Figure 2b. For the direct and conditional Hybrid, these distributions are largely spherical, with the joint Hybrid exhibiting signs of under-regularization with $\lambda = 1.0$. Finally, we regress the unseen sufficient statistics on the learned summaries as an indicator of how much information about the sufficient summaries is linearly encoded by the learned summaries (cf. Figure 2a, **bottom row**). The direct and conditional Hybrids seem to approximate the missing sufficient statistics well, with the direct hybrid achieving the highest explained variance ($R^2$ score). The joint Hybrid fails to learn a linearly separable summary space but achieves on-par estimation (see **Appendix** for full results and the same experiment performed with a larger simulation budget).

## 5.2 COMPREHENSIVE STUDY: JOINT MODELS OF NEURAL AND BEHAVIORAL DATA

**Setup** Next, we perform a systematic evaluation of our methodology within a domain of significant scientific interest – joint models of neural and behavioral data (Turner et al., 2019; Nunez et al., 2022). Simultaneously predicting neural (e.g., electroencephalography, EEG) and behavioral (e.g., response times) data often involves models for which the likelihood is not available in closed-form. Indeed, Ghaderi-Kangavari et al. (2023) formulated 15 mechanistic models aiming to account for single-trial decision-making and EEG waveforms. For each of these models, we train 5 inference networks based on different summary functions (Expert; Learner; Hybrid (D); Hybrid (J); Hybrid (C)) and two simulation budgets (1 000 or 10 000 Simulations), resulting in 10 inference networks per model (see Figure 3, left panel, for a conceptual illustration of a hybrid approach). We then evaluate the performance of each configuration on 1 000 validation simulations per model using two key metrics (Schad et al., 2021): i) the posterior contraction, computed as one minus the ratio of posterior to prior variance, and ii) the mean explained variance, quantified through the $R^2$ score between posterior medians and true data-generating parameters. Such an experiment *necessitates* amortized inference, because it is based on a total of $1\,000 \times 150$ joint posteriors.

**Results** Table 1 summarizes the performance of the 5 different approaches, averaged across all $B = 1\,000$ validation data sets for all 15 models and both training data budgets separately. The best performance is highlighted in bold. In the low training data regime (1 000 simulations), all 3 hybrid approaches outperform the Learner and the Expert approach in both metrics, regardless of the model. Additionally, the Learner exhibits considerably lower performance than the Expert. In the medium data regime (10 000 simulations), the Learner's performance is comparable to that of the Hybrid's for many models. However, for some models, particularly the more complex ones (e.g., $\mathcal{M}_{4a}$), the Hybrid approaches still outperform the Learner in terms of both metrics. Further, when we look at the minima of both metrics the hybrid approximators clearly outperform the other two methods by a wide margin. In contrast to the low data regime, the Expert approach is vastly inferior to the Learner and Hybrid approaches in the medium data regime.

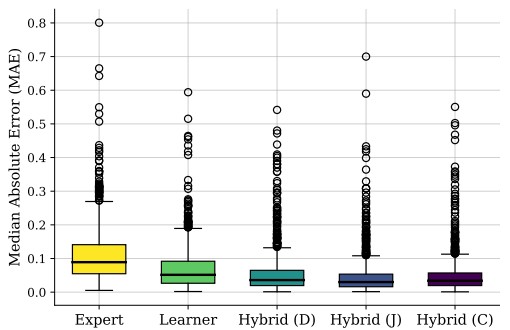 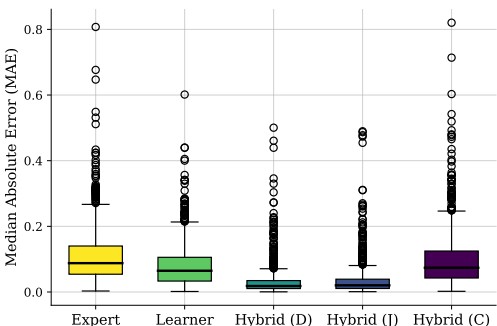

(a) Transformer summary network  (b) Recurrent-convolutional summary network

Figure 4: **Lotka-Volterra model with artificial expert**. Estimation error of all methods on the same 1000 test simulations. The estimation error is measured as the median absolute deviation (MAE) between posterior medians and ground-truth values aggregated across all four model parameters.

To provide a concrete illustration of these results, Figure 3 depicts $1\,000$ posterior draws for two exemplar parameters, $\eta$ and $\sigma$, which govern the mapping from EEG signals to cognitive parameters in $\mathcal{M}_7$. Already in the low data regime, the Hybrid (D) approach demonstrates a pronounced concentration of posterior mass around the true data-generating parameter values (indicated by dashed lines). When more training simulations become available, the Learner catches up with the Hybrids and achieves good sharpness (see **Appendix** for results from other models). Conversely, the Expert's sharpness hardly improves in the medium data regime.

### 5.3 NEURAL EXPERT: LOTKA-VOLTERRA DYNAMIC MODEL

**Setup** The purpose of the last experiment is to apply our hybrid methods to i) a dynamic model generating *non-exchangeable data* ii) using different summary networks (transformer vs. recurrent) and iii) an a pre-trained network acting as an *artificial expert*. We focus on a stochastic Lotka-Volterra (LV) model, which has been extensively studied in the context of SBI (e.g., Lueckmann et al., 2021) The artificial expert is a pre-trained heteroskedastic neural network, which extends the approach of (Jiang et al., 2017) by estimating the posterior variances alongside the posterior means (Kendall & Gal, 2017). The artificial Expert is pre-trained on $B = 1\,000$ simulations, so it yields noisy and imperfect summaries. The same configurations as in the previous experiment are trained with two simulations budgets ($B \in \{1000, 5000\}$).

**Results** Figure 4 summarizes the main results from the medium simulation budget setting ($B = 5000$) in terms of estimation performance on 1000 unseen test simulations from the model. For both types of summary network (but especially the recurrent-convolutional), the direct and joint Hybrid methods achieve the lowest median absolute error (MAE) aggregated across the four model parameters. Further results and calibration diagnostics are available in the **Appendix**.

## 6 CONCLUSION AND FUTURE DIRECTIONS

We introduced three hybrid methods of varying complexity to synthesize expert and learnable summary statistics in amortized Bayesian inference. Our comprehensive evaluation suggests that the simplest hybrid method (directly including the expert representation without explicitly enforcing structure) can reap the most performance gains in low to medium simulation budgets. Still, having explicitly structured (Schmitt et al., 2021) or disentangled (Sorrenson et al., 2020) representations of expert and end-to-end learned summaries may be advantageous in terms of interpretability or robustness, and future research should investigate these issues. One may also incorporate active learning (Bharti et al., 2022) to further tune the expert summaries and reduce the potential impact of model misspecification. Finally, exploring the potential benefits of using hybrid summary statistics for sequential neural estimation methods (e.g., Durkan et al., 2018) seems to be a natural next step.

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

# APPENDIX

## A FREQUENTLY ASKED QUESTIONS (FAQ)

**Q: How can I reproduce the results?**

Code to reproduce all results is available along with the submission and under the following link [removed until publication].

**Q: How can I apply hybrid methods to estimate Bayesian models?**

Amortized simulation-based inference with complex data structures is made possible by different existing software built on top of popular deep learning frameworks (e.g., Radev et al., 2023; Tejero-Cantero et al., 2020).

**Q: Why do I need the expert summaries when I can learn the summaries end-to-end?**

End-to-end learning works best with efficient simulators which can generate synthetic data on demand. Still, it is possible that a summary network is not maximally expressive or unable to train optimally for a variety of reasons. Our experiments suggest that domain knowledge is especially helpful for settings with limited simulation budgets, since it helps the summary network to focus on encoding *incremental information* missing from the expert summaries.

**Q: What is the difference between maximally informative and sufficient summary statistics?**

Sufficient summary statistics in Bayesian analysis satisfy the following strict equality

$$p(\boldsymbol{\theta} \mid \boldsymbol{X}) = p(\boldsymbol{\theta} \mid S(\boldsymbol{X})).$$

In other words, you can replace the data $\boldsymbol{X}$ with a reduced version $S(\boldsymbol{X})$ and still obtain a consistent and correct posterior. Differently, maximally informative statistics are merely *locally optimal* with respect to some distance or divergence between distributions. For instance, using the Kullback-Leibler divergence, a maximally informative summary statistics $S^*$ satisfies:

$$S^* = \arg\min_{S} D_{\mathrm{KL}}\left[p(\boldsymbol{\theta} \mid \boldsymbol{X}) \,\|\, p(\boldsymbol{\theta} \mid S(\boldsymbol{X}))\right] \tag{13}$$

It is evident, that $S^*$ may not be a sufficient statistic in non-convex optimization problems.

**Q: Why do you need a summary network?**

We need these neural data compressors because most real world data comes in various sizes and shapes which vary from application to application or even from inference query to inference query. Thus, we need an interface between the simulator (i.e., Bayesian model) and the inference network which renders the latter independent of the concrete data shape.

**Q: Why do you need three different hybrid methods?**

The hybrid methods explored in this work are only three of the many possible ways to combine expert and learnable representations for improving downstream inference tasks and enforcing specific probabilistic structure. In this work, we specifically focus on methods having somewhat different characteristics and somewhat understood theoretical properties.

**Q: Can I use a different type of generative network for amortized Bayesian inference?**

Definitely, normalizing flows (Papamakarios et al., 2021) are just one attractive family of neural density approximators. In principle, any neural density estimator or sampler, such as compositional score-based models (Geffner et al., 2022) or flow-matching networks (Chen & Lipman, 2023), can be used for the inference network responsible for estimating the intractable posterior. Different architectures do tend to exhibit different empirical behavior though.

# B  METHODS DETAILS

## B.1  PSEUDOCODE: DIRECT HYBRID METHOD

**Algorithm** 1 summarizes the essential steps of the direct hybrid method. This method simply concatenates the learnable and expert summary vectors before passing them to a conditional neural density estimator.

---

**Algorithm 1** Amortized Bayesian inference with a direct hybrid approach.

---

**Require:** Parameter prior, $p(\boldsymbol{\theta})$, randomized simulator $G(\boldsymbol{\theta})$, summary network $L_\vartheta$, inference network $F_\phi$, expert $H$, batch size $B$, simulation budget $N$.
1: Initialize empty data set $\mathcal{D} = \{\}$.
2: **for** $n = 1, \ldots, N$ **do**
3:     Sample from prior: $\boldsymbol{\theta}_n \sim p(\boldsymbol{\theta})$
4:     Perform a simulator run: $\boldsymbol{X}_n = G(\boldsymbol{\theta}_n)$
5:     Add simulations to training data: $\mathcal{D} := \mathcal{D} \cup \{(\boldsymbol{\theta}_n, \boldsymbol{X}_n)\}$
6: **end for**
7: **while** not converged **do**
8:     Sample batch from training data: $\{(\boldsymbol{\theta}_b, \boldsymbol{X}_b)\}_{b=1}^B \sim \mathcal{D}$
9:     Compute learnable summaries: $\{L_\vartheta(\boldsymbol{X}_b)\}_{b=1}^B$
10:     Compute expert summaries: $\{H(\boldsymbol{X}_b)\}_{b=1}^B$
11:     Combine learnable and expert summaries: $\{\boldsymbol{s}_{\vartheta,b} := [L_\vartheta(\boldsymbol{X}_b), H(\boldsymbol{X}_b)]\}_{b=1}^B$
12:     Compute maximum likelihood loss (Eq. 7) over batch $\{(\boldsymbol{\theta}_b, \boldsymbol{s}_{\vartheta,b})\}_{b=1}^B$
13:     Update network parameters $(\phi, \vartheta)$ via backpropagation
14: **end while**

---

## B.2  PSEUDOCODE: JOINT HYBRID METHOD

**Algorithm** 2 summarizes the essential steps of the joint hybrid method. This method first embeds the expert statistics into a new vector space before concatenating them with the learnable vectors. The fully end-to-end learnable joint vector is then passed as the conditioning vector for a conditional neural density estimator.

---

**Algorithm 2** Amortized Bayesian inference with a joint hybrid approach.

---

**Require:** Parameter prior, $p(\boldsymbol{\theta})$, randomized simulator $G(\boldsymbol{\theta})$, summary network $L_\vartheta$, inference network $F_\phi$, embedding network $T_\varphi$, expert $H$, batch size $B$, simulation budget $N$.
1: Initialize empty data set $\mathcal{D} = \{\}$.
2: **for** $n = 1, \ldots, N$ **do**
3:     Sample from prior: $\boldsymbol{\theta}_n \sim p(\boldsymbol{\theta})$
4:     Perform a simulator run: $\boldsymbol{X}_n = G(\boldsymbol{\theta}_n)$
5:     Add simulations to training data: $\mathcal{D} := \mathcal{D} \cup \{(\boldsymbol{\theta}_n, \boldsymbol{X}_n)\}$
6: **end for**
7: **while** not converged **do**
8:     Sample batch from training data: $\{(\boldsymbol{\theta}_b, \boldsymbol{X}_b)\}_{b=1}^B \sim \mathcal{D}$
9:     Compute learnable summaries: $\{L_\vartheta(\boldsymbol{X}_b)\}_{b=1}^B$
10:     Compute expert summaries: $\{H(\boldsymbol{X}_b)\}_{b=1}^B$
11:     Embed expert summaries: $\{\boldsymbol{t}_{\varphi,b} = T_\varphi(H(\boldsymbol{X}_b))\}_{b=1}^B$
12:     Combine learnable and embedded expert summaries: $\{\boldsymbol{s}_{\vartheta,b}^{(\varphi)} := [L_\vartheta(\boldsymbol{X}_b), \boldsymbol{t}_{\varphi,b}]\}_{b=1}^B$
13:     Compute augmented maximum likelihood loss (Eq. 8) over batch $\{(\boldsymbol{\theta}_b, \boldsymbol{s}_{\vartheta,b}^{(\varphi)})\}_{b=1}^B$
14:     Update network parameters $(\phi, \vartheta, \varphi)$ via backpropagation
15: **end while**

---

## B.3 PSEUDOCODE: CONDITIONAL HYBRID METHOD

**Algorithm** 3 summarizes the essential steps of the conditional hybrid method. This method embeds the learnable summary statistics into a latent space conditional on the expert information before combining them with the expert statistics. The combined vector is then passed as the conditioning vector for a conditional neural density estimator. The method requires a `stop_gradient` operator, since the summary and inference network parameters need to be updated by treating the conditional summary model's parameter fixed, and *vice versa*.

---

**Algorithm 3** Amortized Bayesian inference with a conditional hybrid approach.

---

**Require:** Parameter prior, $p(\boldsymbol{\theta})$, randomized simulator $G(\boldsymbol{\theta})$, summary network $L_\vartheta$, inference network $F_\phi$, generative summary network $F_\psi$, expert $H$, batch size $B$, simulation budget $N$.

1: Initialize empty data set $\mathcal{D} = \{\}$.
2: **for** $n = 1, \ldots, N$ **do**
3:     Sample from prior: $\boldsymbol{\theta}_n \sim p(\boldsymbol{\theta})$
4:     Perform a simulator run: $\boldsymbol{X}_n = G(\boldsymbol{\theta}_n)$
5:     Add simulations to training data: $\mathcal{D} := \mathcal{D} \cup \{(\boldsymbol{\theta}_n, \boldsymbol{X}_n)\}$
6: **end for**
7: **while** not converged **do**
8:     Sample batch from training data: $\{(\boldsymbol{\theta}_b, \boldsymbol{X}_b)\}_{b=1}^B \sim \mathcal{D}$
9:     Compute learnable summaries: $\{L_\vartheta(\boldsymbol{X}_b)\}_{b=1}^B$
10:     Compute expert summaries: $\{H(\boldsymbol{X}_b)\}_{b=1}^B$
11:     Compute latent learnable summaries: $\{\boldsymbol{z}_{\vartheta,b}^{(\psi)} := F_\psi(L_\vartheta(\boldsymbol{X}_b); H(\boldsymbol{X}))\}_{b=1}^B$
12:     Combine latent learnable and expert summaries: $\{\boldsymbol{s}_{\vartheta,b}^{(\psi)} := \text{concat}(\boldsymbol{z}_{\vartheta,b}^{(\psi)}, H(\boldsymbol{X}_b))\}_{b=1}^B$
13:     Stop gradient w.r.t. $\psi$
14:     Compute maximum likelihood loss (Eq. 9) over batch $\{(\boldsymbol{\theta}_b, \boldsymbol{s}_{\vartheta,b}^{(\psi)})\}_{b=1}^B$
15:     Update network parameters $(\phi, \vartheta)$ via backpropagation
16:     Stop gradient w.r.t. $\vartheta$
17:     Compute maximum likelihood loss (Eq. 10) over batch $\{(L_\vartheta(\boldsymbol{X}_b), H(\boldsymbol{X}))\}_{b=1}^B$
18:     Update network parameters $\psi$ via backpropagation
19: **end while**

---

## B.4 DERIVATION OF CONDITIONAL HYBRID OPTIMIZATION

Recall, that in the conditional method, we are optimizing the following two objectives simultaneously:

$$\phi^*, \vartheta^* = \arg\min_{\phi,\vartheta} \mathbb{E}_{p(\boldsymbol{\theta},\boldsymbol{X})} \left[ -\log q_\phi(\boldsymbol{\theta} \mid \boldsymbol{z}_\vartheta^{(\psi)}, H(\boldsymbol{X})) \right] \tag{14}$$

$$\psi^* = \arg\min_\psi \mathbb{E}_{p(\boldsymbol{X})} \left[ -\log q_\psi(L_\vartheta(\boldsymbol{X}) \mid H(\boldsymbol{X})) \right]. \tag{15}$$

Setting $\boldsymbol{z}_\vartheta^{(\psi)} = F_\psi(L_\vartheta(\boldsymbol{X}); H(\boldsymbol{X}))$ and keeping $F_\psi$ fixed, the first objective (Eq. 14) follows simply from

$$\phi^*, \vartheta^* = \arg\min_{\phi,\vartheta} \mathbb{E}_{p(\boldsymbol{X})} \left[ D_{\text{KL}} \left[ p(\boldsymbol{\theta} \mid \boldsymbol{X}) \,||\, q_\phi(\boldsymbol{\theta} \mid \boldsymbol{z}_\vartheta^{(\psi)}, H(\boldsymbol{X})) \right] \right] \tag{16}$$

$$= \arg\min_{\phi,\vartheta} \mathbb{E}_{p(\boldsymbol{X})} \left[ \mathbb{E}_{p(\boldsymbol{\theta} \mid \boldsymbol{X})} \left[ \log p(\boldsymbol{\theta} \mid \boldsymbol{X}) - \log q_\phi(\boldsymbol{\theta} \mid \boldsymbol{z}_\vartheta^{(\psi)}, H(\boldsymbol{X})) \right] \right] \tag{17}$$

$$= \arg\min_{\phi,\vartheta} \mathbb{E}_{p(\boldsymbol{X})} \left[ \mathbb{E}_{p(\boldsymbol{\theta} \mid \boldsymbol{X})} \left[ -\log q_\phi(\boldsymbol{\theta} \mid \boldsymbol{z}_\vartheta^{(\psi)}, H(\boldsymbol{X})) \right] \right] \tag{18}$$

$$= \arg\min_{\phi,\vartheta} \mathbb{E}_{p(\boldsymbol{\theta},\boldsymbol{X})} \left[ -\log q_\phi(\boldsymbol{\theta} \mid \boldsymbol{z}_\vartheta^{(\psi)}, H(\boldsymbol{X})) \right], \tag{19}$$

corresponding to conditional maximum likelihood training over the simulator outputs $\{\boldsymbol{\theta}, \boldsymbol{X}\}$. Assuming idealized optimal convergence, Ardizzone et al. (2021) have shown that summary networks

learn maximally informative statistics with respect to the target variable. Since the same reasoning extends to our *doubly conditional* maximum likelihood training, we have, under idealized optimal convergence:

$$I(\boldsymbol{\theta}, \boldsymbol{z}_{\vartheta*}^{(\psi)} \mid H(\boldsymbol{X})) = \max_{\vartheta} I(\boldsymbol{\theta}, \boldsymbol{z}_{\vartheta}^{(\psi)} \mid H(\boldsymbol{X})) \tag{20}$$

Keeping $L_{\vartheta}(\boldsymbol{X})$ fixed, the second objective (Eq. 15) follows from a similar perspective

$$\psi^* = \arg\min_{\psi} \mathbb{E}_{p(\boldsymbol{X})} \left[ D_{\mathrm{KL}}(p(L_{\vartheta}(\boldsymbol{X}) \mid H(\boldsymbol{X})) \,||\, q_{\psi}(L_{\vartheta}(\boldsymbol{X}) \mid H(\boldsymbol{X}))) \right] \tag{21}$$

$$= \arg\min_{\psi} \mathbb{E}_{p(\boldsymbol{X})} \left[ \log p(L_{\vartheta}(\boldsymbol{X}) \mid H(\boldsymbol{X})) - \log q_{\psi}(L_{\vartheta}(\boldsymbol{X}) \mid H(\boldsymbol{X})) \right] \tag{22}$$

$$= \arg\min_{\psi} \mathbb{E}_{p(\boldsymbol{X})} \left[ -\log q_{\psi}(L_{\vartheta}(\boldsymbol{X}) \mid H(\boldsymbol{X})) \right] \tag{23}$$

Here, we can utilize the knowledge that the Kullback-Leibler (KL) divergence is invariant to diffeomorphisms, which means that, assuming perfect convergence under Eq. 21, we have

$$D_{\mathrm{KL}} \left[ q_{\psi^*} \left( \boldsymbol{z}_{\vartheta}^{(\psi^*)} \mid H(\boldsymbol{X}) \right) \,||\, p \left( \boldsymbol{z}_{\vartheta}^{(\psi^*)} \right) \right] = 0, \tag{24}$$

where $p \left( \boldsymbol{z}_{\vartheta}^{(\psi^*)} \right)$ corresponds to the base distribution prescribed through maximum likelihood training. In other words, under perfect convergence, the base distribution becomes independent of $E(\boldsymbol{X})$ and so we have:

$$\boldsymbol{z}_{\vartheta*}^{(\psi^*)} \perp\!\!\!\perp H(\boldsymbol{X}) \tag{25}$$

In other words, during training, the first objective (Eq. 14) encourages the summary network $L_{\vartheta}$ to maximize the information context of the learned summaries with respect to $\boldsymbol{\theta}$. At the same time, the second objective (Eq. 14) encourages the generative network $F_{\psi}$ to make the latent summary vectors maximally independent of the expert's outputs. As a consequence of using a base distribution with independent dimensions, each latent summary dimension will also attempt to encode different information. In practice, since $L_{\vartheta}(\boldsymbol{X})$ in Eq. 15 will change at each iteration, training may become unstable, especially towards the early epochs where $L_{\vartheta}(\boldsymbol{X})$ is still hardly informative for $\boldsymbol{\theta}$ and thus changes substantially from iteration to iteration. Thus, we find it essential to use a Student-T density as a base distribution for robust maximum likelihood training (Alexanderson & Henter, 2020).

## C  IMPLEMENTATION DETAILS AND ADDITIONAL RESULTS

All experiments are implemented in `TensorFlow` (Abadi et al., 2016). Throughout, we use an Adam optimizer (Kingma & Ba, 2014) with an initial learning rate between 0.0005 and 0.001, default hyperparameters, weight decay of 0.0001, and a cosine learning rate decay schedule. All networks are trained on a single desktop computer equipped with an NVIDIA® A2000 graphics card with 12GB of GPU memory.

### C.1  EXPERIMENT 1: PROOF OF CONCEPT

#### C.1.1  MODEL

We use the following toy conjugate Gaussian model as a simulator:

$$\boldsymbol{\theta} \sim \mathcal{N}(\mathbf{0}, \mathbb{I}) \tag{26}$$
$$\boldsymbol{x}_n \sim \mathcal{N}(\boldsymbol{\theta}, \mathbb{I}) \quad \text{for} \quad n = 1, \dots, N, \tag{27}$$

where $N = 50$ and $\boldsymbol{\theta}, \boldsymbol{x} \in \mathbb{R}^4$. The model has an analytic posterior and trivial summary statistics (i.e., the empirical mean vector $\bar{\boldsymbol{x}}$). However, it may prove slippery to learn directly in an amortized manner and limited budget settings due to the fact that not only the $\boldsymbol{x}_n$ are independent among each other, but also the dimensions of each individual $\boldsymbol{x}_n$ are independent. Thus, there is very little structure that the end-to-end learners can extract apart from learning to compute averages.

#### C.1.2  SUMMARY NETWORKS

The set transformer summary network (Lee et al., 2019) features a stack of 2 multi-head attention encoders with 4 attention heads (key, query, and value dimensions of 32).

### C.1.3 INFERENCE NETWORKS

The inference network is a conditional normalizing flows consisting of 6 affine coupling layers (Ardizzone et al., 2019), each having two hidden layer with 128 units, ReLU non-linearities, fixed permutations between the coupling layers, and a Gaussian base distribution.

### C.1.4 SIMULATION-BASED TRAINING

We train all configurations (Expert, Learner, Hybrid (D), Hybrid (J), and Hybrid (C)) for 50 epochs in both the $B = 1000$ and $B = 10\,000$ settings, using a batch size of 32 simulations.

### C.1.5 ADDITIONAL RESULTS

The main text reports results obtained in the low budget scenario ($B = 1000$). Figure 5 depicts the results obtained from the additional experiment with a simulation budget of $B = 10\,000$.

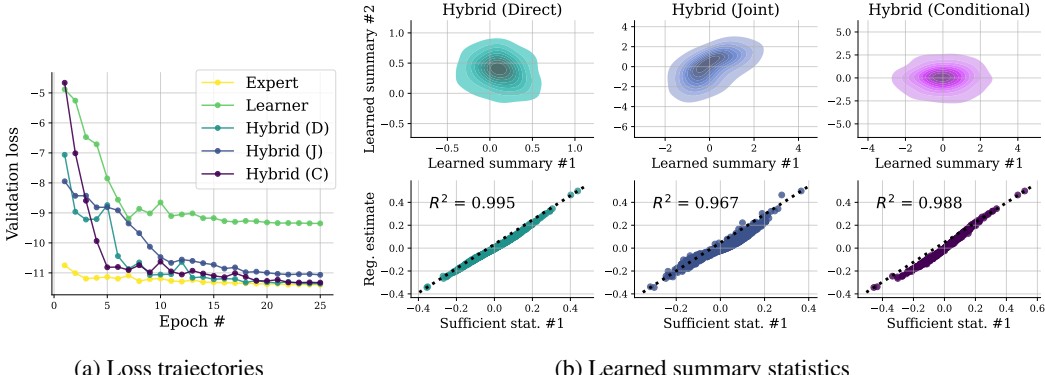

(a) Loss trajectories          (b) Learned summary statistics

Figure 5: **Gaussian toy example with sufficient statistics**. Results from the medium simulation budget ($B = 10\,000$) are shown. Panel a) depicts the trajectory of the average log posterior term (Eq. 7) for all methods. All Hybrid methods quickly converge to the baseline sufficient Expert, except the transformer-based Learner which still fails to separate the independent parameter dimensions. The **upper row** of panel b) depicts the learned summaries for each *Hybrid* method. The **bottom row** of panel b) plots one of the ground-truth sufficient expert statistics vs. its projection obtained from linear regression on the learned summaries.

In addition, Figure 6 and Figure 7 illustrate the estimation performance of each configuration without any hyperparameter tuning.

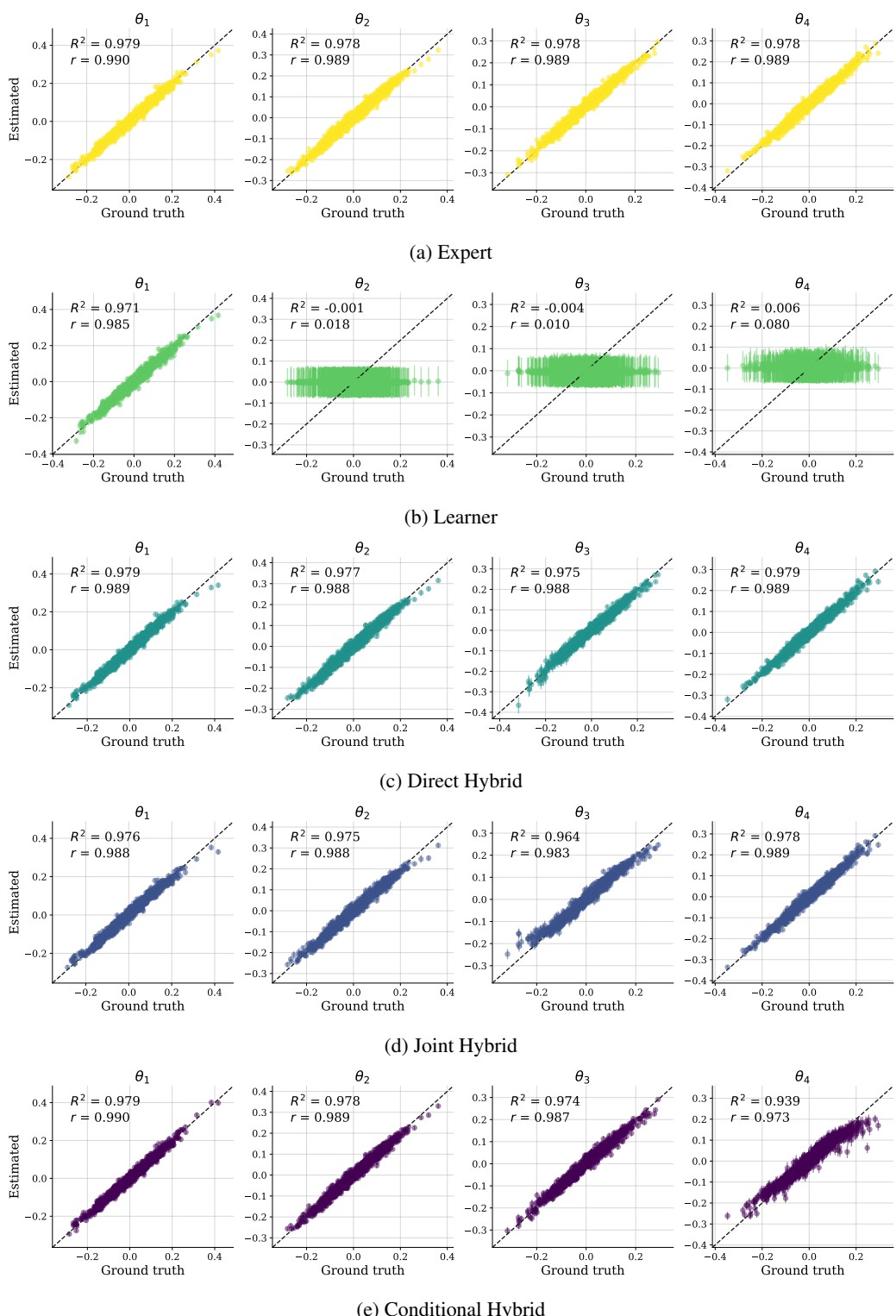

Figure 6: **Gaussian toy model**. Detailed parameter estimation results obtained using a set transformer summary network with a low simulation budget of $B = 1000$. Failure mode of the Learner is clearly illustrated.

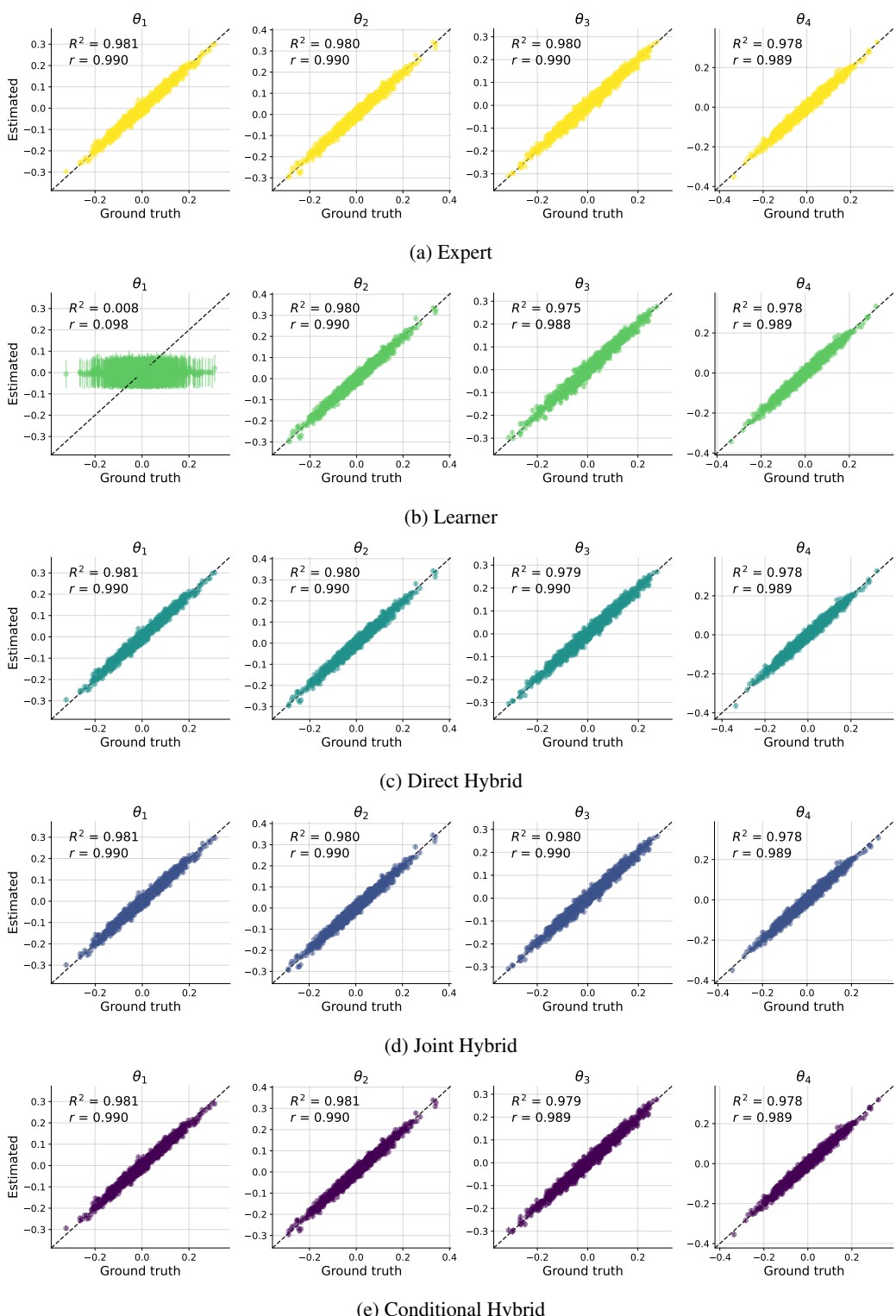

Figure 7: **Gaussian toy model**. Detailed parameter estimation results obtained using a set transformer summary network with a medium simulation budget of $B = 10\,000$. The Learner-only approach improves but still exhibits a failure mode. Some hyperparameter tuning resolves the issue, but the scenario still illustrate a setting where domain expert information can mitigate potential underperformance of the summary network.

## C.2 Experiment 2: Comprehensive study

### C.2.1 Models

The 15 different neurocognitive joint models used in this study aim to predict human behavioral and neural data obtained from simulated two-alternative decision-making tasks (Ghaderi-Kangavari et al., 2023). The multi-source data encompasses single-trial response times and choices as behavioral data, and EEG potentials as neural data.

For the expert summary statistic, we computed quantiles $(0.025, 0.2625, 0.5, 0.7375, 0.975)$ as well as the mean and standard deviation of the empirical response time and EEG data distributions, alongside the mean accuracy, resulting in 19 hand-crafted summary statistics. For the purpose of offline training, we generate data sets consisting of $N = 200$ simulated trials of the generative process implemented by the respective joint model.

### C.2.2 Summary Networks

The summary network is a set transformer (Lee et al., 2019) implementing multi-head attention with 4 attention heads (key, query, and value dimensions of 32) and no layer normalization, as the latter was found to hurt performance in the case of set transformers.

### C.2.3 Inference Networks

The inference network is a conditional normalizing flows consisting of 6 affine coupling layers (Ardizzone et al., 2019), each having two hidden layer with 128 units, ReLU non-linearities, fixed permutations between the coupling layers, and a Gaussian base distribution. Each hidden layers is followed by a dropout layer with a dropout probability of $0.05$.

### C.2.4 Simulation-based training

We train all configurations (Expert, Learner, Hybrid (D), Hybrid (J), and Hybrid (C)) for 100 epochs using a batch size of 32 simulations and 500 validation simulations for monitoring overfitting.

### C.2.5 Model-specific estimation performance

Complementing the evaluation presented in Table 1 of the main text, we conducted a detailed assessment of the performance of the five summary approaches within each of the two settings with different simulation budgets. This involved analyzing uncertainty reduction (posterior contraction) and explained variance (point recovery) for every parameter of each model separately (refer to the subsequent tables). By delving into this more fine-grained analysis, we reaffirm the conclusions drawn in the main text. For nearly every parameter across all models, the Hybrid approaches, particularly the simplest *direct* variant, outperforms both the Learner and the Expert in terms of our two key evaluation metrics.

Table 2: Posterior contraction and explained variance across $1\,000$ test simulations from each parameter of model $\mathcal{M}_{1a}$ separate for each summary type, and two training simulation budgets.

| | $v$ | $a$ | $\beta$ | $\mu_\tau$ | $\tau_m$ | $\sigma$ | $\sigma_{var}$ | Mean | $v$ | $a$ | $\beta$ | $\mu_\tau$ | $\tau_m$ | $\sigma$ | $\sigma_{var}$ | Mean |
|---|---|---|---|---|---|---|---|---|---|---|---|---|---|---|---|---|
| | Posterior contraction (uncertainty reduction) | | | | | | | | Explained variance ($R^2$ score, point recovery) | | | | | | | |
| **$1\,000$ Simulations** | | | | | | | | | | | | | | | | |
| Hybrid (D) | 0.83 | 0.82 | 0.45 | 0.99 | 0.91 | 0.82 | 0.90 | **0.82** | 0.75 | 0.81 | 0.41 | 0.99 | 0.89 | 0.80 | 0.88 | **0.79** |
| Hybrid (J) | 0.85 | 0.83 | 0.48 | 0.98 | 0.90 | 0.66 | 0.81 | 0.79 | 0.74 | 0.78 | 0.29 | 0.99 | 0.88 | 0.55 | 0.74 | 0.71 |
| Hybrid (C) | 0.75 | 0.73 | 0.23 | 0.98 | 0.87 | 0.63 | 0.74 | 0.71 | 0.76 | 0.74 | 0.35 | 0.99 | 0.86 | 0.56 | 0.71 | 0.71 |
| Learner | 0.75 | 0.40 | 0.40 | 0.99 | 0.72 | 0.05 | 0.03 | 0.48 | 0.69 | 0.32 | 0.34 | 0.99 | 0.71 | 0.00 | 0.00 | 0.43 |
| Expert | 0.80 | 0.79 | 0.39 | 0.99 | 0.89 | 0.82 | 0.90 | 0.80 | 0.74 | 0.77 | 0.30 | 0.99 | 0.87 | 0.78 | 0.87 | 0.76 |
| **$10\,000$ Simulations** | | | | | | | | | | | | | | | | |
| Hybrid (D) | 0.97 | 0.97 | 0.95 | 0.99 | 0.98 | 0.98 | 0.98 | **0.98** | 0.95 | 0.94 | 0.93 | 0.99 | 0.98 | 0.97 | 0.97 | **0.96** |
| Hybrid (J) | 0.98 | 0.97 | 0.96 | 0.99 | 0.99 | 0.97 | 0.97 | **0.98** | 0.95 | 0.94 | 0.92 | 0.99 | 0.98 | 0.96 | 0.97 | **0.96** |
| Hybrid (C) | 0.96 | 0.95 | 0.91 | 0.99 | 0.97 | 0.95 | 0.96 | 0.96 | 0.94 | 0.93 | 0.91 | 0.99 | 0.97 | 0.94 | 0.96 | 0.95 |
| Learner | 0.97 | 0.97 | 0.95 | 0.99 | 0.99 | 0.98 | 0.98 | **0.98** | 0.95 | 0.94 | 0.92 | 0.99 | 0.98 | 0.97 | 0.97 | **0.96** |
| Expert | 0.85 | 0.88 | 0.42 | 0.99 | 0.92 | 0.89 | 0.93 | 0.84 | 0.76 | 0.86 | 0.40 | 0.99 | 0.91 | 0.85 | 0.92 | 0.81 |

Table 3: Posterior contraction and explained variance across $1\,000$ test simulations from each parameter of model $\mathcal{M}_{1b}$ separate for each summary type, and two training simulation budgets.

| | $v$ | $a$ | $\beta$ | $\tau_e$ | $\tau_m$ | $\sigma$ | $\sigma_{var}$ | Mean | $v$ | $a$ | $\beta$ | $\tau_e$ | $\tau_m$ | $\sigma$ | $\sigma_{var}$ | Mean |
|---|---|---|---|---|---|---|---|---|---|---|---|---|---|---|---|---|
| | Posterior contraction (uncertainty reduction) | | | | | | | | Explained variance ($R^2$ score, point recovery) | | | | | | | |
| **$1\,000$ Simulations** | | | | | | | | | | | | | | | | |
| Hybrid (D) | 0.87 | 0.85 | 0.60 | 0.99 | 0.90 | 0.81 | 0.88 | **0.84** | 0.81 | 0.81 | 0.65 | 0.99 | 0.90 | 0.79 | 0.88 | **0.83** |
| Hybrid (J) | 0.87 | 0.87 | 0.43 | 0.98 | 0.91 | 0.81 | 0.89 | 0.82 | 0.75 | 0.80 | 0.32 | 0.98 | 0.88 | 0.78 | 0.86 | 0.77 |
| Hybrid (C) | 0.76 | 0.69 | 0.27 | 0.98 | 0.84 | 0.67 | 0.70 | 0.70 | 0.71 | 0.72 | 0.27 | 0.98 | 0.84 | 0.63 | 0.74 | 0.70 |
| Learner | 0.73 | 0.39 | 0.35 | 0.99 | 0.71 | 0.04 | 0.02 | 0.46 | 0.67 | 0.35 | 0.31 | 0.98 | 0.66 | 0.00 | 0.03 | 0.42 |
| Expert | 0.79 | 0.80 | 0.36 | 0.99 | 0.89 | 0.83 | 0.88 | 0.79 | 0.73 | 0.75 | 0.32 | 0.99 | 0.87 | 0.80 | 0.88 | 0.76 |
| **$10\,000$ Simulations** | | | | | | | | | | | | | | | | |
| Hybrid (D) | 0.98 | 0.97 | 0.95 | 0.99 | 0.98 | 0.98 | 0.98 | **0.98** | 0.96 | 0.94 | 0.94 | 0.99 | 0.98 | 0.97 | 0.97 | **0.96** |
| Hybrid (J) | 0.98 | 0.97 | 0.96 | 0.99 | 0.99 | 0.97 | 0.97 | **0.98** | 0.96 | 0.94 | 0.93 | 0.99 | 0.98 | 0.96 | 0.97 | **0.96** |
| Hybrid (C) | 0.96 | 0.94 | 0.92 | 0.99 | 0.96 | 0.89 | 0.94 | 0.94 | 0.95 | 0.91 | 0.92 | 0.98 | 0.96 | 0.87 | 0.93 | 0.93 |
| Learner | 0.97 | 0.97 | 0.95 | 0.99 | 0.99 | 0.98 | 0.98 | **0.98** | 0.96 | 0.94 | 0.93 | 0.99 | 0.98 | 0.97 | 0.97 | **0.96** |
| Expert | 0.86 | 0.89 | 0.48 | 0.99 | 0.93 | 0.89 | 0.93 | 0.85 | 0.75 | 0.85 | 0.41 | 0.99 | 0.90 | 0.85 | 0.91 | 0.81 |

Table 4: Posterior contraction and explained variance across 1 000 test simulations from each parameter of model $\mathcal{M}_{1c}$ separate for each summary type, and two training simulation budgets.

| | $v$ | $a$ | $\beta$ | $\tau_e$ | $\tau_m$ | $\sigma$ | $\sigma_{var}$ | Mean | $v$ | $a$ | $\beta$ | $\tau_e$ | $\tau_m$ | $\sigma$ | $\sigma_{var}$ | Mean |
|---|---|---|---|---|---|---|---|---|---|---|---|---|---|---|---|---|
| | Posterior contraction (uncertainty reduction) | | | | | | | | Explained variance ($R^2$ score, point recovery) | | | | | | | |
| **1 000 Simulations** | | | | | | | | | | | | | | | | |
| Hybrid (D) | 0.85 | 0.93 | 0.45 | 0.99 | 0.91 | 0.53 | 0.57 | 0.75 | 0.79 | 0.91 | 0.44 | 0.99 | 0.91 | 0.47 | 0.46 | **0.71** |
| Hybrid (J) | 0.86 | 0.93 | 0.57 | 0.98 | 0.94 | 0.49 | 0.59 | **0.77** | 0.77 | 0.90 | 0.48 | 0.98 | 0.91 | 0.36 | 0.44 | 0.69 |
| Hybrid (C) | 0.71 | 0.80 | 0.13 | 0.97 | 0.85 | 0.47 | 0.48 | 0.63 | 0.76 | 0.85 | 0.34 | 0.98 | 0.88 | 0.41 | 0.41 | 0.66 |
| Learner | 0.70 | 0.51 | 0.33 | 0.96 | 0.58 | 0.10 | 0.08 | 0.47 | 0.67 | 0.41 | 0.27 | 0.98 | 0.51 | 0.00 | -0.01 | 0.40 |
| Expert | 0.79 | 0.90 | 0.40 | 0.99 | 0.92 | 0.53 | 0.56 | 0.73 | 0.75 | 0.86 | 0.34 | 0.99 | 0.91 | 0.49 | 0.49 | 0.69 |
| **10 000 Simulations** | | | | | | | | | | | | | | | | |
| Hybrid (D) | 0.98 | 0.98 | 0.98 | 0.99 | 0.99 | 0.94 | 0.94 | **0.97** | 0.98 | 0.96 | 0.97 | 0.99 | 0.99 | 0.92 | 0.91 | **0.96** |
| Hybrid (J) | 0.98 | 0.98 | 0.98 | 0.99 | 0.99 | 0.93 | 0.93 | **0.97** | 0.97 | 0.96 | 0.96 | 0.99 | 0.99 | 0.90 | 0.89 | 0.95 |
| Hybrid (C) | 0.96 | 0.96 | 0.94 | 0.99 | 0.97 | 0.65 | 0.71 | 0.88 | 0.96 | 0.94 | 0.94 | 0.99 | 0.97 | 0.65 | 0.68 | 0.88 |
| Learner | 0.98 | 0.98 | 0.97 | 0.99 | 0.98 | 0.18 | 0.89 | 0.85 | 0.97 | 0.95 | 0.96 | 0.98 | 0.98 | 0.13 | 0.87 | 0.84 |
| Expert | 0.90 | 0.96 | 0.68 | 0.99 | 0.96 | 0.80 | 0.83 | 0.87 | 0.80 | 0.93 | 0.59 | 0.99 | 0.96 | 0.79 | 0.77 | 0.83 |

Table 5: Posterior contraction and explained variance across 1 000 test simulations from each parameter of model $\mathcal{M}_2$ separate for each summary type, and two training simulation budgets.

| | $v$ | $a$ | $\beta$ | $\tau_e$ | $\tau_m$ | $\sigma$ | $\sigma_{var}$ | $\gamma$ | Mean | $v$ | $a$ | $\beta$ | $\tau_e$ | $\tau_m$ | $\sigma$ | $\sigma_{var}$ | $\gamma$ | Mean |
|---|---|---|---|---|---|---|---|---|---|---|---|---|---|---|---|---|---|---|
| | Posterior contraction (uncertainty reduction) | | | | | | | | | Explained variance ($R^2$ score, point recovery) | | | | | | | | |
| **1 000 Simulations** | | | | | | | | | | | | | | | | | | |
| Hybrid (D) | 0.93 | 0.85 | 0.74 | 0.70 | 0.83 | 0.51 | 0.84 | 0.84 | **0.78** | 0.90 | 0.84 | 0.79 | 0.56 | 0.76 | 0.39 | 0.83 | **0.79** | 0.73 |
| Hybrid (J) | 0.85 | 0.79 | 0.53 | 0.69 | 0.83 | 0.48 | 0.79 | 0.84 | 0.73 | 0.74 | 0.76 | 0.44 | 0.56 | 0.76 | 0.32 | 0.69 | 0.77 | 0.63 |
| Hybrid (C) | 0.74 | 0.67 | 0.28 | 0.62 | 0.74 | 0.43 | 0.69 | 0.75 | 0.62 | 0.73 | 0.71 | 0.27 | 0.50 | 0.71 | 0.34 | 0.68 | 0.75 | 0.59 |
| Learner | 0.73 | 0.39 | 0.37 | 0.59 | 0.62 | 0.22 | 0.52 | 0.78 | 0.53 | 0.67 | 0.33 | 0.31 | 0.53 | 0.59 | 0.05 | 0.53 | 0.74 | 0.47 |
| Expert | 0.83 | 0.75 | 0.37 | 0.68 | 0.77 | 0.47 | 0.86 | 0.82 | 0.69 | 0.71 | 0.73 | 0.27 | 0.57 | 0.74 | 0.41 | 0.84 | 0.79 | 0.63 |
| **10 000 Simulations** | | | | | | | | | | | | | | | | | | |
| Hybrid (D) | 0.97 | 0.97 | 0.95 | 0.90 | 0.96 | 0.95 | 0.97 | 0.95 | **0.95** | 0.95 | 0.94 | 0.93 | 0.76 | 0.89 | 0.90 | 0.96 | 0.89 | **0.90** |
| Hybrid (J) | 0.97 | 0.96 | 0.95 | 0.84 | 0.94 | 0.90 | 0.96 | 0.93 | 0.93 | 0.95 | 0.92 | 0.92 | 0.72 | 0.88 | 0.78 | 0.94 | 0.86 | 0.87 |
| Hybrid (C) | 0.96 | 0.93 | 0.92 | 0.72 | 0.84 | 0.53 | 0.93 | 0.86 | 0.84 | 0.95 | 0.91 | 0.91 | 0.63 | 0.81 | 0.48 | 0.92 | 0.82 | 0.80 |
| Learner | 0.96 | 0.95 | 0.94 | 0.80 | 0.94 | 0.86 | 0.96 | 0.91 | 0.91 | 0.95 | 0.91 | 0.91 | 0.66 | 0.87 | 0.72 | 0.93 | 0.84 | 0.85 |
| Expert | 0.85 | 0.87 | 0.42 | 0.71 | 0.82 | 0.53 | 0.93 | 0.87 | 0.75 | 0.73 | 0.83 | 0.35 | 0.63 | 0.78 | 0.47 | 0.90 | 0.82 | 0.69 |

Table 6: Posterior contraction and explained variance across 1 000 test simulations from each parameter of model $\mathcal{M}_3$ separate for each summary type, and two training simulation budgets.

| | $v$ | $a$ | $\beta$ | $\tau_e$ | $\tau_m$ | $\sigma$ | $\sigma_{var}$ | $\theta$ | Mean | $v$ | $a$ | $\beta$ | $\tau_e$ | $\tau_m$ | $\sigma$ | $\sigma_{var}$ | $\theta$ | Mean |
|---|---|---|---|---|---|---|---|---|---|---|---|---|---|---|---|---|---|---|
| | Posterior contraction (uncertainty reduction) | | | | | | | | | Explained variance ($R^2$ score, point recovery) | | | | | | | | |
| **1 000 Simulations** | | | | | | | | | | | | | | | | | | |
| Hybrid (D) | 0.63 | 0.35 | 0.28 | 0.99 | 0.74 | 0.48 | 0.51 | 0.96 | 0.62 | 0.59 | 0.25 | 0.25 | 0.98 | 0.68 | 0.46 | 0.48 | 0.96 | **0.58** |
| Hybrid (J) | 0.70 | 0.47 | 0.43 | 0.98 | 0.76 | 0.54 | 0.62 | 0.97 | **0.68** | 0.56 | 0.29 | 0.16 | 0.98 | 0.61 | 0.45 | 0.46 | 0.96 | 0.56 |
| Hybrid (C) | 0.53 | 0.21 | 0.17 | 0.98 | 0.66 | 0.44 | 0.54 | 0.95 | 0.56 | 0.55 | 0.21 | 0.15 | 0.98 | 0.61 | 0.46 | 0.47 | 0.96 | 0.55 |
| Learner | 0.11 | 0.21 | 0.09 | 0.99 | 0.77 | 0.09 | 0.05 | 0.96 | 0.41 | 0.04 | 0.13 | -0.01 | 0.98 | 0.66 | -0.01 | -0.02 | 0.96 | 0.34 |
| Expert | 0.61 | 0.30 | 0.24 | 0.99 | 0.74 | 0.48 | 0.50 | 0.96 | 0.60 | 0.59 | 0.24 | 0.17 | 0.98 | 0.67 | 0.45 | 0.48 | 0.96 | 0.57 |
| **10 000 Simulations** | | | | | | | | | | | | | | | | | | |
| Hybrid (D) | 0.89 | 0.80 | 0.81 | 0.99 | 0.95 | 0.93 | 0.92 | 0.98 | **0.91** | 0.78 | 0.66 | 0.70 | 0.98 | 0.85 | 0.81 | 0.82 | 0.97 | **0.82** |
| Hybrid (J) | 0.89 | 0.77 | 0.81 | 0.99 | 0.94 | 0.87 | 0.86 | 0.98 | 0.89 | 0.79 | 0.63 | 0.68 | 0.98 | 0.83 | 0.75 | 0.75 | 0.97 | 0.80 |
| Hybrid (C) | 0.76 | 0.52 | 0.57 | 0.99 | 0.81 | 0.82 | 0.81 | 0.97 | 0.78 | 0.69 | 0.53 | 0.50 | 0.98 | 0.75 | 0.74 | 0.74 | 0.97 | 0.74 |
| Learner | 0.78 | 0.76 | 0.55 | 0.99 | 0.92 | 0.94 | 0.93 | 0.98 | 0.86 | 0.71 | 0.64 | 0.54 | 0.99 | 0.83 | 0.83 | 0.83 | 0.97 | 0.79 |
| Expert | 0.70 | 0.51 | 0.28 | 0.99 | 0.81 | 0.64 | 0.67 | 0.97 | 0.70 | 0.60 | 0.48 | 0.22 | 0.98 | 0.72 | 0.59 | 0.61 | 0.97 | 0.65 |

Table 7: Posterior contraction and explained variance across 1 000 test simulations from each parameter of model $\mathcal{M}_{4a}$ separate for each summary type, and two training simulation budgets.

| | $v$ | $a$ | $\beta$ | $\tau_e$ | $\tau_m$ | $\tau$ | $\sigma_e$ | $\sigma_k$ | $\sigma_{var}$ | $k$ | $\theta$ | Mean | $v$ | $a$ | $\beta$ | $\tau_e$ | $\tau_m$ | $\tau$ | $\sigma_e$ | $\sigma_k$ | $\sigma_{var}$ | $k$ | $\theta$ | Mean |
|---|---|---|---|---|---|---|---|---|---|---|---|---|---|---|---|---|---|---|---|---|---|---|---|---|
| | Posterior contraction (uncertainty reduction) | | | | | | | | | | | | Explained variance ($R^2$ score, point recovery) | | | | | | | | | | | |
| **1 000 Simulations** | | | | | | | | | | | | | | | | | | | | | | | | |
| Hybrid (D) | 0.92 | 0.84 | 0.79 | 0.75 | 0.46 | 0.74 | 0.41 | 0.54 | 0.41 | 0.52 | 0.43 | **0.62** | 0.88 | 0.79 | 0.79 | 0.58 | 0.29 | 0.57 | 0.18 | 0.35 | 0.32 | 0.27 | 0.35 | **0.49** |
| Hybrid (J) | 0.84 | 0.82 | 0.43 | 0.71 | 0.42 | 0.77 | 0.42 | 0.49 | 0.46 | 0.52 | 0.50 | 0.58 | 0.70 | 0.74 | 0.34 | 0.52 | 0.28 | 0.59 | 0.12 | 0.24 | 0.28 | 0.31 | 0.34 | 0.40 |
| Hybrid (C) | 0.82 | 0.77 | 0.45 | 0.65 | 0.28 | 0.64 | 0.31 | 0.38 | 0.28 | 0.45 | 0.33 | 0.49 | 0.79 | 0.72 | 0.53 | 0.54 | 0.18 | 0.51 | 0.18 | 0.21 | 0.28 | 0.21 | 0.29 | 0.40 |
| Learner | 0.84 | 0.56 | 0.56 | 0.57 | 0.27 | 0.42 | 0.04 | 0.01 | 0.10 | 0.50 | 0.16 | 0.37 | 0.79 | 0.53 | 0.54 | 0.43 | 0.17 | 0.44 | -0.03 | -0.03 | 0.03 | 0.23 | 0.09 | 0.29 |
| Expert | 0.79 | 0.72 | 0.37 | 0.74 | 0.47 | 0.70 | 0.36 | 0.52 | 0.39 | 0.51 | 0.43 | 0.54 | 0.70 | 0.69 | 0.27 | 0.59 | 0.33 | 0.53 | 0.20 | 0.35 | 0.33 | 0.25 | 0.34 | 0.42 |
| **10 000 Simulations** | | | | | | | | | | | | | | | | | | | | | | | | |
| Hybrid (D) | 0.96 | 0.96 | 0.94 | 0.87 | 0.85 | 0.91 | 0.61 | 0.73 | 0.62 | 0.78 | 0.77 | **0.82** | 0.94 | 0.92 | 0.90 | 0.73 | 0.70 | 0.71 | 0.43 | 0.54 | 0.53 | 0.57 | 0.62 | **0.69** |
| Hybrid (J) | 0.97 | 0.96 | 0.94 | 0.86 | 0.78 | 0.89 | 0.57 | 0.71 | 0.59 | 0.74 | 0.77 | 0.80 | 0.94 | 0.91 | 0.90 | 0.73 | 0.63 | 0.69 | 0.39 | 0.52 | 0.51 | 0.55 | 0.62 | 0.67 |
| Hybrid (C) | 0.91 | 0.87 | 0.81 | 0.78 | 0.49 | 0.83 | 0.32 | 0.60 | 0.39 | 0.61 | 0.56 | 0.65 | 0.90 | 0.85 | 0.85 | 0.66 | 0.49 | 0.60 | 0.22 | 0.43 | 0.36 | 0.28 | 0.48 | 0.56 |
| Learner | 0.95 | 0.92 | 0.92 | 0.80 | 0.75 | 0.89 | 0.54 | 0.59 | 0.46 | 0.68 | 0.61 | 0.74 | 0.93 | 0.82 | 0.88 | 0.69 | 0.62 | 0.71 | 0.42 | 0.48 | 0.40 | 0.53 | 0.55 | 0.64 |
| Expert | 0.84 | 0.85 | 0.48 | 0.82 | 0.58 | 0.83 | 0.38 | 0.69 | 0.45 | 0.68 | 0.63 | 0.66 | 0.73 | 0.81 | 0.41 | 0.68 | 0.52 | 0.62 | 0.28 | 0.49 | 0.41 | 0.43 | 0.51 | 0.54 |

Table 8: Posterior contraction and explained variance across 1 000 test simulations from each parameter of model $\mathcal{M}_{4b}$ separate for each summary type, and two training simulation budgets.

| | $v$ | $a$ | $\beta$ | $\mu_{\tau e}$ | $\tau_m$ | $\sigma_e$ | $\sigma_{var}$ | $\theta$ | Mean | $v$ | $a$ | $\beta$ | $\mu_{\tau e}$ | $\tau_m$ | $\sigma_e$ | $\sigma_{var}$ | $\theta$ | Mean |
|---|---|---|---|---|---|---|---|---|---|---|---|---|---|---|---|---|---|---|
| | Posterior contraction (uncertainty reduction) | | | | | | | | | Explained variance ($R^2$ score, point recovery) | | | | | | | | |
| **1 000 Simulations** | | | | | | | | | | | | | | | | | | |
| Hybrid (D) | 0.82 | 0.87 | 0.41 | 0.99 | 0.92 | 0.58 | 0.69 | 0.34 | **0.70** | 0.74 | 0.82 | 0.39 | 0.99 | 0.90 | 0.58 | 0.60 | 0.22 | **0.65** |
| Hybrid (J) | 0.87 | 0.88 | 0.54 | 0.98 | 0.92 | 0.52 | 0.63 | 0.23 | **0.70** | 0.76 | 0.80 | 0.37 | 0.98 | 0.89 | 0.48 | 0.54 | 0.03 | 0.61 |
| Hybrid (C) | 0.72 | 0.77 | 0.20 | 0.98 | 0.87 | 0.59 | 0.64 | 0.16 | 0.61 | 0.74 | 0.74 | 0.26 | 0.99 | 0.86 | 0.51 | 0.57 | 0.15 | 0.60 |
| Learner | 0.74 | 0.39 | 0.36 | 0.99 | 0.72 | 0.07 | 0.07 | 0.01 | 0.42 | 0.68 | 0.37 | 0.31 | 0.99 | 0.71 | 0.01 | 0.03 | 0.00 | 0.39 |
| Expert | 0.81 | 0.83 | 0.38 | 0.99 | 0.91 | 0.61 | 0.69 | 0.31 | 0.69 | 0.74 | 0.77 | 0.28 | 0.99 | 0.89 | 0.60 | 0.62 | 0.23 | 0.64 |
| **10 000 Simulations** | | | | | | | | | | | | | | | | | | |
| Hybrid (D) | 0.98 | 0.97 | 0.96 | 0.99 | 0.99 | 0.78 | 0.85 | 0.50 | **0.88** | 0.97 | 0.95 | 0.95 | 0.99 | 0.98 | 0.73 | 0.73 | 0.40 | **0.84** |
| Hybrid (J) | 0.97 | 0.97 | 0.95 | 0.99 | 0.98 | 0.70 | 0.81 | 0.40 | 0.85 | 0.96 | 0.93 | 0.93 | 0.99 | 0.97 | 0.66 | 0.69 | 0.28 | 0.80 |
| Hybrid (C) | 0.96 | 0.95 | 0.93 | 0.99 | 0.97 | 0.66 | 0.81 | 0.35 | 0.83 | 0.96 | 0.93 | 0.92 | 0.99 | 0.97 | 0.67 | 0.69 | 0.27 | 0.80 |
| Learner | 0.97 | 0.97 | 0.96 | 0.99 | 0.99 | 0.75 | 0.82 | 0.40 | 0.86 | 0.96 | 0.94 | 0.93 | 0.99 | 0.98 | 0.69 | 0.73 | 0.30 | 0.82 |
| Expert | 0.87 | 0.92 | 0.50 | 0.99 | 0.95 | 0.67 | 0.81 | 0.35 | 0.76 | 0.76 | 0.86 | 0.44 | 0.99 | 0.93 | 0.66 | 0.67 | 0.25 | 0.70 |

Table 9: Posterior contraction and explained variance across 1 000 test simulations from each parameter of model $\mathcal{M}_5$ separate for each summary type, and two training simulation budgets.

| | $v$ | $a$ | $\beta$ | $\tau_e$ | $\tau_m$ | $\sigma$ | $\sigma_{var}$ | $a_{slope}$ | Mean | $v$ | $a$ | $\beta$ | $\tau_e$ | $\tau_m$ | $\sigma$ | $\sigma_{var}$ | $a_{slope}$ | Mean |
|---|---|---|---|---|---|---|---|---|---|---|---|---|---|---|---|---|---|---|
| | Posterior contraction (uncertainty reduction) | | | | | | | | | Explained variance ($R^2$ score, point recovery) | | | | | | | | |
| **1 000 Simulations** | | | | | | | | | | | | | | | | | | |
| Hybrid (D) | 0.52 | 0.80 | 0.89 | 0.99 | 0.85 | 0.79 | 0.84 | 0.32 | **0.75** | 0.47 | 0.77 | 0.90 | 0.99 | 0.85 | 0.76 | 0.82 | 0.34 | **0.74** |
| Hybrid (J) | 0.41 | 0.79 | 0.80 | 0.98 | 0.84 | 0.68 | 0.78 | 0.31 | 0.70 | 0.27 | 0.70 | 0.78 | 0.98 | 0.80 | 0.58 | 0.66 | 0.23 | 0.63 |
| Hybrid (C) | 0.26 | 0.70 | 0.70 | 0.98 | 0.74 | 0.59 | 0.64 | 0.21 | 0.60 | 0.27 | 0.70 | 0.76 | 0.98 | 0.76 | 0.58 | 0.63 | 0.24 | 0.62 |
| Learner | 0.25 | 0.54 | 0.77 | 0.99 | 0.53 | 0.42 | 0.55 | 0.07 | 0.51 | 0.18 | 0.47 | 0.79 | 0.98 | 0.52 | 0.37 | 0.50 | 0.03 | 0.48 |
| Expert | 0.26 | 0.72 | 0.76 | 0.98 | 0.80 | 0.72 | 0.76 | 0.31 | 0.67 | 0.25 | 0.69 | 0.78 | 0.99 | 0.81 | 0.70 | 0.74 | 0.32 | 0.66 |
| **10 000 Simulations** | | | | | | | | | | | | | | | | | | |
| Hybrid (D) | 0.84 | 0.93 | 0.97 | 0.99 | 0.97 | 0.98 | 0.98 | 0.52 | **0.90** | 0.73 | 0.89 | 0.96 | 0.99 | 0.96 | 0.97 | 0.96 | 0.49 | **0.87** |
| Hybrid (J) | 0.83 | 0.93 | 0.97 | 0.99 | 0.97 | 0.97 | 0.97 | 0.49 | 0.89 | 0.71 | 0.89 | 0.95 | 0.99 | 0.95 | 0.95 | 0.96 | 0.45 | 0.86 |
| Hybrid (C) | 0.83 | 0.93 | 0.97 | 0.99 | 0.97 | 0.97 | 0.97 | 0.49 | 0.89 | 0.71 | 0.89 | 0.95 | 0.99 | 0.95 | 0.95 | 0.96 | 0.45 | 0.86 |
| Learner | 0.83 | 0.90 | 0.97 | 0.99 | 0.97 | 0.97 | 0.97 | 0.32 | 0.87 | 0.72 | 0.87 | 0.95 | 0.99 | 0.95 | 0.97 | 0.96 | 0.34 | 0.84 |
| Expert | 0.35 | 0.82 | 0.80 | 0.99 | 0.88 | 0.86 | 0.90 | 0.35 | 0.74 | 0.30 | 0.80 | 0.80 | 0.99 | 0.86 | 0.81 | 0.85 | 0.39 | 0.72 |

Table 10: Posterior contraction and explained variance across $1\,000$ test simulations from each parameter of model $\mathcal{M}_6$ separate for each summary type, and two training simulation budgets.

| | $v$ | $a$ | $\beta$ | $\tau_e$ | $\tau_m$ | $\sigma$ | $\sigma_{var}$ | $\lambda$ | Mean | $v$ | $a$ | $\beta$ | $\tau_e$ | $\tau_m$ | $\sigma$ | $\sigma_{var}$ | $\lambda$ | Mean |
|---|---|---|---|---|---|---|---|---|---|---|---|---|---|---|---|---|---|---|
| | Posterior contraction (uncertainty reduction) | | | | | | | | | Explained variance ($R^2$ score, point recovery) | | | | | | | | |
| **1 000 Simulations** | | | | | | | | | | | | | | | | | | |
| Hybrid (D) | 0.82 | 0.74 | 0.87 | 0.99 | 0.82 | 0.74 | 0.79 | 0.37 | **0.77** | 0.78 | 0.68 | 0.87 | 0.99 | 0.77 | 0.69 | 0.76 | 0.28 | **0.73** |
| Hybrid (J) | 0.68 | 0.80 | 0.71 | 0.98 | 0.83 | 0.73 | 0.81 | 0.33 | 0.73 | 0.52 | 0.69 | 0.67 | 0.98 | 0.77 | 0.64 | 0.74 | 0.15 | 0.64 |
| Hybrid (C) | 0.54 | 0.69 | 0.55 | 0.98 | 0.72 | 0.64 | 0.65 | 0.14 | 0.61 | 0.47 | 0.66 | 0.59 | 0.99 | 0.70 | 0.60 | 0.72 | 0.16 | 0.61 |
| Learner | 0.43 | 0.56 | 0.64 | 0.99 | 0.58 | 0.03 | 0.06 | 0.10 | 0.42 | 0.42 | 0.51 | 0.65 | 0.99 | 0.47 | 0.01 | 0.01 | 0.01 | 0.38 |
| Expert | 0.49 | 0.63 | 0.60 | 0.98 | 0.78 | 0.74 | 0.78 | 0.30 | 0.66 | 0.48 | 0.59 | 0.59 | 0.99 | 0.74 | 0.67 | 0.74 | 0.17 | 0.62 |
| **10 000 Simulations** | | | | | | | | | | | | | | | | | | |
| Hybrid (D) | 0.92 | 0.96 | 0.96 | 0.99 | 0.97 | 0.98 | 0.97 | 0.49 | **0.90** | 0.88 | 0.88 | 0.93 | 0.99 | 0.95 | 0.96 | 0.97 | 0.39 | **0.87** |
| Hybrid (J) | 0.92 | 0.94 | 0.96 | 0.99 | 0.97 | 0.97 | 0.97 | 0.46 | **0.90** | 0.87 | 0.86 | 0.92 | 0.99 | 0.94 | 0.95 | 0.96 | 0.34 | 0.86 |
| Hybrid (C) | 0.87 | 0.91 | 0.92 | 0.99 | 0.93 | 0.94 | 0.95 | 0.38 | 0.86 | 0.85 | 0.85 | 0.92 | 0.99 | 0.93 | 0.94 | 0.95 | 0.33 | 0.84 |
| Learner | 0.91 | 0.87 | 0.95 | 0.99 | 0.95 | 0.98 | 0.97 | 0.26 | 0.86 | 0.87 | 0.80 | 0.93 | 0.99 | 0.93 | 0.96 | 0.96 | 0.23 | 0.83 |
| Expert | 0.68 | 0.84 | 0.67 | 0.99 | 0.84 | 0.84 | 0.89 | 0.34 | 0.76 | 0.57 | 0.75 | 0.67 | 0.99 | 0.81 | 0.79 | 0.85 | 0.19 | 0.70 |

Table 11: Posterior contraction and explained variance across $1\,000$ test simulations from each parameter of model $\mathcal{M}_7$ separate for each summary type, and two training simulation budgets.

| | $\mu_v$ | $a$ | $\beta$ | $\tau$ | $\sigma$ | $\eta$ | Mean | $\mu_v$ | $a$ | $\beta$ | $\tau$ | $\sigma$ | $\eta$ | Mean |
|---|---|---|---|---|---|---|---|---|---|---|---|---|---|---|
| | Posterior contraction (uncertainty reduction) | | | | | | | Explained variance ($R^2$ score, point recovery) | | | | | | |
| **1 000 Simulations** | | | | | | | | | | | | | | |
| Hybrid (D) | 0.99 | 0.96 | 0.98 | 0.99 | 0.89 | 0.90 | **0.95** | 0.98 | 0.94 | 0.97 | 0.99 | 0.85 | 0.85 | **0.93** |
| Hybrid (J) | 0.99 | 0.95 | 0.97 | 0.98 | 0.51 | 0.52 | 0.82 | 0.98 | 0.92 | 0.94 | 0.99 | 0.45 | 0.42 | 0.78 |
| Hybrid (C) | 0.98 | 0.94 | 0.94 | 0.98 | 0.83 | 0.86 | 0.92 | 0.98 | 0.94 | 0.95 | 0.99 | 0.85 | 0.84 | 0.92 |
| Learner | 0.98 | 0.20 | 0.14 | 0.80 | 0.36 | 0.50 | 0.50 | 0.97 | 0.19 | 0.09 | 0.80 | 0.42 | 0.43 | 0.48 |
| Expert | 0.98 | 0.93 | 0.93 | 0.99 | 0.41 | 0.47 | 0.78 | 0.98 | 0.89 | 0.93 | 0.99 | 0.46 | 0.42 | 0.78 |
| **10 000 Simulations** | | | | | | | | | | | | | | |
| Hybrid (D) | 0.99 | 0.98 | 0.99 | 1.00 | 0.94 | 0.94 | **0.97** | 0.98 | 0.97 | 0.99 | 1.00 | 0.92 | 0.91 | **0.96** |
| Hybrid (J) | 0.99 | 0.98 | 0.99 | 1.00 | 0.95 | 0.95 | **0.97** | 0.98 | 0.96 | 0.99 | 1.00 | 0.91 | 0.91 | **0.96** |
| Hybrid (C) | 0.98 | 0.96 | 0.96 | 1.00 | 0.86 | 0.87 | 0.94 | 0.98 | 0.95 | 0.97 | 0.99 | 0.87 | 0.88 | 0.94 |
| Learner | 0.99 | 0.98 | 0.99 | 1.00 | 0.94 | 0.94 | **0.97** | 0.98 | 0.96 | 0.99 | 1.00 | 0.91 | 0.91 | **0.96** |
| Expert | 0.98 | 0.96 | 0.97 | 1.00 | 0.50 | 0.54 | 0.83 | 0.98 | 0.94 | 0.97 | 1.00 | 0.47 | 0.48 | 0.81 |

Table 12: Posterior contraction and explained variance across $1\,000$ test simulations from each parameter of model $\mathcal{M}_8$ separate for each summary type, and two training simulation budgets.

| | $v$ | $a$ | $\tau$ | $\sigma$ | $\gamma$ | $\eta$ | Mean | $v$ | $a$ | $\tau$ | $\sigma$ | $\gamma$ | $\eta$ | Mean |
|---|---|---|---|---|---|---|---|---|---|---|---|---|---|---|
| | Posterior contraction (uncertainty reduction) | | | | | | | Explained variance ($R^2$ score, point recovery) | | | | | | |
| **1 000 Simulations** | | | | | | | | | | | | | | |
| Hybrid (D) | 0.89 | 0.97 | 0.99 | 0.33 | 0.97 | 0.61 | **0.80** | 0.86 | 0.96 | 0.99 | 0.31 | 0.95 | 0.48 | **0.76** |
| Hybrid (J) | 0.79 | 0.95 | 0.98 | 0.32 | 0.95 | 0.57 | 0.76 | 0.74 | 0.94 | 0.98 | 0.26 | 0.92 | 0.44 | 0.71 |
| Hybrid (C) | 0.84 | 0.95 | 0.98 | 0.37 | 0.96 | 0.56 | 0.78 | 0.85 | 0.96 | 0.99 | 0.33 | 0.94 | 0.47 | **0.76** |
| Learner | 0.67 | 0.29 | 0.69 | 0.24 | 0.93 | 0.48 | 0.55 | 0.54 | 0.25 | 0.65 | 0.24 | 0.91 | 0.37 | 0.49 |
| Expert | 0.78 | 0.95 | 0.99 | 0.23 | 0.94 | 0.54 | 0.74 | 0.76 | 0.95 | 0.99 | 0.29 | 0.92 | 0.44 | 0.73 |
| **10 000 Simulations** | | | | | | | | | | | | | | |
| Hybrid (D) | 0.95 | 0.99 | 1.00 | 0.79 | 0.99 | 0.89 | **0.93** | 0.93 | 0.98 | 1.00 | 0.65 | 0.97 | 0.78 | **0.89** |
| Hybrid (J) | 0.95 | 0.99 | 1.00 | 0.76 | 0.99 | 0.87 | 0.92 | 0.93 | 0.98 | 1.00 | 0.63 | 0.97 | 0.78 | 0.88 |
| Hybrid (C) | 0.93 | 0.98 | 1.00 | 0.69 | 0.98 | 0.83 | 0.90 | 0.92 | 0.98 | 0.99 | 0.63 | 0.97 | 0.76 | 0.87 |
| Learner | 0.95 | 0.99 | 1.00 | 0.79 | 0.99 | 0.89 | **0.93** | 0.93 | 0.98 | 1.00 | 0.66 | 0.97 | 0.79 | **0.89** |
| Expert | 0.92 | 0.98 | 1.00 | 0.36 | 0.98 | 0.64 | 0.81 | 0.91 | 0.98 | 0.99 | 0.37 | 0.96 | 0.57 | 0.80 |

Table 13: Posterior contraction and explained variance across $1\,000$ test simulations from each parameter of model $\mathcal{M}_9$ separate for each summary type, and two training simulation budgets.

| | $v$ | $a$ | $\beta$ | $t_e$ | $t_m$ | $\sigma_e$ | Mean | $v$ | $a$ | $\beta$ | $t_e$ | $t_m$ | $\sigma_e$ | Mean |
|---|---|---|---|---|---|---|---|---|---|---|---|---|---|---|
| | Posterior contraction (uncertainty reduction) | | | | | | | Explained variance ($R^2$ score, point recovery) | | | | | | |
| **1 000 Simulations** | | | | | | | | | | | | | | |
| Hybrid (D) | 0.82 | 0.93 | 0.39 | 0.99 | 0.97 | 0.99 | **0.85** | 0.71 | 0.89 | 0.32 | 0.99 | 0.96 | 0.99 | **0.81** |
| Hybrid (J) | 0.82 | 0.91 | 0.46 | 0.99 | 0.97 | 0.98 | **0.85** | 0.70 | 0.88 | 0.35 | 0.99 | 0.95 | 0.98 | **0.81** |
| Hybrid (C) | 0.76 | 0.82 | 0.15 | 0.98 | 0.93 | 0.98 | 0.77 | 0.70 | 0.88 | 0.27 | 0.99 | 0.95 | 0.98 | 0.80 |
| Learner | 0.68 | 0.39 | 0.44 | 0.98 | 0.77 | -0.02 | 0.54 | 0.66 | 0.37 | 0.47 | 0.99 | 0.79 | -0.03 | 0.54 |
| Expert | 0.78 | 0.89 | 0.39 | 0.99 | 0.97 | 0.99 | 0.84 | 0.69 | 0.85 | 0.31 | 0.99 | 0.96 | 0.99 | 0.80 |
| **10 000 Simulations** | | | | | | | | | | | | | | |
| Hybrid (D) | 0.99 | 0.98 | 0.99 | 0.99 | 0.99 | 0.99 | **0.99** | 0.98 | 0.96 | 0.98 | 0.99 | 0.99 | 0.99 | **0.98** |
| Hybrid (J) | 0.98 | 0.98 | 0.98 | 0.99 | 0.99 | 0.99 | **0.99** | 0.98 | 0.96 | 0.98 | 0.99 | 0.99 | 0.99 | **0.98** |
| Hybrid (C) | 0.97 | 0.97 | 0.96 | 0.99 | 0.99 | 0.99 | 0.98 | 0.98 | 0.96 | 0.97 | 0.99 | 0.99 | 0.99 | **0.98** |
| Learner | 0.98 | 0.98 | 0.98 | 1.00 | 0.99 | 0.99 | **0.99** | 0.97 | 0.96 | 0.96 | 0.99 | 0.99 | 0.99 | **0.98** |
| Expert | 0.94 | 0.96 | 0.86 | 0.99 | 0.99 | 0.99 | 0.96 | 0.84 | 0.94 | 0.72 | 0.99 | 0.99 | 0.99 | 0.91 |

Table 14: Posterior contraction and explained variance across $1\,000$ test simulations from each parameter of model $\mathcal{M}_{10}$ separate for each summary type, and two training simulation budgets.

| | $v$ | $a$ | $\beta$ | $t_e$ | $t_m$ | $\sigma_e$ | $\gamma$ | Mean | $v$ | $a$ | $\beta$ | $t_e$ | $t_m$ | $\sigma_e$ | $\gamma$ | Mean |
|---|---|---|---|---|---|---|---|---|---|---|---|---|---|---|---|---|
| | Posterior contraction (uncertainty reduction) | | | | | | | | Explained variance ($R^2$ score, point recovery) | | | | | | | |
| **$1\,000$ Simulations** | | | | | | | | | | | | | | | | |
| Hybrid (D) | 0.83 | 0.92 | 0.40 | 0.65 | 0.78 | 0.99 | 0.69 | 0.75 | 0.71 | 0.88 | 0.35 | 0.55 | 0.74 | 0.99 | 0.67 | **0.70** |
| Hybrid (J) | 0.86 | 0.92 | 0.54 | 0.69 | 0.82 | 0.97 | 0.73 | **0.79** | 0.72 | 0.86 | 0.35 | 0.54 | 0.72 | 0.98 | 0.67 | 0.69 |
| Hybrid (C) | 0.76 | 0.84 | 0.21 | 0.58 | 0.77 | 0.97 | 0.62 | 0.68 | 0.74 | 0.85 | 0.37 | 0.54 | 0.73 | 0.98 | 0.66 | **0.70** |
| Learner | 0.72 | 0.32 | 0.39 | 0.61 | 0.66 | 0.07 | 0.66 | 0.49 | 0.68 | 0.24 | 0.39 | 0.53 | 0.60 | 0.00 | 0.66 | 0.44 |
| Expert | 0.75 | 0.86 | 0.35 | 0.66 | 0.78 | 0.98 | 0.66 | 0.72 | 0.69 | 0.81 | 0.31 | 0.56 | 0.75 | 0.99 | 0.67 | 0.68 |
| **$10\,000$ Simulations** | | | | | | | | | | | | | | | | |
| Hybrid (D) | 0.98 | 0.98 | 0.98 | 0.67 | 0.81 | 0.99 | 0.71 | **0.88** | 0.98 | 0.96 | 0.98 | 0.58 | 0.77 | 0.99 | 0.68 | **0.85** |
| Hybrid (J) | 0.98 | 0.98 | 0.98 | 0.68 | 0.82 | 0.99 | 0.72 | **0.88** | 0.97 | 0.95 | 0.97 | 0.58 | 0.76 | 0.99 | 0.68 | **0.85** |
| Hybrid (C) | 0.97 | 0.97 | 0.95 | 0.63 | 0.79 | 0.99 | 0.66 | 0.85 | 0.97 | 0.95 | 0.96 | 0.57 | 0.77 | 0.99 | 0.67 | 0.84 |
| Learner | 0.98 | 0.98 | 0.97 | 0.66 | 0.81 | 0.98 | 0.70 | 0.87 | 0.97 | 0.95 | 0.96 | 0.58 | 0.77 | 0.98 | 0.68 | 0.84 |
| Expert | 0.93 | 0.96 | 0.83 | 0.61 | 0.77 | 0.99 | 0.68 | 0.82 | 0.85 | 0.93 | 0.76 | 0.58 | 0.77 | 0.99 | 0.68 | 0.79 |

Table 15: Posterior contraction and explained variance across $1\,000$ test simulations from each parameter of model $\mathcal{M}_{11}$ separate for each summary type, and two training simulation budgets.

| | $v$ | $a$ | $\beta$ | $ndt$ | $\eta$ | Mean | $v$ | $a$ | $\beta$ | $ndt$ | $\eta$ | Mean |
|---|---|---|---|---|---|---|---|---|---|---|---|---|
| | Posterior contraction (uncertainty reduction) | | | | | | Explained variance ($R^2$ score, point recovery) | | | | | |
| **$1\,000$ Simulations** | | | | | | | | | | | | |
| Hybrid (D) | 0.99 | 0.98 | 0.97 | 0.98 | 0.99 | **0.98** | 0.99 | 0.95 | 0.97 | 0.98 | 0.99 | **0.97** |
| Hybrid (J) | 0.99 | 0.96 | 0.96 | 0.96 | 0.99 | 0.97 | 0.99 | 0.92 | 0.95 | 0.96 | 0.99 | 0.96 |
| Hybrid (C) | 0.99 | 0.95 | 0.90 | 0.94 | 0.99 | 0.95 | 0.99 | 0.92 | 0.93 | 0.96 | 0.99 | 0.96 |
| Learner | 0.99 | 0.75 | 0.86 | 0.41 | 0.99 | 0.80 | 0.99 | 0.71 | 0.80 | 0.44 | 0.99 | 0.79 |
| Expert | 0.99 | 0.94 | 0.90 | 0.96 | 0.99 | 0.96 | 0.99 | 0.89 | 0.91 | 0.95 | 0.99 | 0.95 |
| **$10\,000$ Simulations** | | | | | | | | | | | | |
| Hybrid (D) | 0.99 | 0.98 | 0.99 | 1.00 | 0.99 | **0.99** | 0.99 | 0.96 | 0.98 | 0.99 | 0.99 | **0.98** |
| Hybrid (J) | 0.99 | 0.98 | 0.98 | 0.99 | 0.99 | **0.99** | 0.99 | 0.96 | 0.98 | 0.99 | 0.99 | **0.98** |
| Hybrid (C) | 0.99 | 0.98 | 0.98 | 0.99 | 0.99 | **0.99** | 0.99 | 0.95 | 0.98 | 0.99 | 0.99 | **0.98** |
| Learner | 0.99 | 0.98 | 0.98 | 1.00 | 0.99 | **0.99** | 0.99 | 0.96 | 0.98 | 0.99 | 0.99 | **0.98** |
| Expert | 0.99 | 0.97 | 0.96 | 0.99 | 0.99 | 0.98 | 0.99 | 0.94 | 0.96 | 0.99 | 0.99 | 0.97 |

Table 16: Posterior contraction and explained variance across $1\,000$ test simulations from each parameter of model $\mathcal{M}_{12}$ separate for each summary type, and two training simulation budgets.

| | $v$ | $a$ | $\beta$ | $ndt$ | $\sigma_e$ | $\gamma$ | Mean | $v$ | $a$ | $\beta$ | $ndt$ | $\sigma_e$ | $\gamma$ | Mean |
|---|---|---|---|---|---|---|---|---|---|---|---|---|---|---|
| | Posterior contraction (uncertainty reduction) | | | | | | | Explained variance ($R^2$ score, point recovery) | | | | | | |
| **1 000 Simulations** | | | | | | | | | | | | | | |
| Hybrid (D) | 0.84 | 0.95 | 0.80 | 0.93 | 0.99 | 0.73 | **0.87** | 0.78 | 0.86 | 0.80 | 0.92 | 0.99 | 0.62 | **0.83** |
| Hybrid (J) | 0.84 | 0.92 | 0.80 | 0.89 | 0.99 | 0.71 | 0.86 | 0.75 | 0.81 | 0.74 | 0.84 | 0.99 | 0.59 | 0.79 |
| Hybrid (C) | 0.81 | 0.90 | 0.71 | 0.87 | 0.99 | 0.66 | 0.82 | 0.78 | 0.85 | 0.78 | 0.90 | 0.99 | 0.60 | 0.82 |
| Learner | 0.68 | 0.67 | 0.64 | 0.34 | 0.99 | 0.41 | 0.62 | 0.62 | 0.60 | 0.61 | 0.37 | 0.98 | 0.47 | 0.61 |
| Expert | 0.79 | 0.72 | 0.76 | 0.89 | 0.99 | 0.63 | 0.80 | 0.73 | 0.67 | 0.79 | 0.90 | 0.99 | 0.55 | 0.77 |
| **10 000 Simulations** | | | | | | | | | | | | | | |
| Hybrid (D) | 0.96 | 0.98 | 0.98 | 0.99 | 0.99 | 0.93 | **0.97** | 0.95 | 0.92 | 0.96 | 0.99 | 0.99 | 0.86 | **0.94** |
| Hybrid (J) | 0.96 | 0.99 | 0.98 | 0.99 | 0.99 | 0.93 | **0.97** | 0.94 | 0.92 | 0.96 | 0.98 | 0.99 | 0.84 | **0.94** |
| Hybrid (C) | 0.93 | 0.98 | 0.95 | 0.99 | 0.99 | 0.88 | 0.95 | 0.93 | 0.91 | 0.95 | 0.98 | 0.99 | 0.83 | 0.93 |
| Learner | 0.94 | 0.98 | 0.96 | 0.99 | 0.99 | 0.88 | 0.96 | 0.93 | 0.91 | 0.95 | 0.98 | 0.99 | 0.82 | 0.93 |
| Expert | 0.92 | 0.96 | 0.92 | 0.99 | 0.99 | 0.85 | 0.94 | 0.89 | 0.85 | 0.92 | 0.97 | 0.98 | 0.75 | 0.89 |

### C.2.6 2D-POSTERIOR EVALUATION

In the following, we present the bivariate posterior distributions of two parameters obtained from the remaining $14$ neurocognitive joint model, which were not included in the main text. These findings reaffirm the conclusions drawn in the main text: the direct Hybrid approach consistently outperforms both the Expert and Learner models, particularly in scenarios with limited data. Notably, under certain circumstances, the Learner is capable to catch up with the Hybrid, as more simulations become accessible for training. Conversely, the Expert model tends to show diminishing returns in performance gains with increased training data.

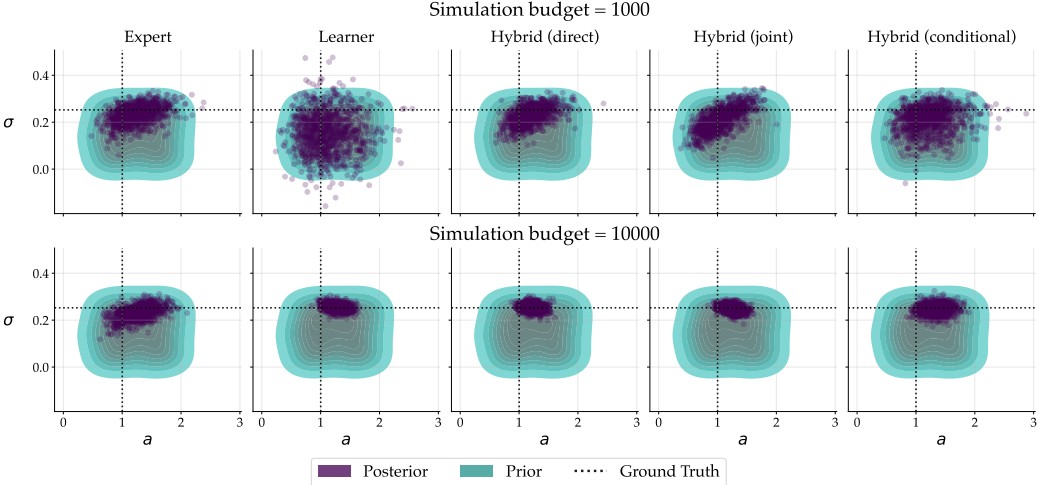

Figure 8: Example bivariate posteriors for two parameters obtained from model $\mathcal{M}_{1a}$. Additionally, the corresponding prior distributions are depicted in cyan. The true data generating parameters indicated by black dotted lines.

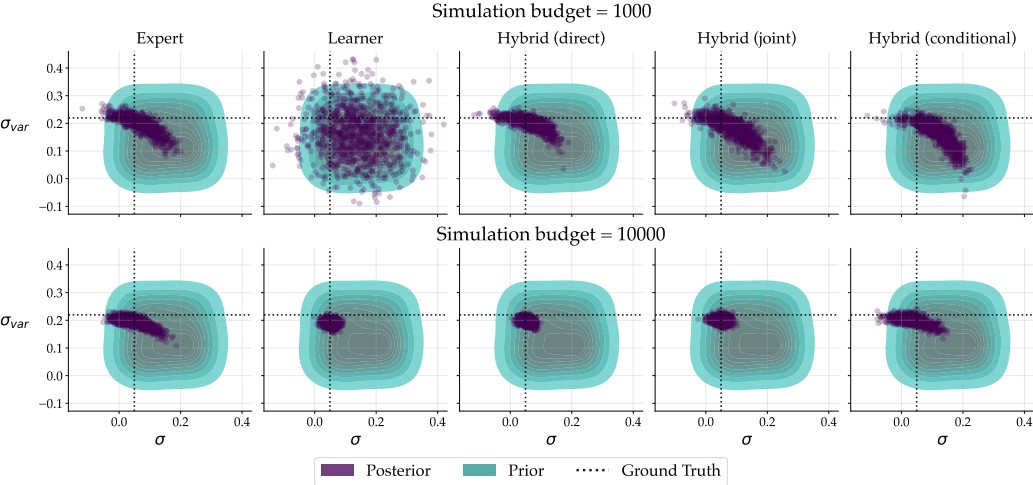

Figure 9: Example bivariate posteriors for two parameters obtained from model $\mathcal{M}_{1b}$. Additionally, the corresponding prior distributions are depicted in cyan. The true data generating parameters indicated by black dotted lines.

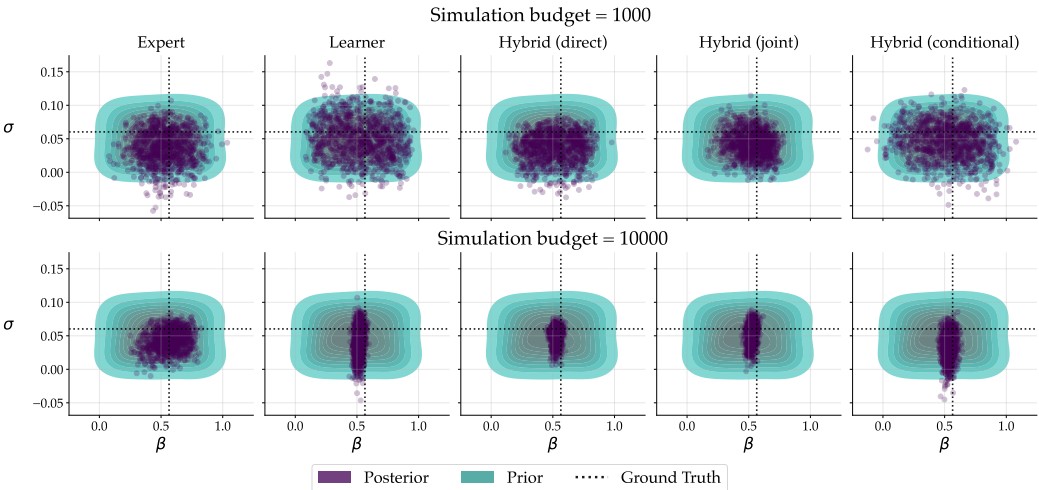

Figure 10: Example bivariate posteriors for two parameters obtained from model $\mathcal{M}_{1c}$. Additionally, the corresponding prior distributions are depicted in cyan. The true data generating parameters indicated by black dotted lines.

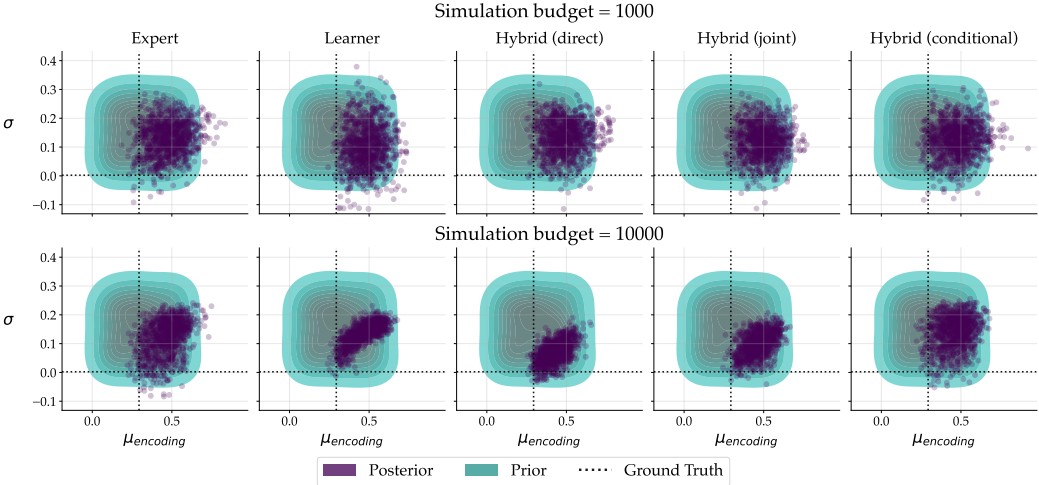

Figure 11: Example bivariate posteriors for two parameters obtained from model $\mathcal{M}_2$. Additionally, the corresponding prior distributions are depicted in cyan. The true data generating parameters indicated by black dotted lines.

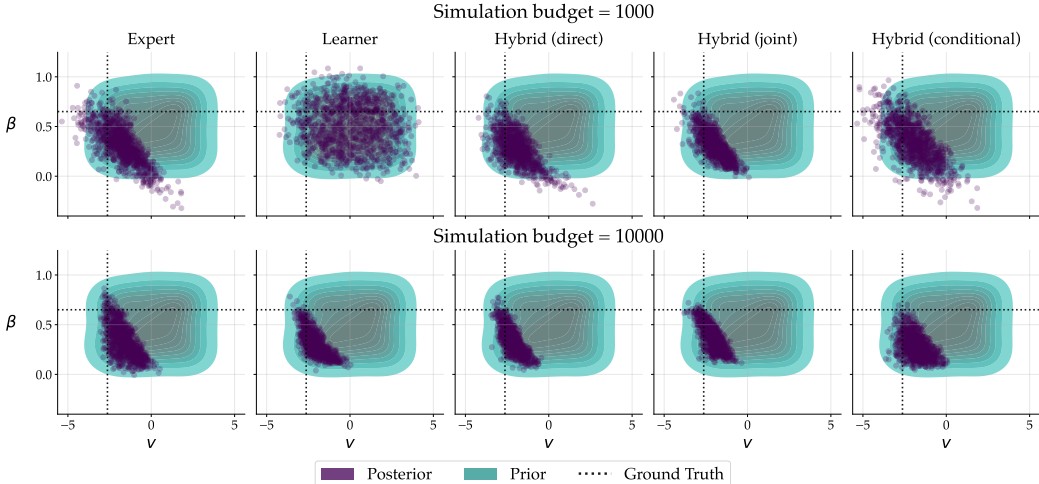

Figure 12: Example bivariate posteriors for two parameters obtained from model $\mathcal{M}_3$. Additionally, the corresponding prior distributions are depicted in cyan. The true data generating parameters indicated by black dotted lines.

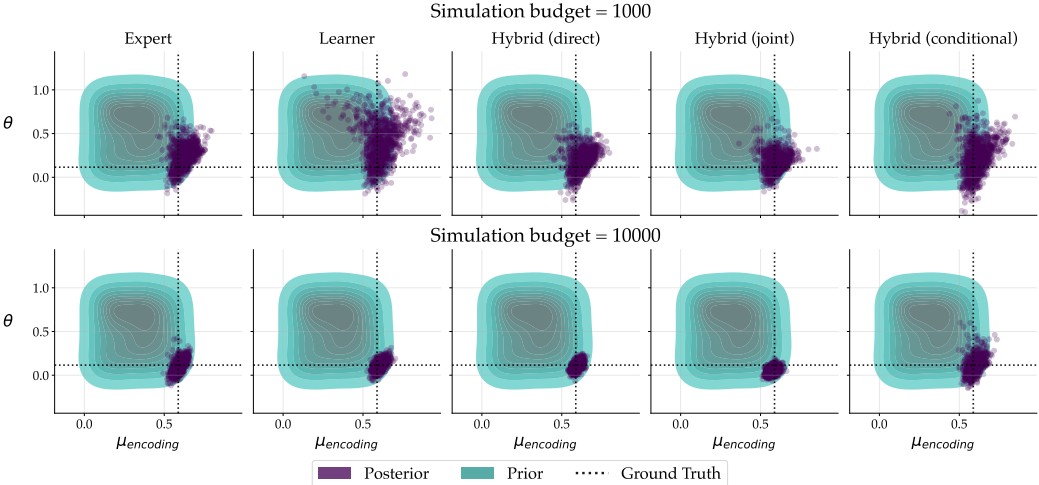

Figure 13: Example bivariate posteriors for two parameters obtained from model $\mathcal{M}_{4a}$. Additionally, the corresponding prior distributions are depicted in cyan. The true data generating parameters indicated by black dotted lines.

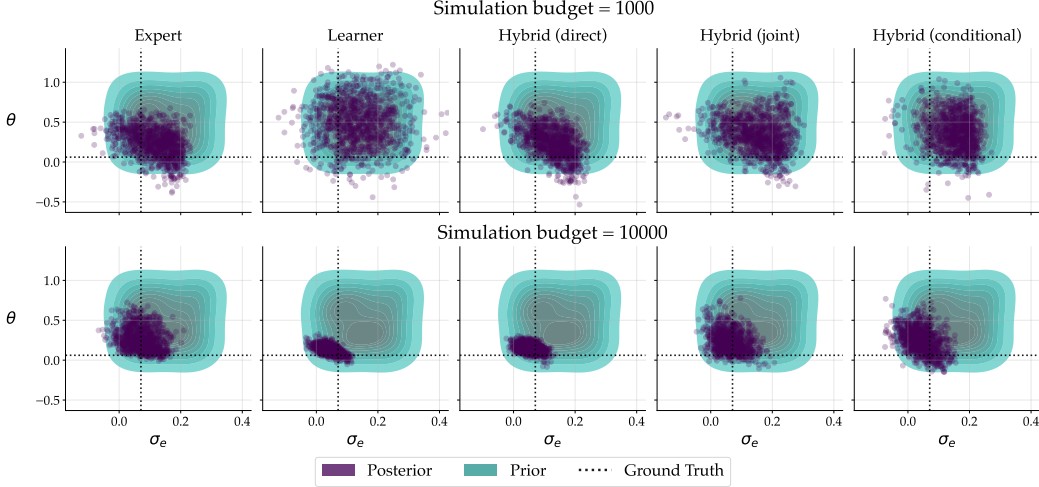

Figure 14: Example bivariate posteriors for two parameters obtained from model $\mathcal{M}_{4b}$. Additionally, the corresponding prior distributions are depicted in cyan. The true data generating parameters indicated by black dotted lines.

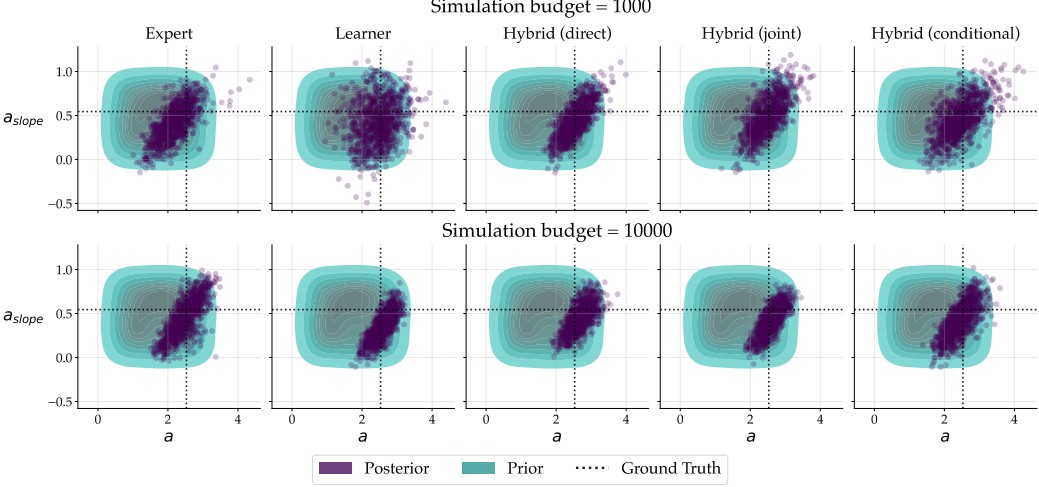

Figure 15: Example bivariate posteriors for two parameters obtained from model $\mathcal{M}_5$. Additionally, the corresponding prior distributions are depicted in cyan. The true data generating parameters indicated by black dotted lines.

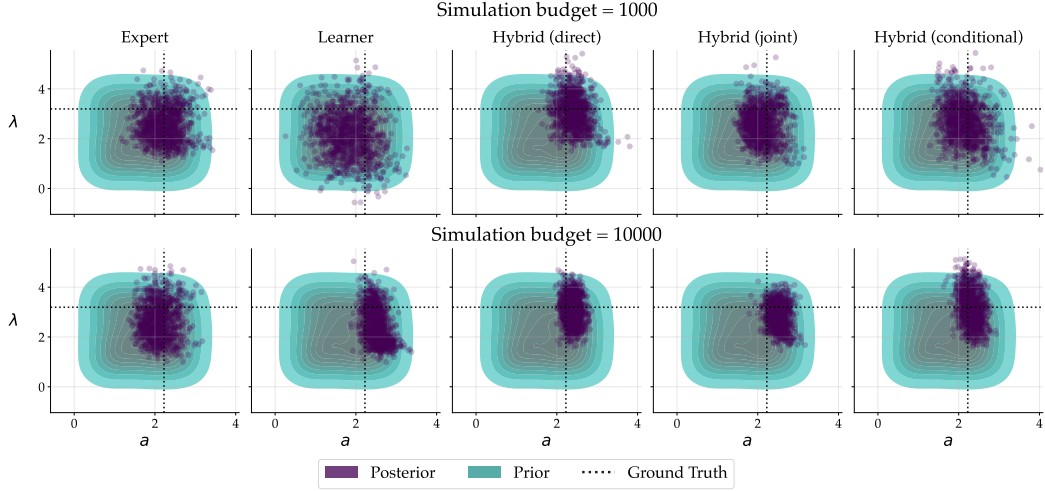

Figure 16: Example bivariate posteriors for two parameters obtained from model $\mathcal{M}_6$. Additionally, the corresponding prior distributions are depicted in cyan. The true data generating parameters indicated by black dotted lines.

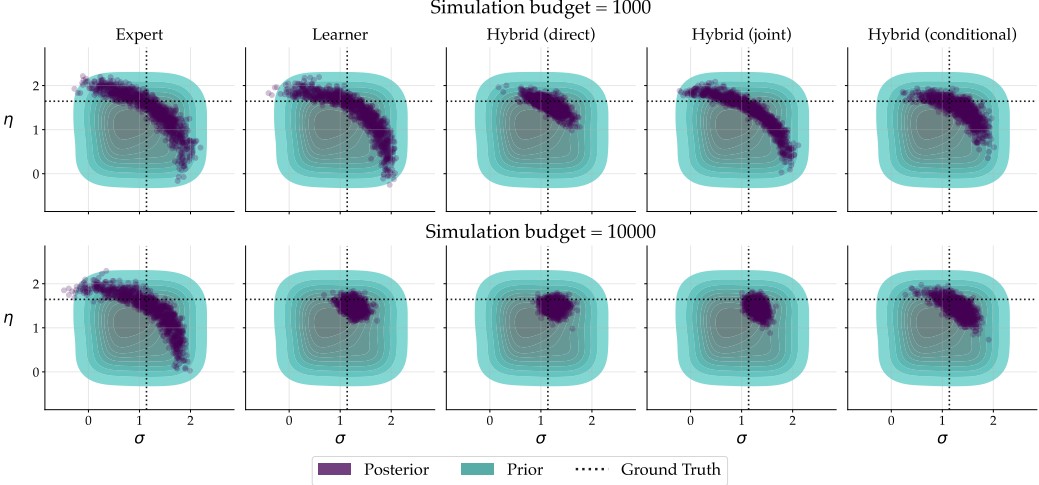

Figure 17: Example bivariate posteriors for two parameters obtained from model $\mathcal{M}_7$. Additionally, the corresponding prior distributions are depicted in cyan. The true data generating parameters indicated by black dotted lines.

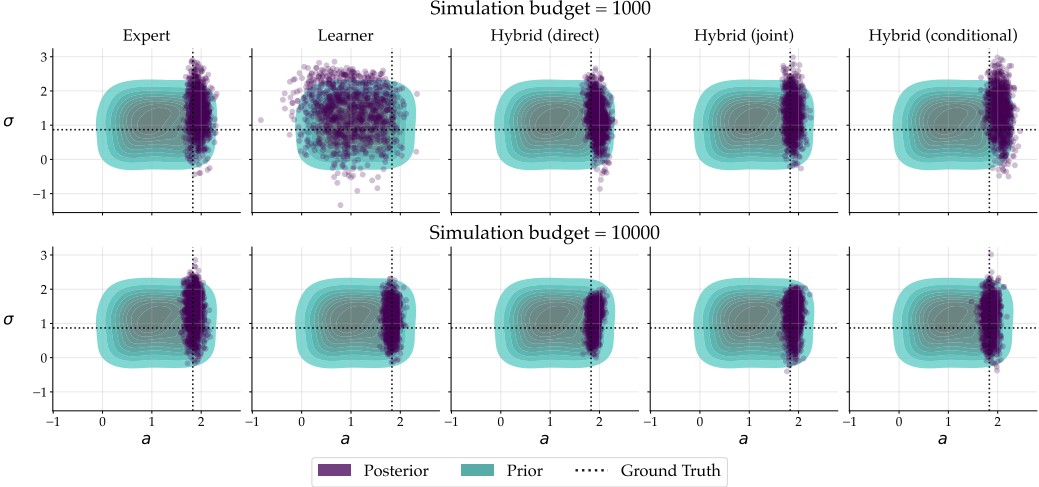

Figure 18: Example bivariate posteriors for two parameters obtained from model $\mathcal{M}_8$. Additionally, the corresponding prior distributions are depicted in cyan. The true data generating parameters indicated by black dotted lines.

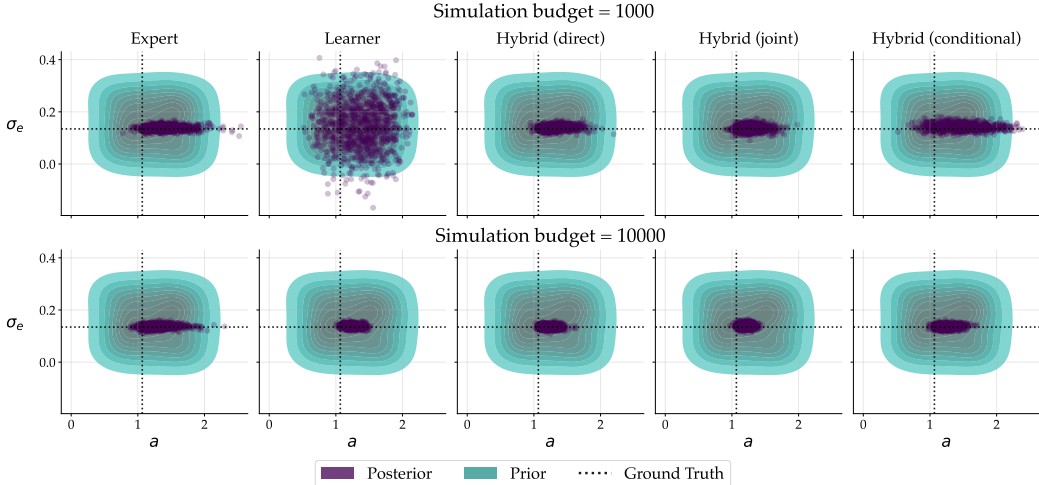

Figure 19: Example bivariate posteriors for two parameters obtained from model $\mathcal{M}_9$. Additionally, the corresponding prior distributions are depicted in cyan. The true data generating parameters indicated by black dotted lines.

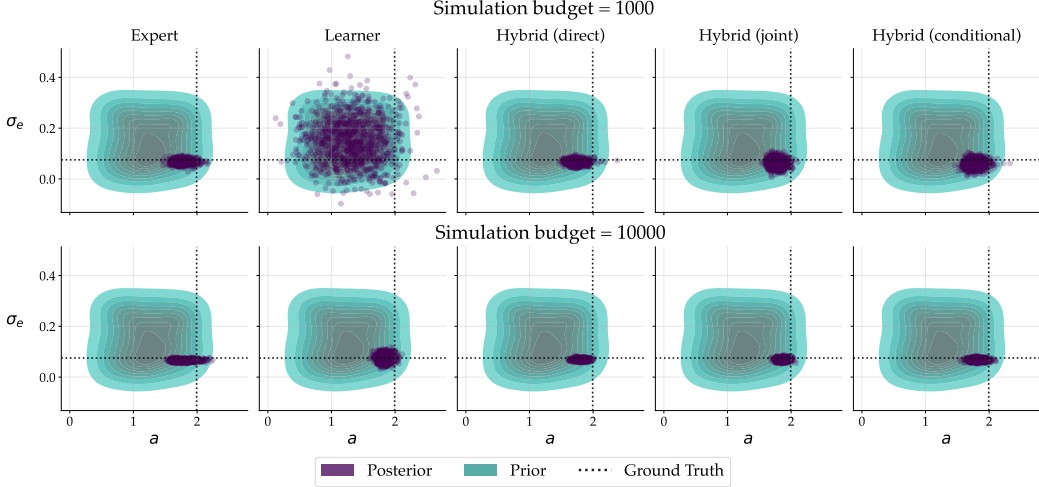

Figure 20: Example bivariate posteriors for two parameters obtained from model $\mathcal{M}_{10}$. Additionally, the corresponding prior distributions are depicted in cyan. The true data generating parameters indicated by black dotted lines.

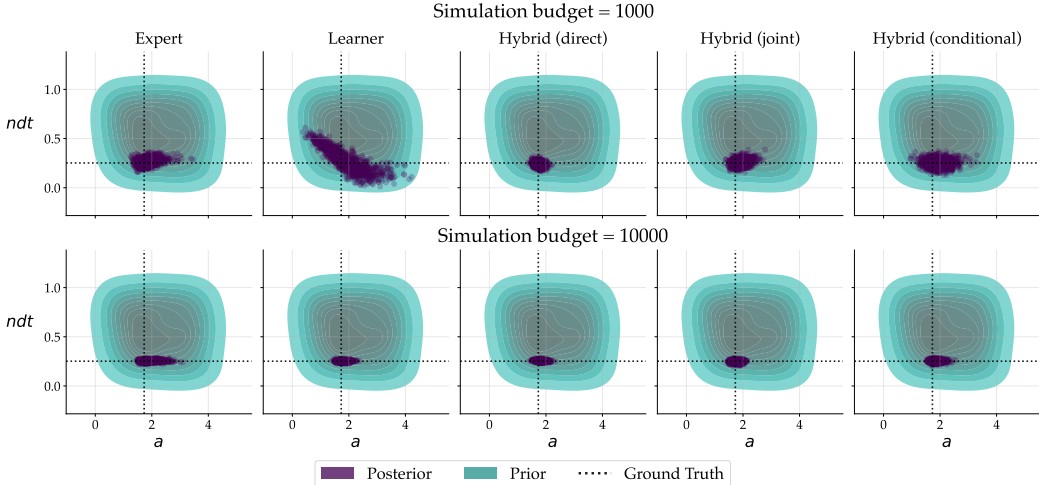

Figure 21: Example bivariate posteriors for two parameters obtained from model $\mathcal{M}_{11}$. Additionally, the corresponding prior distributions are depicted in cyan. The true data generating parameters indicated by black dotted lines.

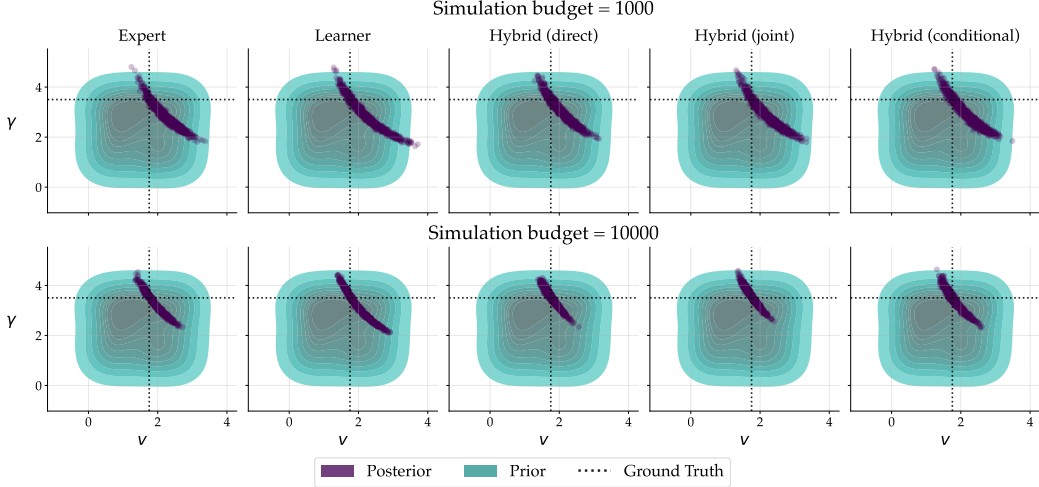

Figure 22: Example bivariate posteriors for two parameters obtained from model $\mathcal{M}_{12}$. Additionally, the corresponding prior distributions are depicted in cyan. The true data generating parameters indicated by black dotted lines.

### C.2.7 Amortized Bayesian model comparison

As an additional assessment of the performance of the different summary configurations(Expert, Learner, Hybrid), we also performed amortized Bayesian model comparison (Radev et al., 2021a). This involves training a posterior model probability network using these different approaches and then assessing their performance in comparing the 15 joint models based on simulated data. We employed the following set of 4 key metrics for assessing the comparison quality: i) accuracy of model detection, ii) a combination of precision and recall measured by the F1-Score, iii) calibration evaluated using the *expected calibration error* (ECE), and iv) mean absolute error (MAE) between true and predicted model. The neural networks were trained on a total of 50 000 synthetic data sets generated from the 15 joint models.

Upon reviewing the results presented in Table 17, it is evident that the Hybrid approximator demonstrates superior performance in 3 out of 4 metrics. In the fourth metric, the ECE, it performs nearly as well as the Expert approach.

Table 17: Amortized Bayesian model comparison performance of the there amortizer types measured with 4 metrics: the accuracy of model detection, a combination of precision and recall (F1-Score), expected calibration error (ECE), and mean absolute error (MAE).

|         | Accuracy | F1-Score | ECE   | MAE   |
|---------|----------|----------|-------|-------|
| Hybrid  | **0.716** | **0.714** | 0.013 | **0.046** |
| Learner | 0.686    | 0.687    | 0.020 | 0.048 |
| Expert  | 0.612    | 0.608    | **0.012** | 0.061 |

### C.3 Experiment 3: Lotka-Volterra dynamic model

The implementation of the stochastic Lotka-Volterra (LV) simulator follows the formulation in Lueckmann et al. (2021). The LV model describes the non-linear dynamics of ecological systems in which a population of predators interacts with a population of prey, resulting in bivariate time series $\boldsymbol{X} \in \mathbb{R}^{T \times 2}$. Four rate parameters $\boldsymbol{\theta} \in \mathbb{R}^4$ govern the deterministic latent dynamics

$$\frac{dx_1}{dt} = \theta_1 x_1 - \theta_2 x_1 x_2 \tag{28}$$

$$\frac{dx_2}{dt} = -\theta_3 x_2 + \theta_4 x_1 x_2, \tag{29}$$

which are subsequently corrupted through a Log-Normal observation model (Lueckmann et al., 2021) to arrive at the observed time series $\boldsymbol{X} \in \mathbb{R}^{T \times 2}$. Here, we use the original setting with $T = 30$ and initial conditions $(x_1(0), x_2(0)) = (30, 1)$.

### C.4 Expert Statistics

The artificial expert is a pre-trained heteroskedastic regression neural network extending the approach of (Jiang et al., 2017) by learning the posterior variances alongside the posterior means (Kendall & Gal, 2017). The expert network is pre-trained on $B = 1\,000$ simulations and thus represents a noisy, but still knowledgeable expert capable of extracting some useful information from the observed time series.

### C.5 Summary Networks

### C.5.1 Transformer-based networks

The transformer summary network features a stack of 2 multi-head attention encoders with 4 attention heads (key, query, and value dimensions of 32) and standard positional encoding. In addition, it borrows ideas from temporal fusion architectures (Lim et al., 2021) by including an LSTM network for dynamically learning temporal information. The LSTM network processes the multivariate

time series and returns a template vector which is combined with the encoded time series via cross-attention.

### C.5.2 SEQUENTIAL NETWORKS

The sequential summary network represents a combination of a recurrent and a 1D convolutional network, an architecture that has successfully been used in the context of simulation-based inference for dynamic time series models (Radev et al., 2021b).

### C.6 INFERENCE NETWORKS

The inference network for inferring the hidden parameters is a normalizing flow parameterized by 4 spline coupling layers (Durkan et al., 2018), 2 hidden layers containing 128 units each, ReLU activations, and a spherical Gaussian as a base (latent) distribution. The generative summary network for the conditional hybrid is a normalizing flow parameterized by 6 affine coupling layers (Ardizzone et al., 2019) with 2 hidden layers containing 128 units each, ReLU activations, and a diagonal multivariate Student-T latent distribution with $df = 100$.

## C.7 Additional results

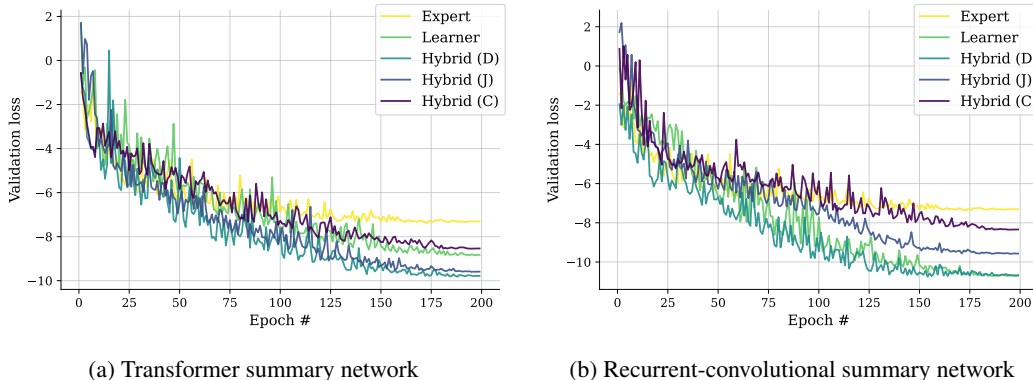

(a) Transformer summary network

(b) Recurrent-convolutional summary network

Figure 23: Validation loss trajectories obtained by training all network configurations for 200 epochs with a low simulation budget $B = 1000$.

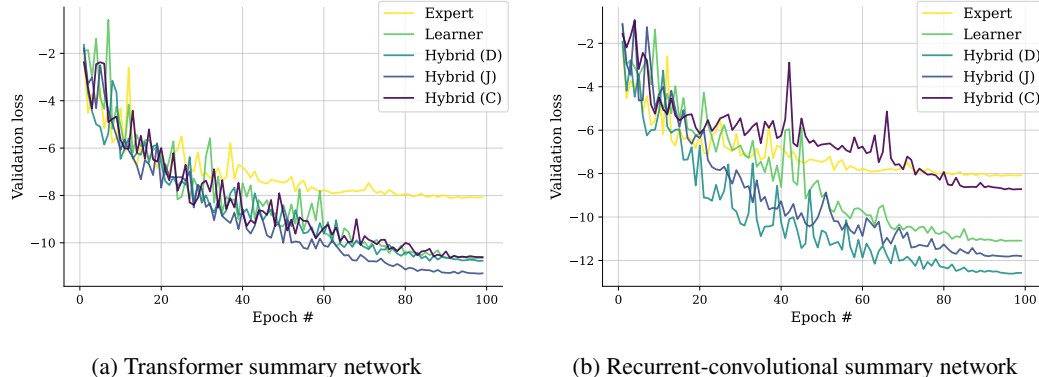

(a) Transformer summary network

(b) Recurrent-convolutional summary network

Figure 24: Validation loss trajectories obtained by training all network configurations for 100 epochs with a medium simulation budget $B = 5000$.

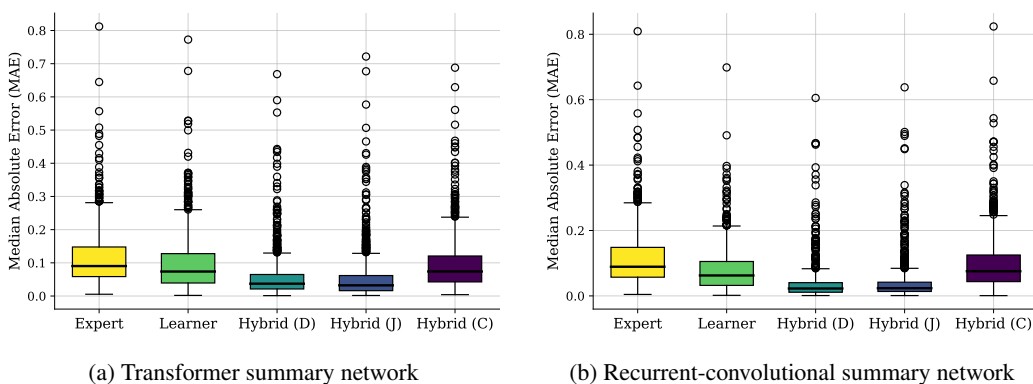

(a) Transformer summary network

(b) Recurrent-convolutional summary network

Figure 25: **Lotka-Volterra model with artificial expert** ($B = 1000$ setting). Estimation error of all methods on the same 1000 test simulations. The estimation error is measured as the median absolute deviation (MAE) between posterior medians and ground-truth values aggregated across all four model parameters.

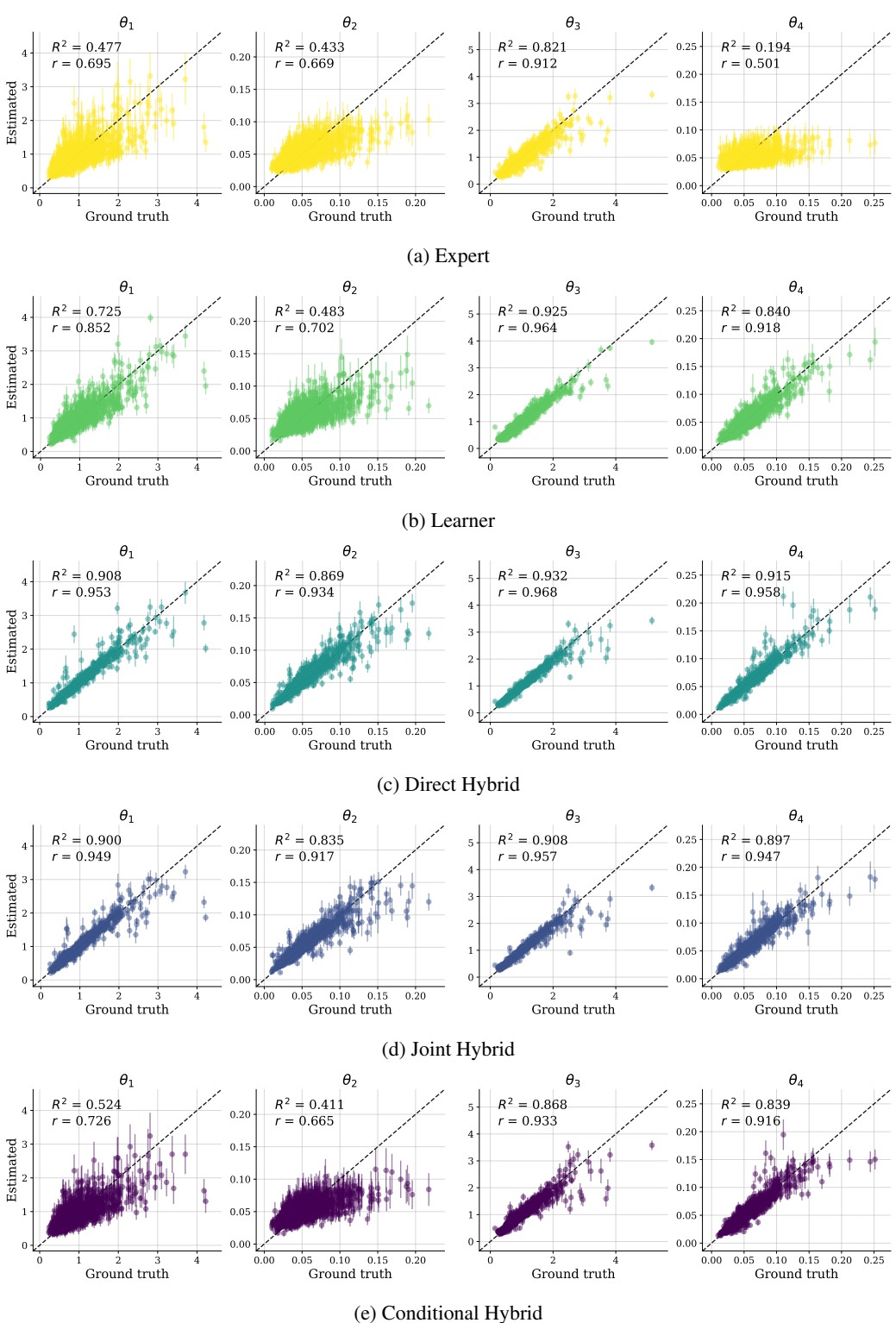

Figure 26: Detailed parameter estimation results obtained using a recurrent-convolutional summary network with a low simulation budget of $B = 1000$.

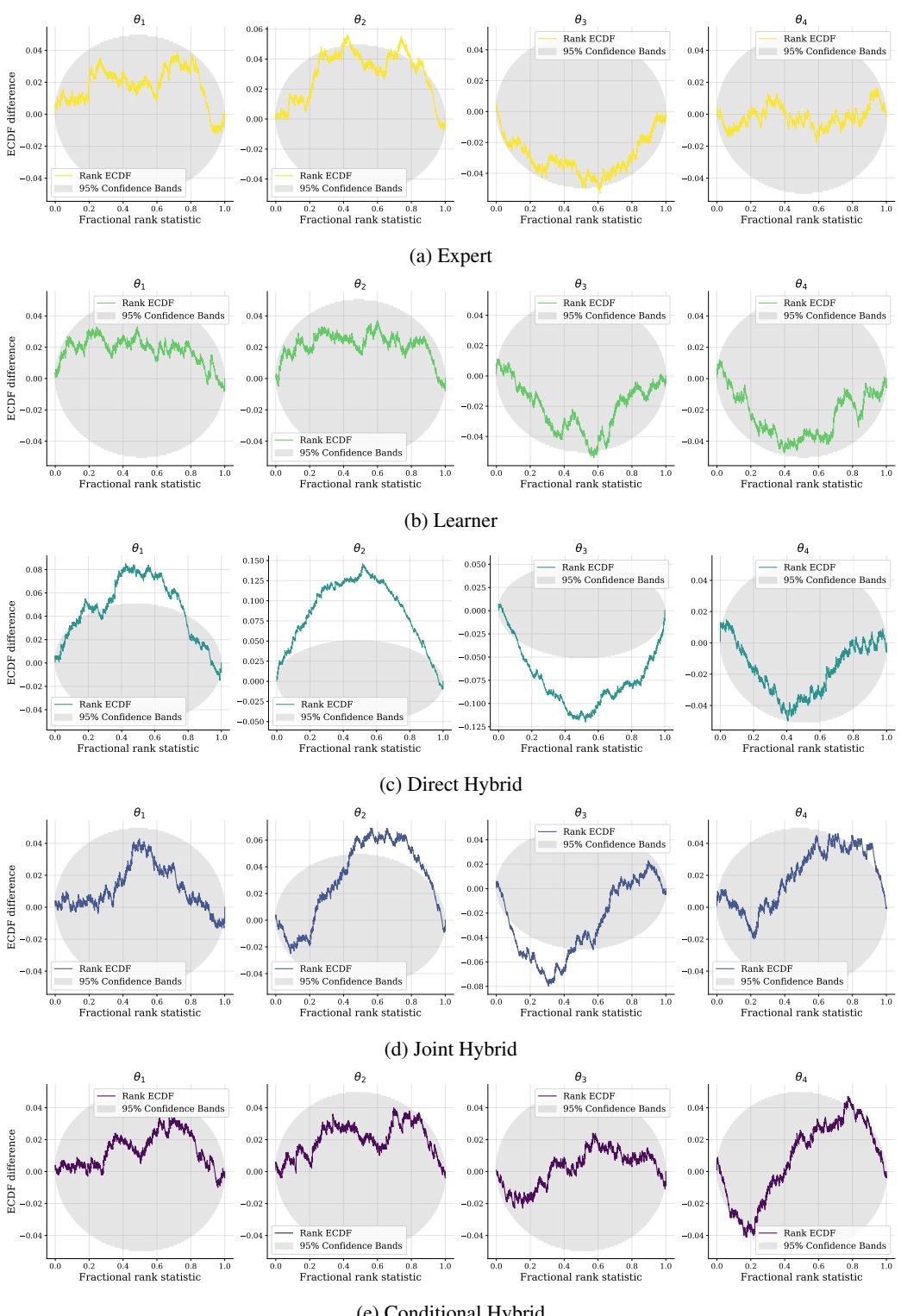

Figure 27: Detailed simulation-based calibration (Talts et al., 2018) results obtained using a recurrent-convolutional summary network with a low simulation budget of $B = 1000$.

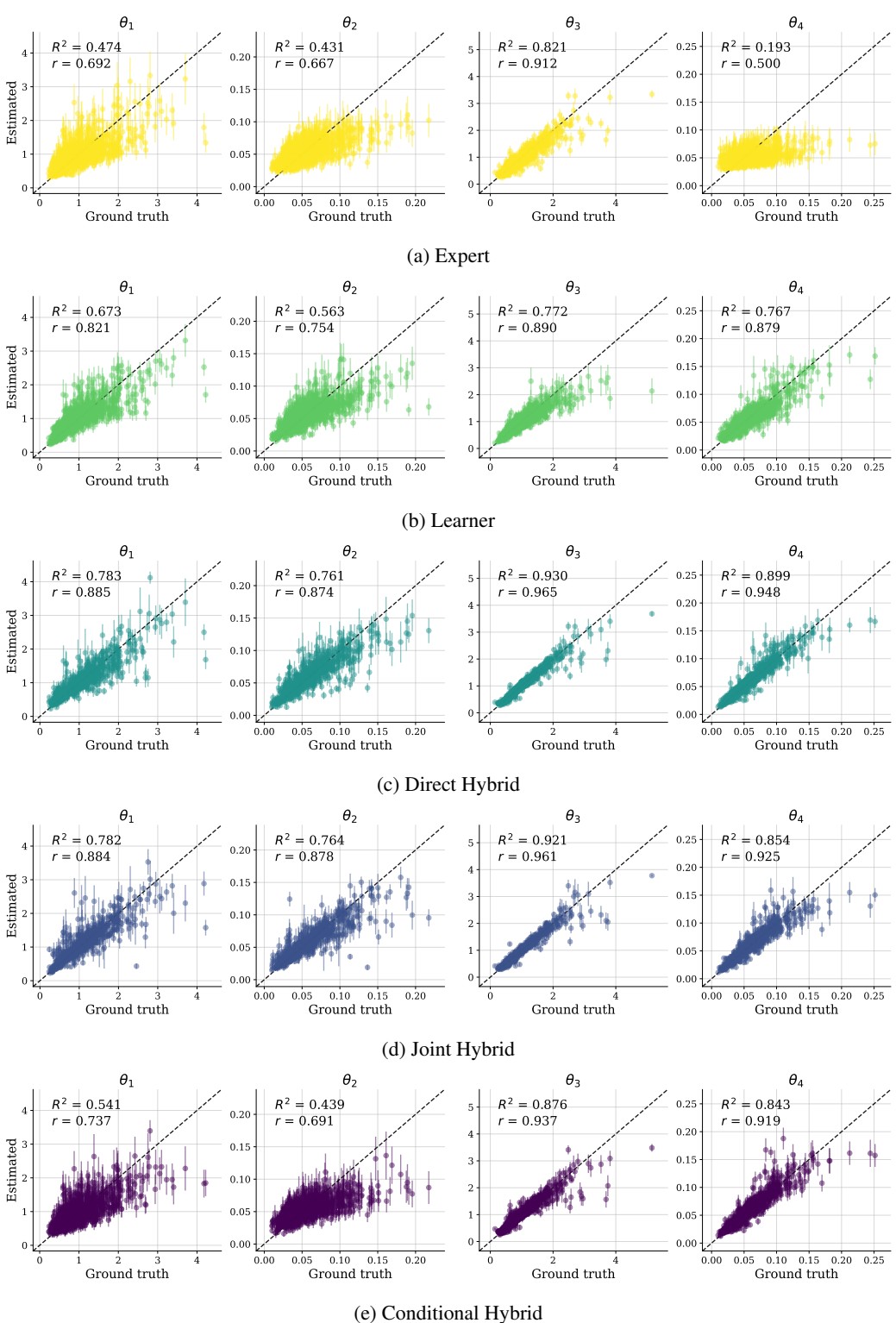

Figure 28: Detailed parameter estimation results obtained using a transformer summary network with a low simulation budget of $B = 1000$.

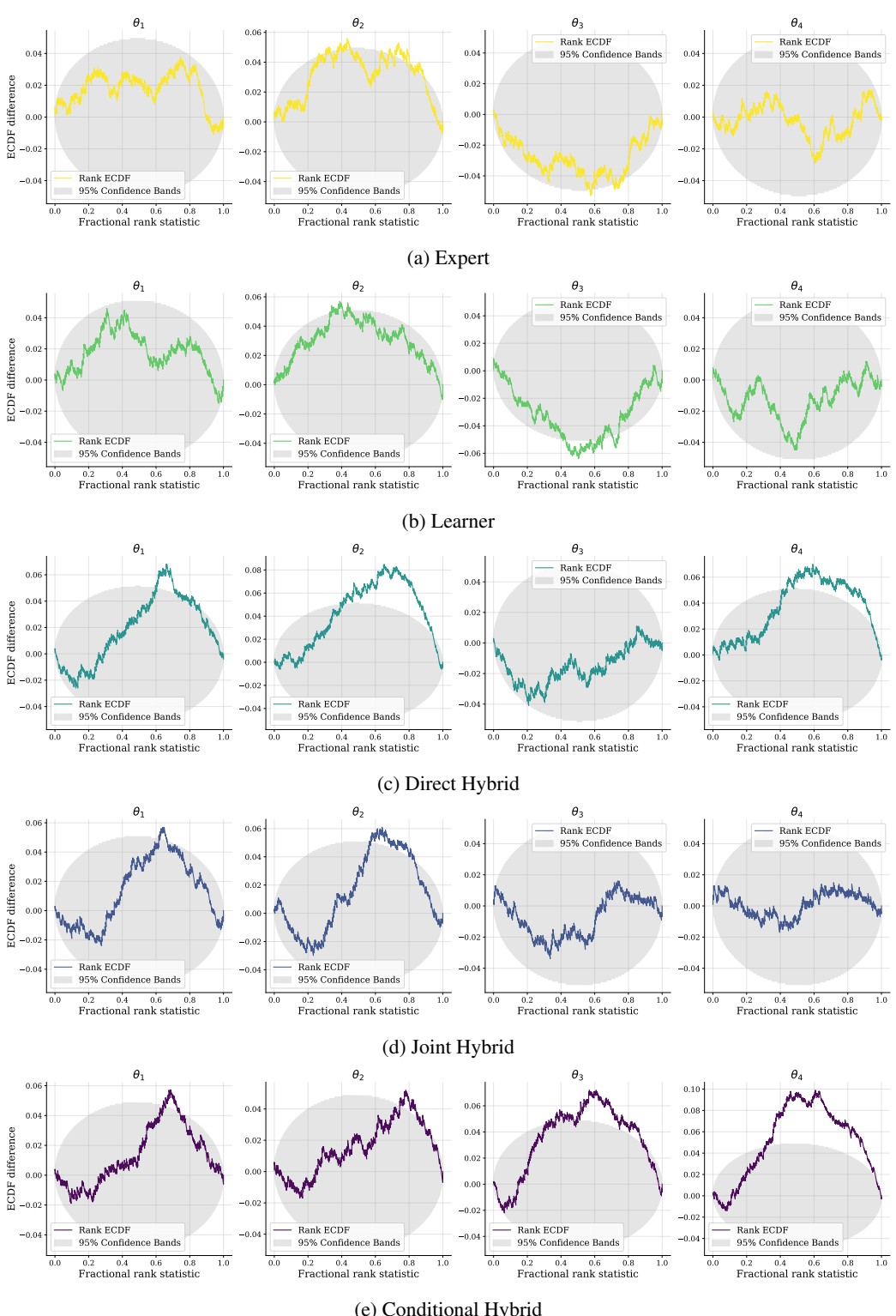

Figure 29: Detailed simulation-based calibration (Talts et al., 2018) results obtained using a temporal fusion transformer summary network with a low simulation budget of $B = 1000$.

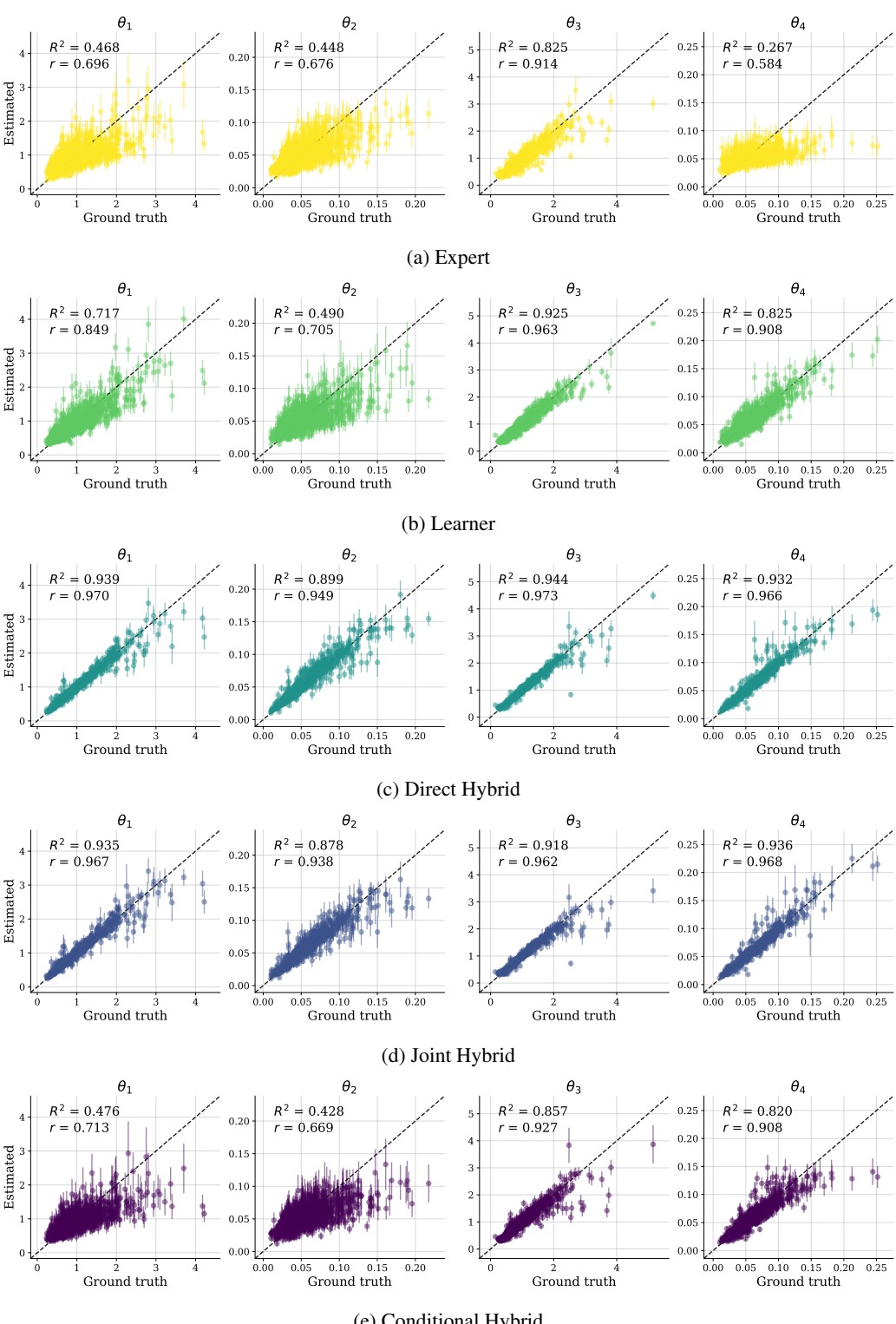

Figure 30: Detailed parameter estimation results obtained using a recurrent-convolutional summary network with a medium simulation budget of $B = 5000$.

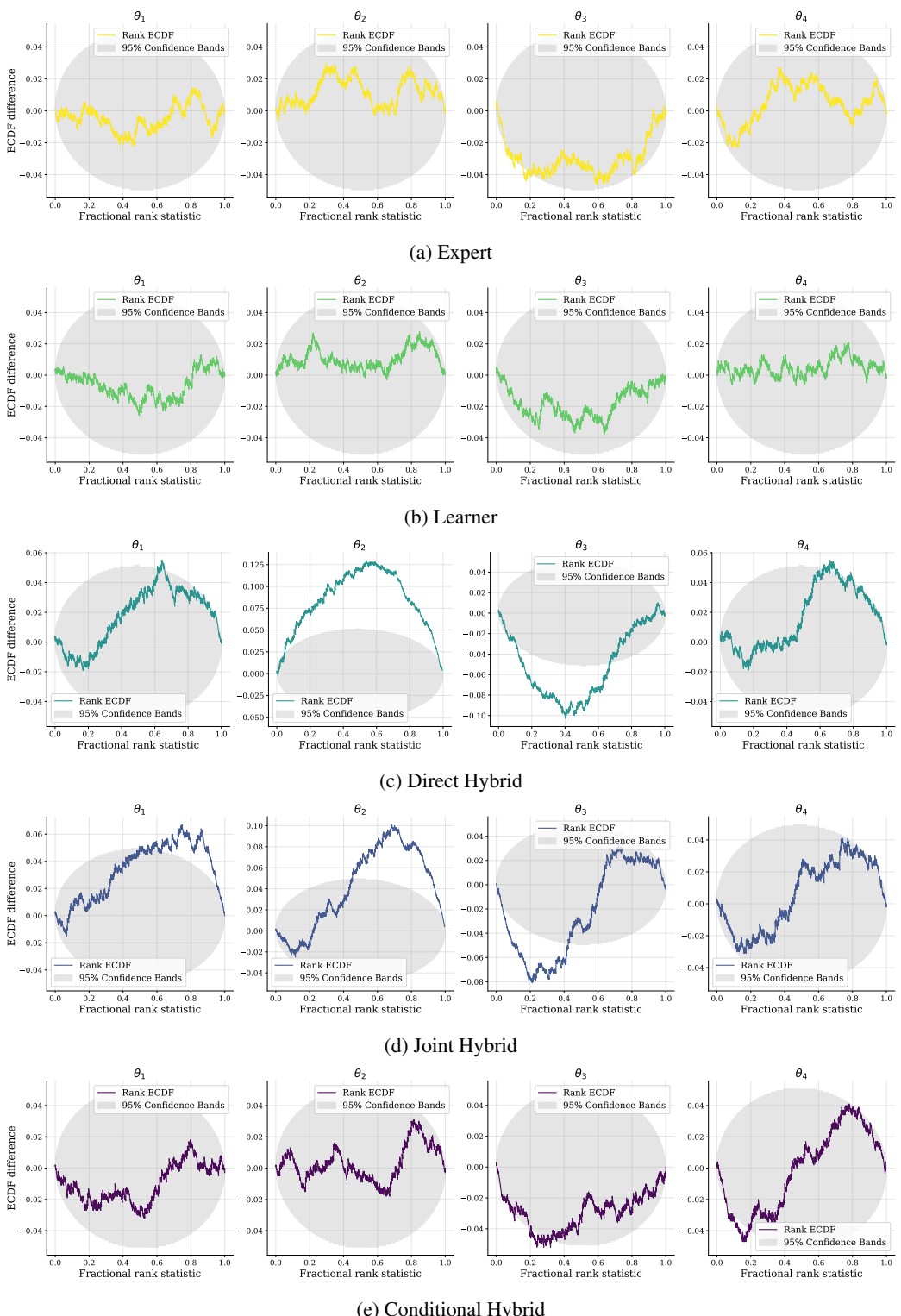

Figure 31: Detailed simulation-based calibration (Talts et al., 2018) results obtained using a recurrent-convolutional summary network with a medium simulation budget of $B = 5000$.

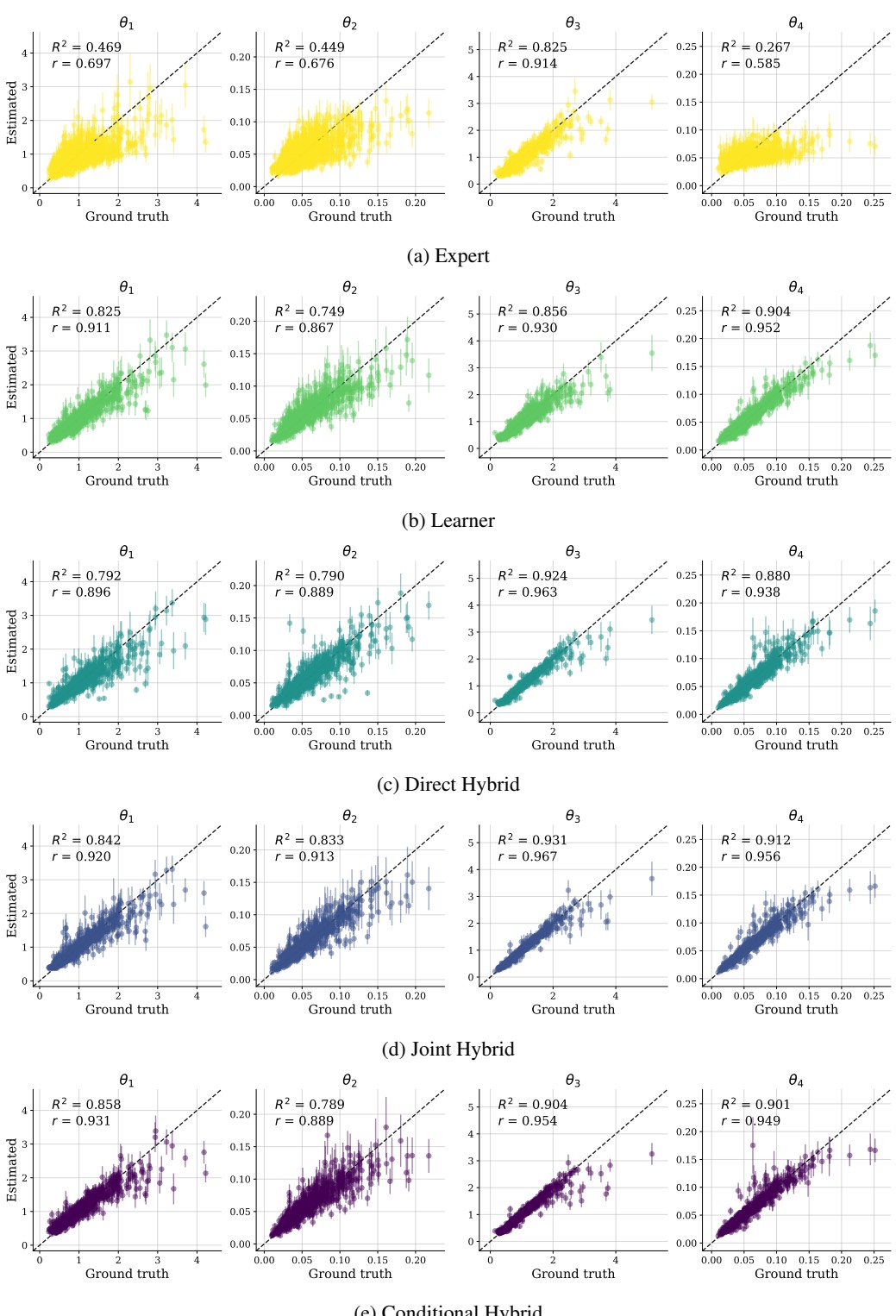

Figure 32: Detailed parameter estimation results obtained using a transformer summary network with a medium simulation budget of $B = 5000$.

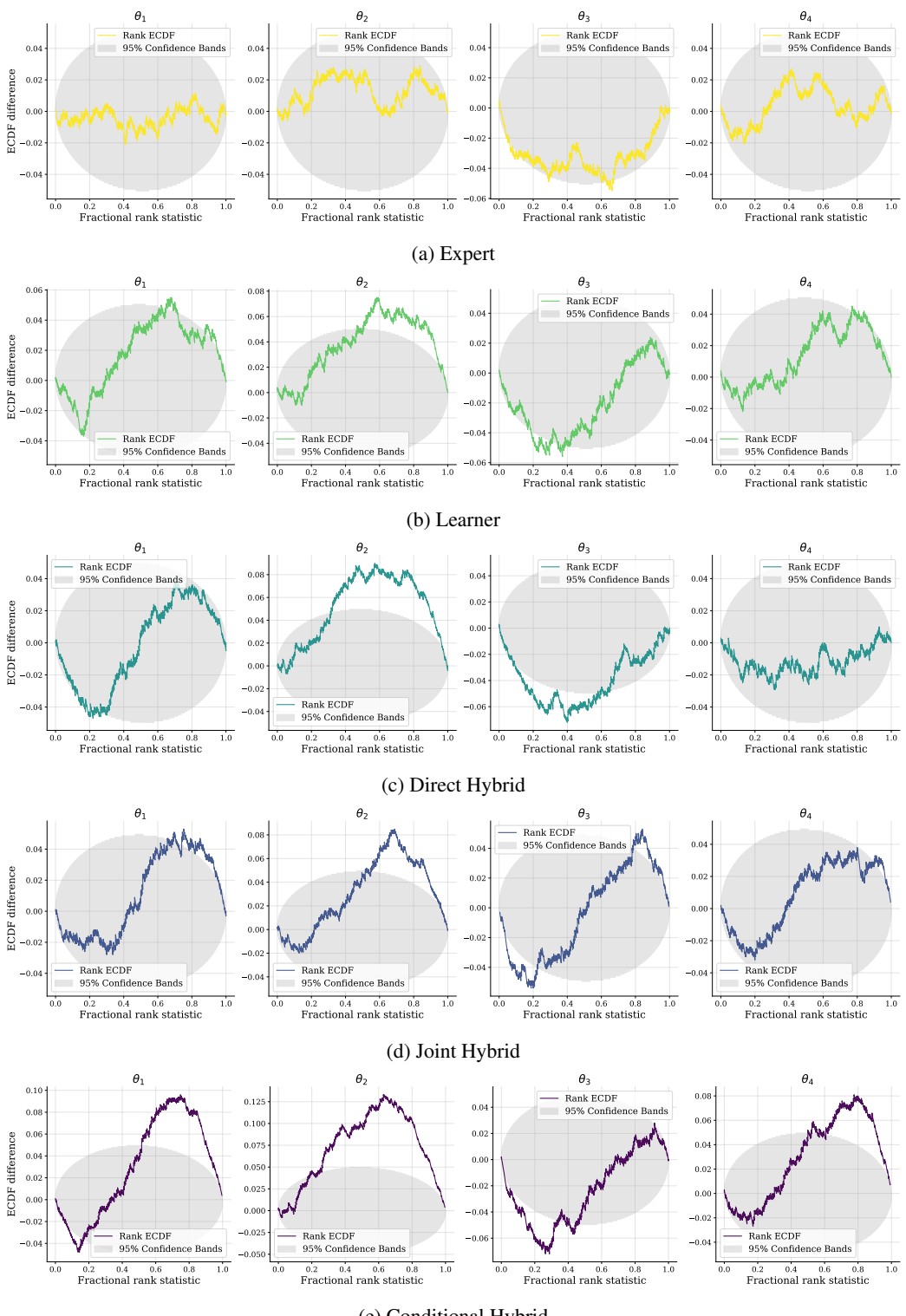

Figure 33: Detailed simulation-based calibration (Talts et al., 2018) results obtained using a temporal fusion transformer summary network with a medium simulation budget of $B = 5000$.

