# OpenReview forum: "Amortized Bayesian Inference with Hybrid Expert-in-the-Loop and Learnable Summary Statistics"
_ICLR.cc/2024/Conference — ICLR 2024 Conference Withdrawn Submission_

### Official Review · Reviewer_QZwH · 2023-10-26

**Soundness:** 3 good
**Presentation:** 3 good
**Contribution:** 2 fair
**Rating:** 3
**Confidence:** 4

**Summary:**

The authors consider the problem of Amortized Bayesian Inference (ABI), and in particular the framework set up by simulation-based inference (SBI), which refers to minimization of the Forward KL divergence between the ground truth posterior distribution and the variational approximation. While there have been a number of works that tackle the problem of ABI, and SBI in particular, the authors look at cases when summary statistics of the data are partially available. In particular, the authors propose models that take in both the partial summary statistics defined by an expert in the domain, as well as some learned summary statistics, and then learn the amortized predictor based on the combination of the two. They also perform analysis into different bottlenecks when leveraging the combination of the two statistics, eg. using independence assumptions or low mutual information based settings, and show that typically just using a naive combination of the two summary statistics outperforms the other candidates.

**Strengths:**

- The problem is well formulated and studied, and the mathematical grounding behind the setup is well presented.
- The authors consider a relevant scenario where partial expert statistics may be available, which can be the case in some scientific applications where some domain knowledge can be leveraged.

**Weaknesses:**

- The motivation behind proposing both joint and conditional hybrid methods is unclear, as requiring an independence structure in the summary statistics seems to be fairly limiting (J). In particular, it might make more sense to have independence between the expert and learned summary statistics (so a block-sparse structure). It is not entirely clear why one should care about independence in the summary statistics. The same reasoning holds for the conditional hybrid model.
- The models considered in this work are fairly small, which could explain the optimization challenges and suboptimality seen in the Learner-based setup. In particular, it would be nice if the authors could replicate these results with a fairly overparameterized model (eg. 4-6 layers, 4-8 heads, and 512 dimensions).
- Additionally, some analysis into increasing the batch size, the number of simulations seen, and the number of context points provided would be helpful in understanding if given enough training budget and examples, does the learner still severely lag the hybrid systems.
- While it seems like a relevant practical problem where partial expert statistics might be present, the authors perform experiments solely on synthetic domains. It would strengthen the paper substantially if the authors could identify a real-world problem setup where indeed some expert knowledge is present, simulations from such a system can be obtained, and at inference new data as well as partial summary statistics are provided which can be used to perform bayesian inference on this new data.
- Another related problem is the case where between different simulations, different amounts of expert statistics are known, which would be closer tied to real world setting where based on the data obtained, even with the same simulation engine different partial statistics might be available.

I believe that a fundamental limitation of this work is the lack of motivation for proposing Joint and Conditional Hybrid models as there is no reason why they would perform superior to the straight-forward direct method. They might be nice ablations to do, but a lot of this analysis can be pushed to the appendix to make room for more real-world applications on some scientific domain, or analysis into the trends of the learner with increasing data and complexity.

Finally, the authors should also consider the following relevant work on ABI

*Kingma, Diederik P., and Max Welling. "Auto-encoding variational bayes." arXiv preprint arXiv:1312.6114 (2013).*

*Garnelo, M., Schwarz, J., Rosenbaum, D., Viola, F., Rezende, D. J., Eslami, S. M., & Teh, Y. W. (2018). Neural processes. arXiv preprint arXiv:1807.01622.*

*Mittal, S., Bracher, N. L., Lajoie, G., Jaini, P., & Brubaker, M. A. (2023, July). Exploring Exchangeable Dataset Amortization for Bayesian Posterior Inference. In ICML 2023 Workshop on Structured Probabilistic Inference {\&} Generative Modeling.*

**Questions:**

- Apologies, but the section on Conditional Hybrid Method is unclear and not very well presented. While I understand the main goal, the use of $F_\psi$ is not clear, especially with the overloading of notation in $F_\psi$ and $q_\psi$. How does the training procedure lead to training of $F_\psi$ and what is it actually trying to model? To me it seems like the goal is to model learner statistics that have low mutual information from the expert statistics, but the notation and math is not clear.
- In Table 1, the performance obtained by the Hybrid (D) model is better than that obtained by the Expert summary statistics. For any given task, leveraging solely the expert statistics should be the oracle. Am I understanding this incorrectly, or is the expert statistics used in this data partial or approximate?

---

### Official Review · Reviewer_727i · 2023-10-30

**Soundness:** 2 fair
**Presentation:** 3 good
**Contribution:** 1 poor
**Rating:** 3
**Confidence:** 4

**Summary:**

The paper introduces three methods to leverage both handcrafted and learned summary statistic for amortized Bayesian inference. The first method feed a concactenation of the learned and handcrafted summary stastistics to the inference network. The inference and summary networks are hence trained jointly. The second method aims to force all the summary statistics to be independent. The third method forces independence between the handcrafted and learned summary statistics. Empirically, it is shown that the first method outperforms both the second and third methods as well as methods using only learned or handcrafted summary statistics on almost all benchmarks.

**Strengths:**

* This is overall a high-quality manuscript. It is well written and the figures are nice.
* The first and second methods are sound. It provides an efficient and easy-to-implement solution to the problem.

**Weaknesses:**

* My biggest concern is that none of the three presented methods are both proven to be significant and novel. Leveraging handcrafted summary statistics is relevant to improve performance in low-data regimes. The first method is hence significant. However,  feeding a concatenation of the learned and handcrafted summary statistics to the inference network can be thought of as feature engineering which is not novel to me. The second and third methods are novel to me. However, their added value is not straightforward. Indeed the second method aims to "learn a decorrelated joint representation of the expert and learnable summary statistics" and the third method aims to "learn summary representations that are guaranteed to be statistically independent from the expert summaries without transforming the latter in any way". However, it is not clear to me why we would want such a thing. Empirically, it has been shown in this paper to yield worse results than the first method. The authors mention that it could improve interpretability and robustness but further explanations would be needed.

* I think there is an error in the proof in Appendix B.4. To go from equation (21) to (22), $D_{KL}(p(L_{\vartheta}(X) | H(X)) || q_\psi(L_{\vartheta}(X) | H(X)))$ has been replaced by $\log p(L_{\vartheta}(X) | H(X)) - \log q_\psi(L_{\vartheta}(X) | H(X))$ while it should have been replaced by $E_{p(L_{\vartheta}(X) | H(X))} \left[ \log p(L_{\vartheta}(X) | H(X)) - \log q_\psi(L_{\vartheta}(X) | H(X)) \right]$

* Experiments evaluate the quality of the inference. However, the added value of the second and third methods lies in the independence between summary statistics and this has not been evaluated empirically.

* In the first sentence of section 3.5 "learn summary representations that are guaranteed to be statistically independent from the expert summaries without transforming the latter in any way", I think the word guaranteed is too strong here. Indeed this is only the case when the global optimum is reached. This further strengthens the need to empirically evaluate it.

**Questions:**

* Could you clarify why we need independence between summary statistics?
* Did I miss something that allows to go from equation (21) to equation (22)?

---

### Official Review · Reviewer_u538 · 2023-10-30

**Soundness:** 3 good
**Presentation:** 3 good
**Contribution:** 3 good
**Rating:** 6
**Confidence:** 2

**Summary:**

This paper studies how to combine summary statistics acquired from experts and a neural network (learner). The paper studies three strategies for combination: Direct hybrid method, which directly concatenate all the statistics; the joint hybrid method which transforms the experts' summary statistics via a network; and the conditional hybrid method, which uses an auxiliary to learn representations independent from the expert statistics. The paper then evaluates these methods on a wide range of tasks. Broadly, the proposed hybrid approach shows superior performance compared with either experts model or the learner model alone. Interestingly, the most naive approach of simply concatenating all the summary statistics shows the best performance among all the variations. Nevertheless, I believe the results and insights provided by the paper could serve as a starting point for the community to further investigate the exper-in-the-loop SBI.

**Strengths:**

- The paper proposes three different approaches for combining summary statistics from experts. All of them are well-motivated and technically sound.

- The empirical evaluation is thorough: It tests through a wide range of tasks, from toy dataset to real world settings. The proposed hybrid method shows better performance in all settings.

- The results show that, the simplest direct hybrid method shows great performance, this is some interesting observations and could be a strong baseline for future researchers.

**Weaknesses:**

- The joint and conditional hybrid approaches both encourage the learner summary statistics to be independent from expert summary statistics, but I did not see examples or results demonstrating the advantages or effects of the independence.

- Training extra networks could cause additional overhead and resources for hyperparameter tuning. These overheads should be discussed in more detail in the paper. (Although I expect the main overheads to come from the simulations?)

**Questions:**

For Eq. 8, is it possible to just use the using KL divergence rather than MMD? KL divergence could be easier to optimize and implement compared with MMD distance.

---

### Official Review · Reviewer_mwUj · 2023-10-31

**Soundness:** 3 good
**Presentation:** 3 good
**Contribution:** 2 fair
**Rating:** 5
**Confidence:** 4

**Summary:**

The authors present three approaches for creating hybrid (learned and hand-crafted) low-dimensional summaries of the observed data within the framework of amortized Bayesian inference. They then compare these approaches through numerical experiments.

**Strengths:**

**Originality.** The authors introduce three novel formulations for hybrid learned summary statistics.

**Quality and clarity.** The paper is generally well-written and well-structured, making it easy to follow.

**Significance.** Designing summary statistics is very important for amortized Bayesian inference.

**Weaknesses:**

* My main concern is that in these experiments the unknown is very low-dimensional, and it remains unclear if these methods are a viable option for high-dimensional inverse problems, e.g., 3D medical and geophysical imaging. ABI is particularly relevant in these high-dimensional problems where MCMC methods do not stand a chance in terms of scalability.

* Authors are specifically interested in low simulation budget settings. To this end, they use a "small" number of samples to train the neural networks and compare the performance of the methods. However, it is not clear how the performance of these methods changes as the number of training samples further decreases or increases. To put it differently, what constitutes a low simulation budget and how can authors make general conclusions about the performance of these methods  based on a fixed value of training samples? Ideally, the results should be presented as a function of the number of training samples.

* It is unclear how one go about choosing the summary statistics dimension in this method for real-world problems.

* Same goes for the weight parameter in the loss function (Eq. 8).

* I do not see any guarantees that any if these methods will eventually approximate the true posterior distribution. I.e., How do we know beforehand if a particular choice of summary network will allow the KL divergence term to go to zero?

**Questions:**

* Regarding hand-crafted summary statistics, especially for ABI problems where the likelihood is defined by PDEs, many papers out there use an initial estimation of the unknown model (e.g., least-squares or even applying the adjoint of the forward operator to data) instead of observed data for ABI. Can authors comments on this and provide more references to these type of methods.

* How sensitive is the final posterior estimate to the choice of summary network? Clearly, in the low simulation budget setting, we will not be able to derive the KL divergence to zero, so depending on the choice of summary network, the final posterior estimate will be different.

* These three methods should be able to recover the true posterior distribution in the very high simulation budget setting. An experiment with a very high number of training samples would be very helpful to demonstrate this.

---

### Official Review · Reviewer_qzC3 · 2023-11-01

**Soundness:** 2 fair
**Presentation:** 3 good
**Contribution:** 3 good
**Rating:** 6
**Confidence:** 3

**Summary:**

The paper proposes “expert-in-the-loop” approaches which combines expert and learned summary statistics for Amortized Bayesian inference problems. They propose three hybrid approaches: The first one directly conditions the neural approximation on both of the statistics. The second one learns to transform the expert statistics in the same lower dimensional representation as the learned representation. The third one learns a conditional model of learned representations given expert statistics. The paper evaluates these approaches on real data and simulated scenarios and compares them against directly learned or expert-based approaches. Hybrid approaches are shown to yield better estimations with the direct approach providing superior results especially in lower data regimes.

**Strengths:**

Providing robust estimates for lower data regimes is challenging and this paper provides a neat way to combine learnable representations of data with expert opinions. Particularly, the simplicity of the direct approach stands out. The paper is well written and organized.

**Weaknesses:**

The conclusions about the proposed results are sensitive to low/medium data regimes and I think the paper can benefit from explaining this better through more ablations.

**Questions:**

- The authors note that the conditional hybrid method might be the most challenging to optimize. Can you share your observations on this points for the experiments? Given that, the direct method seemed to perform the best, what do you think are the advantages of this approach?
- Given the sensitivity of the conclusions to low/medium data regimes, I am curious to understand at what point the learned representations start being favorable over the hybrid approaches. Do the authors have any ablations on this point for the experiments conducted in this paper?